# Maximum Likelihood Training of Parametrized Diffusion Model

## Abstract

Whereas diverse variations of diffusion models exist in image synthesis, the previous variations have not innovated on a topic of their diffusing mechanism that was always kept linear. Meanwhile, it is intuitive that there would be more promising diffusion patterns to result in a distribution closer to the approximating data distribution. This paper introduces such adaptive and nonlinear diffusion method for score-based diffusion models. Unlike static and linear Variance Exploring (VE) or Variance Preserving (VP) Stochastic Differential Equations (SDE) of the previous diffusion models, our Parametrized Diffusion Model (PDM) learns the optimal diffusion process by combining a normalizing flow and a diffusion process. Specifically, PDM utilizes the flow to nonlinearly transform a data variable into a latent variable, and PDM applies the diffusion process to the transformed latent variable with the linear diffusing mechanism. Subsequently, PDM enjoys a nonlinear and learned diffusion from the perspective of the data variable. We train PDM with a variational proxy of the log-likelihood, and we prove that variational gap between the variational bound and the log-likelihood becomes tight when the normalizing flow is optimal.

## 1 Introduction

Diffusion models have recently achieved successes on a task of sample generations, and some researches claim the state-of-the-art performance over Generative Adversarial Networks (GAN) (Karras et al., 2019). This success is highlighted particularly in the community of likelihood-based models, including normalizing flows (Grcić et al., 2021), autoregressive models (Parmar et al., 2018), and variational autoencoders (VAE) (Vahdat and Kautz, 2020). Moreover, this success is noteworthy because it is achieved merely using linear diffusing mechanisms, such as VESDEs (Song and Ermon, 2020) and VPSDEs (Ho et al., 2020).

This paper expands these linear diffusing mechanisms of VE/VPSDE to a data-adaptive trainable nonlinear diffusion. To motivate the expansion, though there are structural similarities between diffusion models and VAEs, the forward diffusion in a diffusion model has not been trained in existing literature while its counterpart, which is the encoder of VAE, is trainable. Rather, the current diffusion models assume the linear diffusing mechanism to be fixed throughout training procedure. Because of this static nature of the diffusing mechanism, *variational gap* (Cremer et al., 2018) between the log-likelihood and the Evidence Lower BOund (ELBO) remains to be strictly positive unless the score perfectly estimates the data score. This variational gap prevents the score training from being Maximum Likelihood Estimation (MLE). This gap is a fundamental motivation to develop a trainable encoder of the diffusion model, so that the gap can be tight.

As we tighten variational gap by training a nonlinear forward diffusion, the biggest challenge comes from an intractable optimization loss. The denoising diffusion loss requires a transition probability to be closed-form in order to achieve fast optimization, but a nonlinear diffusing mechanism, in general, has no closed-form perturbation probability. Hence, our innovation becomes designing a diffusion model with a tractable loss while the diffusion is learnable and flexible, which we achieve by merging previous diffusion models and the tractable variable transformation.

Our innovation concentrates on the theoretic and the practical aspects of a new diffusion, which is tractable, learnable, and flexible. Theoretically, we observe that a nonlinear diffusing mechanism can be transformed to a linear diffusion under an invertible transformation, and vice versa. In practice,

we implement PDM by merging a normalizing flow model and a diffusion model, which constructs a tractable, learnable, and flexible nonlinear diffusing mechanism. From this construction, we prove that variational gap can be further tightened by training the forward diffusion, nonlinearly. Also, we demonstrate the state-of-the-art performance in CelebA on Fréchet Inception Distance (FID) (Heusel et al., 2017). We summarize our key contributions, as below.

- PDM expands the scope of a forward diffusion process from linear and fixed dynamics to nonlinear and trainable dynamics.
- PDM minimizes variational gap between the log-likelihood and ELBO, lower than the gap of linear diffusion models because PDM learns a data-adaptive nonlinear diffusion.
- PDM explicitly proves in Theorem 2 that the optimal generative distribution equals to the real distribution, which was not the case for a linear diffusion model.

## 2 PRELIMINARY

A diffusion model is constructed with bidirectional stochastic processes. The *forward* direction diffuses an input variable to a noise variable, and the *reverse* direction denoises the random noise to construct a realistic input data instance. The diffusion model learns this reverse direction (generative process) by estimating data score. We provide a brief summary of diffusion models, as below.

**Forward Diffusion** At the beginning of model build-up, a diffusion process requires a diffusing mechanism on an input variable. This paper assumes that the diffusion process is governed by a SDE of $\mathrm{d}\mathbf{x}_t = \mathbf{f}(\mathbf{x}_t, t)\,\mathrm{d}t + \mathbf{G}(\mathbf{x}_t, t)\,\mathrm{d}\boldsymbol{\omega}_t$ with $\mathbf{x}_0 \sim p_r$, where $p_r$ is the distribution from a real-world dataset; and $\mathbf{x}_t$ is the solution of the SDE.

**Reverse Diffusion** The theory of stochastic calculus guarantees that we could create an identical diffusion process, $\{\mathbf{x}_t\}_{t=0}^T$, by solving an associated reverse SDE backwards in time (Anderson, 1982). The associated reverse SDE is

$$\mathrm{d}\mathbf{x}_t = \left[\mathbf{f}(\mathbf{x}_t, t) - \mathbf{G}(\mathbf{x}_t, t)\mathbf{G}^T(\mathbf{x}_t, t)\nabla_{\mathbf{x}_t} \log p_t(\mathbf{x}_t)\right]\mathrm{d}t + \mathbf{G}(\mathbf{x}_t, t)\,\mathrm{d}\bar{\boldsymbol{\omega}}_t, \quad \mathbf{x}_T \sim p_T, \quad (1)$$

where $\boldsymbol{\omega}_t$ and $\bar{\boldsymbol{\omega}}_t$ are standard Wiener processes with time flows forward and backward, respectively; and $p_t$ is a probability law of $\mathbf{x}_t$.

**Generative Diffusion** A diffusion model approximates the above SDE 1 to eventually yield an estimation on the data distribution of $p_r$. In SDE 1, previous literature setup that drift and diffusion terms, $\mathbf{f}$ and $\mathbf{G}$, are determined a-priori in the forward diffusion. However, data score, $\nabla_{\mathbf{x}_t} \log p_t(\mathbf{x}_t)$, is intractable to compute, so we estimate this data score with a score network of $\mathbf{s}_{\boldsymbol{\theta}}(\mathbf{x}_t, t)$ in order to mimic the reverse diffusion with our generative process. This score network approximates the reverse diffusion by plugging the estimated score in place of data score with the below generative process:

$$\mathrm{d}\mathbf{x}_t^{\boldsymbol{\theta}} = \left[\mathbf{f}(\mathbf{x}_t^{\boldsymbol{\theta}}, t) - \mathbf{G}(\mathbf{x}_t^{\boldsymbol{\theta}}, t)\mathbf{G}^T(\mathbf{x}_t^{\boldsymbol{\theta}}, t)\mathbf{s}_{\boldsymbol{\theta}}(\mathbf{x}_t^{\boldsymbol{\theta}}, t)\right]\mathrm{d}t + \mathbf{G}(\mathbf{x}_t^{\boldsymbol{\theta}}, t)\,\mathrm{d}\bar{\boldsymbol{\omega}}_t, \quad \mathbf{x}_T^{\boldsymbol{\theta}} \sim \pi. \quad (2)$$

The generative process starts from a prior distribution ($\pi$), and it constructs time-continuous random variables $\mathbf{x}_t^{\boldsymbol{\theta}}$ by solving a SDE 2 backwards in time. The generated stochastic process is denoted by $\{\mathbf{x}_t^{\boldsymbol{\theta}}\}_{t=0}^T$, and we omit $\boldsymbol{\theta}$ in the superscript if no confusion arises. With this generative process, we define a generative distribution, $\mathbf{x}_0^{\boldsymbol{\theta}} \sim p_{\boldsymbol{\theta}}$, as the probability density of the generated random variable at time $t = 0$.

**Score Estimation** We train the score network by a variational bound of the log-likelihood, given by

$$\mathbb{E}_{\mathbf{x}_0}\left[-\log p_{\boldsymbol{\theta}}(\mathbf{x}_0)\right] \leq \mathcal{L}\left(\{\mathbf{x}_t\}_{t=0}^T, \lambda = g^2; \boldsymbol{\theta}\right) - \mathbb{E}_{\mathbf{x}_T}\left[\log \pi(\mathbf{x}_T)\right]$$
$$= \int_0^T g^2(t)\mathcal{L}_t\left(\{\mathbf{x}_t\}_{t=0}^T; \boldsymbol{\theta}\right)\mathrm{d}t - \mathbb{E}_{\mathbf{x}_T}\left[\log \pi(\mathbf{x}_T)\right], \quad (3)$$

where $\mathcal{L}_t\left(\{\mathbf{x}_t\}_{t=0}^T; \boldsymbol{\theta}\right) = \mathbb{E}_{\mathbf{x}_0, \mathbf{x}_t}\left[\|\mathbf{s}_{\boldsymbol{\theta}}(\mathbf{x}_t, t) - \nabla_{\mathbf{x}_t} \log p_{0t}(\mathbf{x}_t|\mathbf{x}_0)\|_2^2\right]$ up to a constant, where $p_{0t}(\mathbf{x}_t|\mathbf{x}_0)$ is a transition probability from $\mathbf{x}_0$ to $\mathbf{x}_t$. Here, $\lambda$ is a weighting function that determines the level of contribution for each diffusion time on the overall diffusion loss, $\mathcal{L}(\{\mathbf{x}_t\}_{t=0}^T, \lambda; \boldsymbol{\theta})$ (Song et al., 2020). Variational bound holds when the weighting function is the likelihood weighting ($g$) (Song et al., 2021), where $\mathbf{G}$ is a scalar-valued $g$ function.

## 3 MOTIVATION OF NONLINEAR DIFFUSING MECHANISM

Though it has long been theoretically and empirically grounded to train the encoder part in VAE, such solid ground is not accomplished in diffusion models. This section analyzes structural similarities between a VAE model and a diffusion model, which brings the foundation of a data-adaptive nonlinear diffusing mechanism.

### 3.1 VARIATIONAL GAP OF VAE

Given Negative ELBO (NELBO) of Negative Log-Likelihood (NLL) in VAE as

$$-\log p_{\boldsymbol{\theta}}(\mathbf{x}) \leq \mathbb{E}_{q_{\boldsymbol{\phi}}(\mathbf{z}|\mathbf{x})}\big[-\log p_{\boldsymbol{\theta}}(\mathbf{x}|\mathbf{z})\big] + D_{KL}\big(q_{\boldsymbol{\phi}}(\mathbf{z}|\mathbf{x})\|p(\mathbf{z})\big) \tag{4}$$

$$= -\log p_{\boldsymbol{\theta}}(\mathbf{x}) + D_{KL}\big(q_{\boldsymbol{\phi}}(\mathbf{z}|\mathbf{x})\|p_{\boldsymbol{\theta}}(\mathbf{z}|\mathbf{x})\big), \tag{5}$$

we have a pair of interpretations on this NELBO. First, when we focus on Eq. 4, NELBO 1) aids data reconstruction by optimizing $\mathbb{E}_{q_{\boldsymbol{\phi}}(\mathbf{z}|\mathbf{x})}\big[-\log p_{\boldsymbol{\theta}}(\mathbf{x}|\mathbf{z})\big]$; and NELBO 2) regularizes the inference distribution of the encoder to a prior distribution by $D_{KL}\big(q_{\boldsymbol{\phi}}(\mathbf{z}|\mathbf{x})\|p(\mathbf{z})\big)$. On the other hand, if we concentrate on Eq. 5, NELBO bounds NLL by approximating an intractable decoder posterior of $p_{\boldsymbol{\theta}}(\mathbf{z}|\mathbf{x})$ with a tractable encoder posterior of $q_{\boldsymbol{\phi}}(\mathbf{z}|\mathbf{x})$. A vanilla VAE (Kingma and Welling, 2013) assumes this approximate posterior to be a Gaussian distribution with mean and diagonal covariance estimated by amortized inference: $q_{\boldsymbol{\phi}}(\mathbf{z}|\mathbf{x}) = \mathcal{N}\big(\mathbf{z}; \mu_{\boldsymbol{\phi}}(\mathbf{x}), \sigma^2_{\boldsymbol{\phi}}(\mathbf{x})\mathbf{I}\big)$. By expanding the flexibility of this approximate posterior into a variational family of general distributions, for instance, Rezende and Mohamed (2015) resulted in tighter NELBO, which leads the optimization of VAE closer to MLE, and their choice of a generalizable model was normalizing flow.

To connect this NELBO to a diffusion loss, we restate NELBO in the language of a stochastic process. Having that VAE attains a stochastic process of bivariate random variables, $\{\mathbf{x}, \mathbf{z}\}$, NELBO is reformulated to a KL divergence between two joint distributions modeled in bidirectional ways:

$$
\begin{aligned}
D_{KL}(p_r\|p_{\boldsymbol{\theta}}) \leq & D_{KL}(p_r\|p_{\boldsymbol{\theta}}) + \mathbb{E}_{p_r(\mathbf{x})}\big[D_{KL}(q_{\boldsymbol{\phi}}(\mathbf{z}|\mathbf{x})\|p_{\boldsymbol{\theta}}(\mathbf{z}|\mathbf{x}))\big] \\
= & \mathbb{E}_{p_r(\mathbf{x})}\left[\log \frac{p_r(\mathbf{x})}{p_{\boldsymbol{\theta}}(\mathbf{x})} + \mathbb{E}_{q_{\boldsymbol{\phi}}(\mathbf{z}|\mathbf{x})}\left[\log \frac{q_{\boldsymbol{\phi}}(\mathbf{z}|\mathbf{x})}{p_{\boldsymbol{\theta}}(\mathbf{z}|\mathbf{x})}\right]\right] \\
= & \mathbb{E}_{p_r(\mathbf{x})q_{\boldsymbol{\phi}}(\mathbf{z}|\mathbf{x})}\left[\log \frac{p_r(\mathbf{x})q_{\boldsymbol{\phi}}(\mathbf{z}|\mathbf{x})}{p_{\boldsymbol{\theta}}(\mathbf{x})p_{\boldsymbol{\theta}}(\mathbf{z}|\mathbf{x})}\right] \\
= & D_{KL}(q_{\boldsymbol{\phi}}(\mathbf{x}, \mathbf{z})\|p_{\boldsymbol{\theta}}(\mathbf{x}, \mathbf{z})),
\end{aligned}
\tag{6}
$$

The inequality 6, i.e., $D_{KL}(p_r\|p_{\boldsymbol{\theta}}) \leq D_{KL}(q_{\boldsymbol{\phi}}(\mathbf{x}, \mathbf{z})\|p_{\boldsymbol{\theta}}(\mathbf{x}, \mathbf{z}))$, is well-known by itself in the field of information theory (Duchi, 2016) by the name of *data processing inequality*. This restated bound interprets that VAE is essentially a bimodeling approach of a joint distribution on the bivariate stochastic process, $\{\mathbf{x}, \mathbf{z}\}$. On the forward direction $(\mathbf{x} \rightarrow \mathbf{z})$, a latent variable is conditioned on a data variable, and the joint distribution is modeled by $q_{\boldsymbol{\phi}}(\mathbf{x}, \mathbf{z}) = p_r(\mathbf{x})q_{\boldsymbol{\phi}}(\mathbf{z}|\mathbf{x})$. On the reverse direction $(\mathbf{z} \rightarrow \mathbf{x})$, a generative data variable is conditioned on the latent variable, and the joint distribution is modeled by $p_{\boldsymbol{\theta}}(\mathbf{x}, \mathbf{z}) = p(\mathbf{z})p_{\boldsymbol{\theta}}(\mathbf{x}|\mathbf{z})$. Under this equivalent framework, we present an analytic tool to measure the closeness of NELBO and NLL (see Table 5) by

$$
\begin{aligned}
\text{Gap}(q_{\boldsymbol{\phi}}, p_{\boldsymbol{\theta}}) = & \text{NELBO} - \text{NLL} \\
= & D_{KL}(q_{\boldsymbol{\phi}}(\mathbf{x}, \mathbf{z})\|p_{\boldsymbol{\theta}}(\mathbf{x}, \mathbf{z})) - D_{KL}(p_r\|p_{\boldsymbol{\theta}}) \\
= & \mathbb{E}_{p_r(\mathbf{x})}\big[D_{KL}(q_{\boldsymbol{\phi}}(\mathbf{z}|\mathbf{x})\|p_{\boldsymbol{\theta}}(\mathbf{z}|\mathbf{x}))\big],
\end{aligned}
$$

and we denote this quantity as *variational gap* (Cremer et al., 2018). Though it is one of central tasks in VAE to minimize this variational gap (Kingma et al., 2016), only limited works (Burda et al., 2015; Neal, 2001) have estimated variational gap due to its computational burden. We introduce the tractable computation of variational gap in the next section on diffusion models.

### 3.2 VARIATIONAL GAP OF DIFFUSION MODELS

This section constructs an analogy of a diffusion model to VAE. To begin with, we observe that the classical data processing inequality in VAE holds on a stochastic process with bivariate random variables on a finite-dimensional Euclidean space. On the other hand, the time horizon of a diffusion

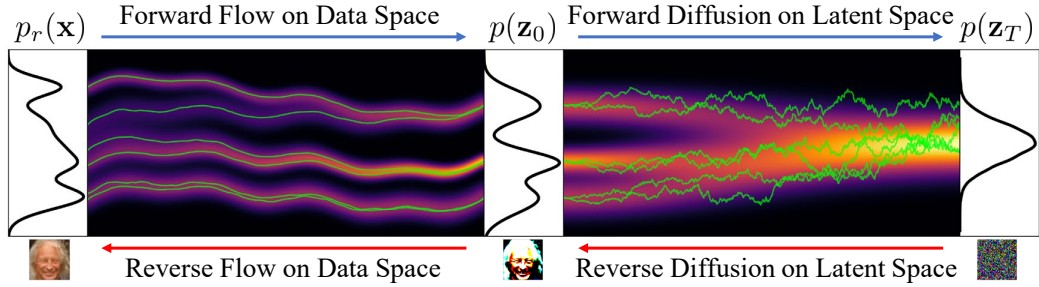

Figure 1: Schematic view of PDM. Normalizing flow transforms a data variable to a latent variable, and a linear diffusion (VE/VPSDE) diffuses the latent variable to a prior distribution. Data generation starts from the prior distribution, and the generation denoises randomly generated noise to a latent variable, and the generation eventually transforms the latent variable into the data space.

model is continuous, so it contains an infinite number of random variables in its stochastic process as $\{\mathbf{x}_t\}_{t=0}^T$. Fortunately, the finite-dimensional classical data processing inequality on the bivariate random variables is directly extendable to a stochastic process with a continuous index. Formally, an infinite-variate joint distribution is called a path measure (Oksendal, 2013), which is defined as a probability law on a space of continuous functions induced by $\{\mathbf{x}_t\}_{t=0}^T$. Therefore, the path measure is an alternative concept of a joint distribution for a stochastic process with a continuous index. The path measure for the forward diffusion in a diffusion model corresponds to a joint distribution for the encoder part in VAE, and similarly, the path measure for a generative diffusion corresponds to the decoder part in VAE. If $\boldsymbol{\mu}$ is the forward (inference) path measure, and if $\boldsymbol{\nu_\theta}$ is the generative path measure; then the data processing inequality becomes

$$D_{KL}(p_r \| p_{\boldsymbol{\theta}}) \leq D_{KL}(\boldsymbol{\mu} \| \boldsymbol{\nu_\theta}).$$

The real distribution of $p_r$ is a marginal distribution of $\boldsymbol{\mu}$ at $t = 0$, and the generative distribution of $p_{\boldsymbol{\theta}}$ becomes a marginal distribution of $\boldsymbol{\nu_\theta}$ at $t = 0$. Under this concept, the data processing inequality means that a distributional divergence of $D_{KL}(\boldsymbol{\mu} \| \boldsymbol{\nu_\theta})$ between two path measures lower bounded by a divergence of $D_{KL}(p_r \| p_{\boldsymbol{\theta}})$ between two *sliced* path measures at $t = 0$.

Previous works (Song et al., 2021; Huang et al., 2021) regard this variational bound being equivalent to a diffusion loss,

$$D_{KL}(\boldsymbol{\mu} \| \boldsymbol{\nu_\theta}) = \mathcal{L}\big(\{\mathbf{x}_t\}_{t=0}^T, g^2; \boldsymbol{\theta}\big) + D_{KL}(p_T \| \pi),$$

where $\mathbf{G}(\mathbf{x}_t, t) = g(t)$ is a scaler function on $t$, and $\lambda(t)$ is a square of the diffusion term $g^2(t)$. Since the previous diffusion models optimize the diffusion loss, the KL divergence of $D_{KL}(\boldsymbol{\mu} \| \boldsymbol{\nu_\theta})$ is an equivalent target of optimization on the diffusion loss. Subsequently, we can credit variational gap as a training performance to tighten NELBO to NLL by

$$\text{Gap}(\boldsymbol{\mu}, \boldsymbol{\nu_\theta}) = D_{KL}(\boldsymbol{\mu} \| \boldsymbol{\nu_\theta}) - D_{KL}(p_r \| p_{\boldsymbol{\theta}}).$$

This variational gap satisfies (see Appendix B.2)

$$\text{Gap}(\boldsymbol{\mu}, \boldsymbol{\nu_\theta}) = \mathbb{E}_{p_r(\mathbf{x}_0)}\Big[D_{KL}\big(\boldsymbol{\mu}(\{\mathbf{x}_t\}|\mathbf{x}_0) \| \boldsymbol{\nu_\theta}(\{\mathbf{x}_t\}|\mathbf{x}_0)\big)\Big],$$

so variational gap is minimized if an inference posterior, $\boldsymbol{\mu}(\{\mathbf{x}_t\}|\mathbf{x}_0)$, estimates the intractable generative posterior, $\boldsymbol{\nu_\theta}(\{\mathbf{x}_t\}|\mathbf{x}_0)$. However, unlike VAE, the inference posterior is not trainable, which restricts variational gap to be strictly positive in previous diffusion models. We note that a diffusion model provides tractable NLL using the Instantaneous Change of Variable (Song et al., 2020), and this tractable variational gap becomes useful in analyzing a diffusion model quality.

## 4 PARAMETRIZED DIFFUSION MODEL

### 4.1 NO CLOSED-FORM TRANSITION PROBABILITY ON NONLINEAR DIFFUSION

As far as we know, all previously proposed SDEs, including VESDE and VPSDE, are categorized as linear SDEs because they have linear drift term of $\mathbf{f}(\mathbf{x}_t, t)$. This drift term determines the dynamics of the diffusion, so a data-adaptive diffusion could be attained simply by parametrizing drift and diffusion terms. However, a transition probability with nonlinear coefficients for a SDE, $p_{0t}(\mathbf{x}_t|\mathbf{x}_0)$, is intractable in general, and a diffusion loss becomes intractable to optimize. In fact, it is worth noting that a closed-form of the transition probability exists only on a certain type of SDEs, like VESDE and VPSDE.

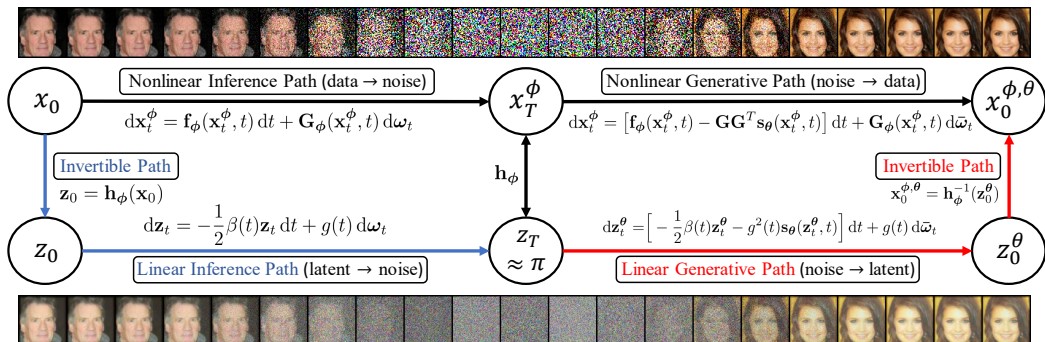

Figure 2: PDM attains a ladder structure between a data space and a latent space by the invertibility of normalizing flow. A score network estimates data score following blue arrows, and a model generates images following red arrows.

## 4.2 PARAMETRIZED FORWARD DIFFUSION WITH NORMALIZING FLOW

To construct a tractable transition probability while keeping nonlinearity, we observe that a nonlinear diffusion can be decomposed into a linear diffusion and a nonlinear invertible transformation, where the transformation provides the flexible nonlinearity on a diffusion process. Figure 1 and 2 provide a schematic view of our model. Normalizing flow transforms a data input to a latent variable, and a diffusion model linearly permeates the latent variable to a noise distribution.

Concretely, let us define $\mathbf{z}_0$ to be a transformed latent variable, $\mathbf{z}_0 = \mathbf{h}_\phi(\mathbf{x}_0)$, where $\phi$ is flow model parameters. Then, a canonical diffusion with linear dynamics of

$$\mathrm{d}\mathbf{z}_t = -\frac{1}{2}\beta(t)\mathbf{z}_t\,\mathrm{d}t + g(t)\,\mathrm{d}\boldsymbol{\omega}_t, \quad \mathbf{z}_0 = \mathbf{h}_\phi(\mathbf{x}_0) \text{ with } \mathbf{x}_0 \sim p_r,$$

describes a diffusing mechanism of a latent variable. Since the transformation is invertible, if we define an inverse random variable by $\mathbf{x}_t^\phi = \mathbf{h}_\phi^{-1}(\mathbf{z}_t)$, the dynamics of a stochastic process of $\{\mathbf{x}_t^\phi\}_{t=0}^T$ is governed by a SDE of

$$\mathrm{d}\mathbf{x}_t^\phi = \mathbf{f}_\phi(\mathbf{x}_t^\phi, t)\,\mathrm{d}t + \mathbf{G}_\phi(\mathbf{x}_t^\phi, t)\,\mathrm{d}\boldsymbol{\omega}_t, \quad \mathbf{x}_0^\phi \sim p_r,$$

where drift and diffusion terms are derived by

$$\begin{cases} \mathbf{f}_\phi(\mathbf{x}_t, t) & = -\frac{1}{2}\beta\big[\nabla\mathbf{h}_\phi\big]^{-1}\mathbf{h}_\phi + \frac{1}{2}g^2\mathrm{tr}\big(\nabla^2\mathbf{h}_\phi^{-1}\big) \\ \mathbf{G}_\phi(\mathbf{x}_t, t) & = g\big[\nabla\mathbf{h}_\phi\big]^{-1}, \end{cases}$$

from the Ito's lemma (Oksendal, 2013). See full derivation and explanation in Appendix B.1. This diffusion process is *induced diffusion* on the data space that is defined indirectly by transforming back from the latent space to the data space. The nonlinearity of this induced diffusion fully depends on the invertible transformation as illustrated in Figure 2.

It is not straightforward to directly construct a generative reverse process on the data space, $\mathbf{x}_t^{\phi,\boldsymbol{\theta}}$. Figures 1 and 2 build the generative process by red arrows using the transformation. Starting from a prior distribution of $\pi$, we denoise a random variable from the prior to $\mathbf{z}_0^{\boldsymbol{\theta}}$ according to the generative process of the linear diffusion on the latent space, and we transform the fully denoised latent of $\mathbf{z}_0^{\boldsymbol{\theta}}$ back to the data space $\mathbf{x}_0^{\phi,\boldsymbol{\theta}} = \mathbf{h}_\phi^{-1}(\mathbf{z}_0^{\boldsymbol{\theta}})$.

## 4.3 TRACTABLE VARIATIONAL BOUND OF PARAMETRIZED DIFFUSION MODEL

The joint modeling of normalizing flow and diffusion enables a tractable training of nonlinear dynamics. The log-likelihood on a data space reduces to the log-likelihood on a latent space by adding the log determinant of Jacobian to the latent log-likelihood. This calculation of log-likelihoods provides Theorem 1 that bounds the log-likelihood with a tractable ELBO.

**Theorem 1.** *Suppose that $p_{\phi,\boldsymbol{\theta}}(\mathbf{x}_0)$ is the log-likelihood of a generative random variable $\mathbf{x}_0^{\phi,\boldsymbol{\theta}}$. Then, the log-likelihood is bounded by*

$$\mathbb{E}_{p_r(\mathbf{x}_0)}\big[-\log p_{\phi,\boldsymbol{\theta}}(\mathbf{x}_0)\big] \leq \mathcal{L}\big(\{\mathbf{x}_t\}_{t=0}^T, g^2; \{\phi,\boldsymbol{\theta}\}\big),$$

*where*

$$\mathcal{L}\big(\{\mathbf{x}_t\}_{t=0}^T, g^2; \{\boldsymbol{\phi}, \boldsymbol{\theta}\}\big) = -\mathbb{E}_{p_r(\mathbf{x}_0)}\Big[\log\Big|\det\Big(\frac{\partial\mathbf{h}_{\boldsymbol{\phi}}}{\partial\mathbf{x}_0}\Big)\Big|\Big] + \mathcal{L}\big(\{\mathbf{z}_t\}_{t=0}^T, g^2; \boldsymbol{\theta}\big) - \mathbb{E}_{\mathbf{z}_T}\big[\log\pi(\mathbf{z}_T)\big],$$
(7)

*with $\mathcal{L}\big(\{\mathbf{z}_t\}_{t=0}^T, g^2; \boldsymbol{\theta}\big) = \int_0^T g^2(t)\mathbb{E}_{\mathbf{z}_0,\mathbf{z}_t}\big[\|\mathbf{s}_{\boldsymbol{\theta}}(\mathbf{z}_t, t) - \nabla_{\mathbf{z}_t}\log p_{0t}(\mathbf{z}_t|\mathbf{z}_0)\|_2^2\big]\,\mathrm{d}t$, up to a constant. Here, $p_{0t}(\mathbf{z}_t|\mathbf{z}_0)$ is a transition probability of the linear diffusion process in the latent space.*

PDM ELBO in Eq. 7 is different from the vanilla diffusion ELBO in Eq. 3 from three perspectives. First, the normalizing flow loss, i.e., the log determinant of Jacobian, is added to the loss of PDM ELBO. This normalizing flow loss represents a transformation of a data space into a latent space. Second, the remaining terms, related to the diffusion model, is a linear diffusion on the latent space with linear dynamics, which provides tractable optimization, rather than an intractable data diffusion of nonlinear dynamics. Third, the optimization loss in Eq. 7 estimates the score on the latent space by a score network of $\mathbf{s}_{\boldsymbol{\theta}}(\mathbf{z}_t, t)$, while the diffusion loss in Eq. 3 estimates the score on the data space by a score network of $\mathbf{s}_{\boldsymbol{\theta}}(\mathbf{x}_t, t)$.

### 4.4 OPTIMIZATION LOSSES

Previous works (Ho et al., 2020; Song et al., 2021; Vahdat et al., 2021) have empirically demonstrated that training a diffusion model with a likelihood weighting, $\lambda(t) = g^2(t)$, could bring worse performances than applying other weighting functions. For instance, Ho et al. (2020); Song and Ermon (2020) introduce a variance weighting function, $\lambda(t) = \sigma^2(t)$, and Ho et al. (2020) show that this weighting outperforms a likelihood weighting in terms of the sample generation.

Within our framework, we also observe that a variance weighting function is advantageous over a likelihood weighting. Henceforth, we optimize a flow model and a diffusion model with different losses as follows:

$$\text{Flow:} \quad \min_{\boldsymbol{\phi}} \mathcal{L}(\{\mathbf{x}_t\}_{t=0}^T, g^2; \{\boldsymbol{\phi}, \boldsymbol{\theta}\})$$

$$\text{Diffusion:} \quad \min_{\boldsymbol{\theta}} \mathcal{L}(\{\mathbf{z}_t\}_{t=0}^T, \{g^2 \text{ or } \sigma^2\}; \boldsymbol{\theta}),$$

Note that the first term of the log Jacobian and the third term of the cross entropy in Eq. 7 are independent to score parameters, $\boldsymbol{\theta}$. Therefore, diffusion parameters are not influenced by these terms, in which only the second term remains to update the diffusion parameters.

### 4.5 TIGHT VARIATIONAL GAP OF PARAMETRIZED FORWARD DIFFUSION

This section theoretically analyzes variational gap of PDM. To construct variational gap of PDM, we enumerate few nomenclatures for path measures (see Table 6). On a data space, $\boldsymbol{\mu}_{\boldsymbol{\phi}}^d$ is a path measure for an *inference* process; and $\boldsymbol{\nu}_{\boldsymbol{\phi},\boldsymbol{\theta}}^d$ is a path measure for a *generative* process. On a latent space, $\boldsymbol{\mu}_{\boldsymbol{\phi}}^l$ is a path measure for an *inference* process; and $\boldsymbol{\nu}_{\boldsymbol{\theta}}^l$ is a path measure for a *generative* process. Here, $\boldsymbol{\nu}_{\boldsymbol{\theta}}^l$ does not depend on $\boldsymbol{\phi}$ because the generative diffusion on the latent space starts from the $\boldsymbol{\phi}$-independent noise distribution ($\pi$); and because the diffusion proceeds a denoising process by following a score estimation of $\mathbf{s}_{\boldsymbol{\theta}}(\mathbf{z}_t, t)$ that depends only on $\boldsymbol{\theta}$.

Variational gap on the data space becomes

$$\text{Gap}_d(\boldsymbol{\mu}_{\boldsymbol{\phi}}^d, \boldsymbol{\nu}_{\boldsymbol{\phi},\boldsymbol{\theta}}^d) = \text{NELBO} - \text{NLL} \tag{8}$$

$$= D_{KL}(\boldsymbol{\mu}_{\boldsymbol{\phi}}^d \| \boldsymbol{\nu}_{\boldsymbol{\phi},\boldsymbol{\theta}}^d) - D_{KL}(p_r \| p_{\boldsymbol{\phi},\boldsymbol{\theta}}), \tag{9}$$

where the subscript in $\text{Gap}_d$ represents the data space. Recall that training the encoder in VAE guarantees tighter variational gap because the encoder distribution of $q_{\boldsymbol{\phi}}(\mathbf{x}, \mathbf{z})$ approaches toward the decoder distribution of $p_{\boldsymbol{\theta}}(\mathbf{x}, \mathbf{z})$ by minimizing NELBO of $D_{KL}\big(q_{\boldsymbol{\phi}}(\mathbf{x}, \mathbf{z}) \| p_{\boldsymbol{\theta}}(\mathbf{x}, \mathbf{z})\big)$. However, this analogy is not valid in variational gap in Eq. 8 because both inference and generative path measures depend on $\boldsymbol{\phi}$. This means that the generative path measure could arbitrarily deviate from the inference path measure when we optimize $\boldsymbol{\phi}$.

Using the change of variable, however, variational gap is equivalent to

$$\text{Gap}_d(\boldsymbol{\mu}_{\boldsymbol{\phi}}^d, \boldsymbol{\nu}_{\boldsymbol{\phi},\boldsymbol{\theta}}^d) = D_{KL}(\boldsymbol{\mu}_{\boldsymbol{\phi}}^d \| \boldsymbol{\nu}_{\boldsymbol{\phi},\boldsymbol{\theta}}^d) - D_{KL}(p_r \| p_{\boldsymbol{\phi},\boldsymbol{\theta}})$$
$$= D_{KL}(\boldsymbol{\mu}_{\boldsymbol{\phi}}^l \| \boldsymbol{\nu}_{\boldsymbol{\theta}}^l) - D_{KL}(p_{\boldsymbol{\phi}} \| p_{\boldsymbol{\theta}})$$
$$= \text{Gap}_l(\boldsymbol{\mu}_{\boldsymbol{\phi}}^l, \boldsymbol{\nu}_{\boldsymbol{\theta}}^l).$$

With this derivation, optimizing an invertible transformation guarantees tighter variational gap because the generative measure of $\boldsymbol{\nu}_{\boldsymbol{\theta}}^l$ is now fixed on $\boldsymbol{\phi}$. Also, we note that this variational gap reduces to a KL divergence between the inference posterior and the generative posterior by

$$\text{Gap}_d(\boldsymbol{\mu}_{\boldsymbol{\phi}}^d, \boldsymbol{\nu}_{\boldsymbol{\phi},\boldsymbol{\theta}}^d) = \text{Gap}_l(\boldsymbol{\mu}_{\boldsymbol{\phi}}^l, \boldsymbol{\nu}_{\boldsymbol{\theta}}^l) = \mathbb{E}_{\mathbf{z}_0}\big[D_{KL}\big(\boldsymbol{\mu}_{\boldsymbol{\phi}}(\{\mathbf{z}_t\}|\mathbf{z}_0) \| \boldsymbol{\nu}_{\boldsymbol{\theta}}(\{\mathbf{z}_t\}|\mathbf{z}_0)\big)\big], \quad (10)$$

which provides another analogy to VAE. We enumerate variational gaps of the explained models in Table 5. In fact, we show in Theorem 2 that the optimum of PDM guarantees that the generative distribution $p_{\boldsymbol{\phi}^*,\boldsymbol{\theta}}$ equals to the data distribution $p_r$, and variational gap reduces to zero in the optimum. See Appendix A for proofs.

**Theorem 2.** *Suppose a flow network is flexible enough to transform $p_r$ to arbitrary continuous distribution. For any $\boldsymbol{\theta}$, if $\nabla.\mathbf{s}_{\boldsymbol{\theta}}(\cdot, t)$ is symmetric for any $t \in [0, T]$ and $\nabla.\log \pi(\cdot) \equiv \mathbf{s}_{\boldsymbol{\theta}}(\cdot, T)$, then there exists $\boldsymbol{\phi}^*$ such that $\text{Gap}_d(\boldsymbol{\mu}_{\boldsymbol{\phi}^*}^d, \boldsymbol{\nu}_{\boldsymbol{\phi}^*,\boldsymbol{\theta}}^d) = 0$. Furthermore, we have*

$$\boldsymbol{\phi}^* = \arg\min_{\boldsymbol{\phi}} \text{Gap}_d(\boldsymbol{\mu}_{\boldsymbol{\phi}}^d, \boldsymbol{\nu}_{\boldsymbol{\phi},\boldsymbol{\theta}}^d) = \arg\min_{\boldsymbol{\phi}} D_{KL}(\boldsymbol{\mu}_{\boldsymbol{\phi}}^d \| \boldsymbol{\nu}_{\boldsymbol{\phi},\boldsymbol{\theta}}^d) = \arg\min_{\boldsymbol{\phi}} D_{KL}(p_r \| p_{\boldsymbol{\phi},\boldsymbol{\theta}}).$$

To the best of our knowledge, Theorem 2 is the first analysis in the community of diffusion models that the theoretic optimum of a model completely recovers the data distribution. To proceed this argument, let us consider a linear diffusion model in a data space with no flow model. Then, a marginal distribution of the forward process is $p_T$ at $t = T$, which differs from a prior distribution of $\pi$. In addition, suppose that the diffusion model reaches to the optimum point, i.e., its score network estimates the exact data score by $\mathbf{s}_{\boldsymbol{\theta}^*}(\mathbf{x}_t, t) = \nabla_{\mathbf{x}_t} \log p_t(\mathbf{x}_t)$. Then, a generative transition probability equals to the groundtruth (reverse) transition probability of $p_{T0}(\mathbf{x}_0|\mathbf{x}_T)$. Therefore, the generative distribution, $p_{\boldsymbol{\theta}^*}(\mathbf{x}_0) = \int \pi(\mathbf{x}_T) p_{T0}(\mathbf{x}_0|\mathbf{x}_T) \, d\mathbf{x}_T$, naturally diverges from the real distribution, $p_r(\mathbf{x}_0) = \int p_T(\mathbf{x}_T) p_{T0}(\mathbf{x}_0|\mathbf{x}_T) \, d\mathbf{x}_T$, because of the mismatch of $p_T$ and $\pi$. This implies that the original linear diffusion model cannot estimate the real distribution, $p_r(\mathbf{x}_0)$. In contrast to the original diffusion model, applying normalizing flow to a diffusion model enables PDM to recover the real distribution by Theorem 2. See Figures 8 and 9 for further illustration.

## 5 RELATED WORK

Recently, a diffusion model has been jointly modeled in conjunction with other deep generative models. There are two major previous works on this line of research. First, Vahdat et al. (2021) (LSGM) in

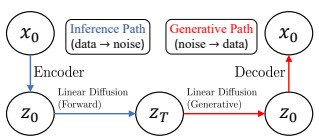

Figure 3: LSGM.

Figure 3 applies a diffusion model to the latent space of VAE. Since VAE does not attain an invertible structure, LSGM is not suggesting the expansion of diffusing mechanisms. In other words, LSGM cannot provide diffused data learning at the middle of the process because LSGM does not induce the diffusion process on the data space due to the lack of VAE's invertibility. Consequently, the analyzing tool for LSGM should be variational gap of VAE, rather than that of the diffusion model.

Second, Song et al. (2021) (ScoreFlow) in Figure 4 simply concatenate normalizing flow on top of a diffusion model to provide a dequantized data distribution only at the initial diffusion time. Despite of the structural similarity of ScoreFlow and PDM, a major difference of ScoreFlow and PDM lies on their density model constructions: $p_{\boldsymbol{\theta}}$ in Scoreflow and $p_{\boldsymbol{\phi},\boldsymbol{\theta}}$ in PDM. ScoreFlow constructs

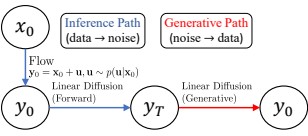

Figure 4: ScoreFlow.

$p_{\boldsymbol{\theta}}$ only with a diffusion model, so the flow and the diffusion networks are not jointly combined in the loss function of $\int p_r(\mathbf{x}) \int_{[0,1)^d} q_{\boldsymbol{\phi}}(\mathbf{u}|\mathbf{x}) \log \frac{p_{\boldsymbol{\theta}}(\mathbf{x}+\mathbf{u})}{q_{\boldsymbol{\phi}}(\mathbf{u}|\mathbf{x})} \, d\mathbf{u} \, d\mathbf{x}$, where $\mathbf{u}$ is a dequantization variable that lies in a unit cube, $[0, 1)^d$. In other words, ScoreFlow adopts the flow *outside* of a model density, so the flow is independent to the model density. In fact, ScoreFlow can be best understood as a dequantization technique (Ho et al., 2019), whose goal is estimating the optimal dequantizing region

Table 1: Performance comparison on CIFAR-10.

| SDE | Name | Model | NLL | NELBO | Gap | FID |
|---|---|---|---|---|---|---|
| VE | NCSN++ | Diff Baseline | 3.45 | 4.43 | 0.98 | **2.20** |
|  | PDM (FID) | Diff & Flow | **3.04** | **3.09** | **0.05** | 3.36 |
| VP | DDPM++ (FID) | Diff Baseline | 3.13 | 3.29 | 0.16 | **3.16** |
|  | PDM (FID) | Diff & Flow | **3.04** | **3.09** | **0.05** | 6.79 |
|  | PDM (FID, dec) | Diff & Flow-v2 | 3.11 | 3.17 | 0.06 | 4.43 |
|  | DDPM++ (NLL) | Diff Baseline | 3.05 | 3.22 | 0.17 | 9.68 |
|  | PDM (NLL) | Diff & Flow | 2.98 | **2.99** | **0.01** | 9.29 |
|  | PDM (NLL, dec) | Diff & Flow-v2 | **2.96** | **2.99** | 0.03 | **7.23** |
|  | DDPM++ (NLL, IS) | Diff Baseline | **2.91** | 3.10 | 0.19 | **5.70** |
|  | PDM (NLL, IS) | Diff & Flow | 2.99 | 3.01 | 0.02 | 8.94 |
|  | PDM (NLL, dec, IS) | Diff & Flow-v2 | 2.94 | **2.95** | **0.01** | 6.84 |
|  | LSGM-109M (balanced) | Diff & VAE | - | 2.96 | - | 4.60 |
|  | LSGM-480M (balanced) | Diff & VAE | - | 2.95 | - | 2.17 |
|  | LSGM-480M (NLL) | Diff & VAE | - | **2.87** | - | 6.89 |
|  | LSGM-480M (FID) | Diff & VAE | - | 3.43 | - | **2.10** |

Table 2: Performance comparison on CelebA.

| Model | NLL | NELBO | Gap | FID |
|---|---|---|---|---|
| NCSN++ (VE) | 2.39 | 3.96 | 1.57 | 3.95 |
| PDM (VE, FID) | **2.00** | **2.09** | **0.09** | **2.50** |
| DDPM++ (VP) | **1.79** | 2.19 | 0.40 | 3.03 |
| PDM (VP, FID) | 2.04 | **2.10** | **0.06** | **2.23** |
| UDM | **1.93** | - | - | **2.78** |
| DDGM | - | - | - | 2.92 |
| CR-NVAE | - | **1.86** | - | - |
| NCP-VAE | - | - | - | 5.25 |
| DenseFlow-74-10 | 1.99 | - | - | - |
| Styleformer | - | - | - | 3.66 |

on $[0, 1)^d$ to minimize dequantization gap (Hoogeboom et al., 2020). Therefore, variational gap of ScoreFlow remains to be strictly positive because training the flow does not optimize the forward diffusing mechanism. On the other hand, PDM constructs flow-embedded model density of $p_{\phi,\theta}$, and PDM optimizes the loss of $\int p_r(\mathbf{x}) \log \int_{[0,1)^d} p_{\phi,\theta}(\mathbf{x} + \mathbf{u}) \, \mathrm{d}\mathbf{u} \, \mathrm{d}\mathbf{x}$ that puts normalizing flow *inside* of the model density. Although we train PDM with a uniform dequantization (Theis et al., 2015) by default, we could apply the variational dequantization to PDM as well.

# 6 EXPERIMENTS

This section quantitatively and qualitatively analyzes suggested PDM on CIFAR-10 (Krizhevsky et al., 2009) and CelebA (Liu et al., 2015) $64 \times 64$. We compute NLL for density estimations in bits per dimension scale by solving probability flow (Song et al., 2020) backwards in time, and we compare baseline performances of Song et al. (2020; 2021) computed on a uniform dequantization. For sample generations, we use the Predictor-Corrector algorithm (Song et al., 2020) for PDM on VESDE, and we use the numerical ODE sampler of the probability flow (Song et al., 2020) with the RK45 method on VPSDE. Throughout experimental results, we use NCSN++ (VE) and DDPM++ (VP) (Song et al., 2020) as backbones of diffusion models, and a pair of ResNet-based flow models (Chen et al., 2019; Ma et al., 2020) as backbones of flow models. We give a full description on models for our experiments in Appendix C. We name PDM (FID) and PDM (NLL) for experiments that use weighting functions as $\sigma^2$ and $g^2$, respectively.

## 6.1 QUANTITATIVE RESULTS

We compare PDM to baselines in Table 1 on VESDE and VPSDE as linear diffusing mechanisms. Although minor performance degradation on the sample performance, Table 1 empirically demonstrates reduced variational gap. In all experimental settings, variational gaps significantly drop less than 0.1. On the other hand, variational gaps of the baseline diffusions on VPSDE for the FID model and the NLL model are 0.16 and 0.19, respectively. We conclude that PDM effectively decreases variational gap by training a parametrized inference path measure.

Table 2 compares experiments on CelebA to the current state-of-the-art baseline models in various disciplines: the diffusion model of UDM (Kim et al., 2021) and DDGM (Nachmani et al., 2021), the VAE model of CR-NVAE (Sinha and Dieng, 2021), the flow model of DenseFlow (Grcić et al., 2021), and the GAN model of Styleformer (Park and Kim, 2021). We observe a pair of implications. First, training a diffusing mechanism with PDM largely reduces variational gap. In particular, PDM reduces this inference gap from 1.57 to 0.09 on the NCSN++ backbone with VESDE and from 0.40 to 0.05 on the DDPM++ backbone with VPSDE. Also, we emphasize that NELBO of PDM largely outperforms NELBO of the baseline models. Second, our PDM reports the state-of-the-art performance with 2.23 FID score in the CelebA sample generation. This opposite trend on FID (compared to the results in CIFAR-10) indicates that a nonlinear diffusion is more effective on datasets of higher dimensions. In fact, this is a quite intuitive result because as the data dimension increases, the family of feasible diffusion expands exponentially, and the data-optimal diffusing mechanism is more likely to deviate from linear diffusions.

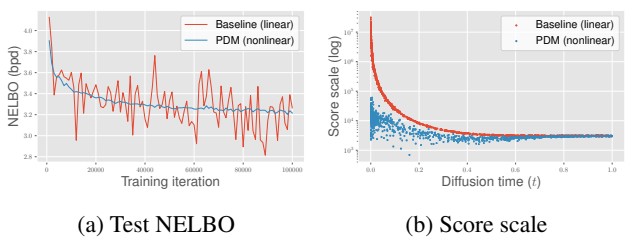

(a) Test NELBO (b) Score scale

Figure 5: Comparison of the linear and nonlinear diffusing mechanisms on a shallow network.

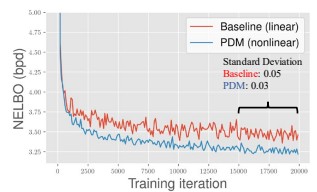

Figure 6: Comparison of the linear diffusion model and PDM on a deep network.

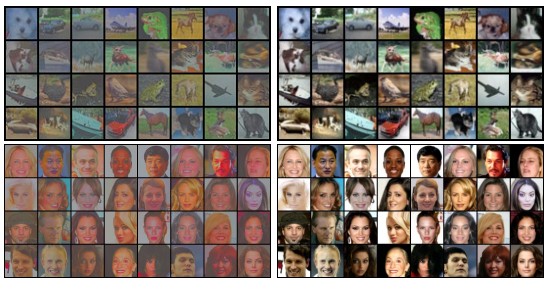

(a) Samples from $\mathbf{z}_0$ (b) Samples from $\mathbf{x}_0$

Figure 7: Non cherry-picked samples from latent/data.

Table 4: Ablation study on flow layers.

|  | L (# params) | NLL | NELBO | Gap |
|---|---|---|---|---|
| PDM | 4 (1.2M) | 3.19 | 3.30 | 0.11 |
| ResFlow |  | 5.72 | - | - |
| PDM | 8 (2.3M) | 3.19 | 3.24 | 0.05 |
| ResFlow |  | 4.48 | - | - |
| PDM | 32 (10.6M) | **3.04** | **3.09** | **0.05** |
| ResFlow |  | 3.72 | - | - |

## 6.2 QUALITATIVE RESULTS

**Test NELBO** To figure out an effect of a nonlinear diffusing mechanism on the training procedure, Figure 5 (a) shows test NELBO by training iterations on a diffusion model with 35M parameters. Without any modification on hyperparameters, we train PDM with a shallow ResFlow (Chen et al., 2019) of 3M parameters, and Figure 5 (a) shows that the training on PDM surprisingly stabilizes test NELBO. Figure 5 (b) illustrates that such training stability can be attributed to nearly identical score scales. As described in Figure 5 (b), the score scale on a data space is unbounded as the diffusion time goes to zero, and this extreme scale variation eventually leads to training instability on the baseline diffusion models. On the other hand, adopting flow transformation on a data space mitigates such extreme scale variations that destabilize training.

Figure 6 shows test NELBO on a diffusion model with 107.6M number of parameters. To stabilize the diffusion model, we apply the AdamW optimizer (Loshchilov and Hutter, 2017) with 0.01 weight decay, and Exponential Moving Average (Song et al., 2020) of 0.9999. Using the ResFlow model with 10.6M parameters, Figure 6 indicates that PDM consistently outperforms the baseline diffusion in terms of NELBO with smaller variations.

**Flow Depth** The flow model architecture is a building block that fundamentally determines the family of inference path measures. Table 4 empirically demonstrates that an inflexible variational family results in large variational gap on CIFAR-10. Additionally, the absolute magnitude of NLL performance keeps decreasing as the number of layers (L) increases.

**Sample Generation** Figure 7 presents samples from (a) a latent space and (b) a data space. Since a sample generation on the latent space is unnormalized, we normalize the sampled latent variables to be bounded on the range of $[0, 255]$, so we can visualize the latent information in image. Figure 7 indicates that a diffusion model and a flow model have distinctive and complementary roles in sample generation: 1) the diffusion model constructs the global context and 2) the flow model enriches this global context by colorizing it into a realistic image.

## 7 CONCLUSION

This paper expands the linear diffusing mechanism to be nonlinear by combining an invertible transformation and a diffusion model. This nonlinear diffusing mechanism learns the forward diffusion process out of variational family of inference path measures, and such optimization provides tighter variational gap compared to a baseline model with linear diffusion.

## 8    ETHICS STATEMENT

As our work improves sample quality, our methodology could be a source of fake datasets, i.e. fake images. This is a common threat from the line of deep generative modeling.

## 9    REPRODUCIBILITY STATEMENT

Every proofs and derivations are explained in details on Appendix A and B. Also, for reproducibility, we would release our code as soon as the discussion period opens through an anonymous repository to reviewers. We note that most of our code is built based on the released code of Song et al. (2020) in order to compare with baselines in a fair setting introduced in Song et al. (2020) and Song et al. (2021). Additionally, we provide the details on training/evaluation/neural architecture in Appendix C.

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

## A PROOFS OF THEOREM 1 AND THEOREM 2

**Theorem 1.** *Suppose that $p_{\phi,\theta}(\mathbf{x}_0)$ is the log-likelihood of the generative random variable $\mathbf{x}_0^{\phi,\theta}$. Then, the log-likelihood is bounded by*

$$\mathbb{E}_{p_r(\mathbf{x}_0)}\big[-\log p_{\phi,\theta}(\mathbf{x}_0)\big] \leq \mathcal{L}_{\phi,\theta}\big(\{\mathbf{x}_t\}_{t=0}^{T}; g^2\big),$$

*where*

$$\mathcal{L}_{\phi,\theta}\big(\{\mathbf{x}_t\}_{t=0}^{T}; g^2\big) = -\mathbb{E}_{p_r(\mathbf{x}_0)}\Big[\log\Big|\det\Big(\frac{\partial \mathbf{h}_\phi}{\partial \mathbf{x}_0}\Big)\Big|\Big] + \mathcal{L}_{\theta}\big(\{\mathbf{z}_t\}_{t=0}^{T}; g^2\big) - \mathbb{E}_{\mathbf{z}_T}\big[\log\pi(\mathbf{z}_T)\big]$$

$$+ \frac{d}{2}\int_0^T \beta(t) - \frac{g^2(t)}{\sigma^2(t)}\,\mathrm{d}t,$$

*with $\sigma^2(t) = 1 - e^{-\int_0^t \beta(s)\,\mathrm{d}s} + \int_0^t[g^2(s) - \beta(s)]\,\mathrm{d}s$.*

*Proof of Theorem 1.* From the change of variable, the transformation of $\mathbf{z}_0 = \mathbf{h}_\phi(\mathbf{x}_0)$ induces

$$p_r(\mathbf{x}_0) = \frac{p_0(\mathbf{z}_0)}{\Big|\det\Big(\frac{\partial \mathbf{h}_\phi}{\partial \mathbf{x}_0}\Big)\Big|^{-1}},$$

and thus the entropy of the data distribution becomes

$$\mathcal{H}(p_r) = -\int p_r(\mathbf{x}_0)\log p_r(\mathbf{x}_0)\,\mathrm{d}\mathbf{x}_0$$

$$= -\int p_0(\mathbf{z}_0)\log\frac{p_0(\mathbf{z}_0)}{\Big|\det\Big(\frac{\partial \mathbf{h}_\phi}{\partial \mathbf{x}_0}\Big)\Big|^{-1}}\,\mathrm{d}\mathbf{z}_0$$

$$= -\int p_r(\mathbf{x}_0)\log\Big|\det\Big(\frac{\partial \mathbf{h}_\phi}{\partial \mathbf{x}_0}\Big)\Big|\,\mathrm{d}\mathbf{x}_0 - \int p_0(\mathbf{z}_0)\log p_0(\mathbf{z}_0)\,\mathrm{d}\mathbf{z}_0$$

$$= -\mathbb{E}_{p_r(\mathbf{x}_0)}\Big[\log\Big|\det\Big(\frac{\partial \mathbf{h}_\phi}{\partial \mathbf{x}_0}\Big)\Big|\Big] - \int p_0(\mathbf{z}_0)\log p_0(\mathbf{z}_0)\,\mathrm{d}\mathbf{z}_0$$

$$= -\mathbb{E}_{p_r(\mathbf{x}_0)}\Big[\log\Big|\det\Big(\frac{\partial \mathbf{h}_\phi}{\partial \mathbf{x}_0}\Big)\Big|\Big] + \mathcal{H}(p_0).$$

From Theorem 4 of Song et al. (2021), the entropy at $t = 0$ equals to

$$\mathcal{H}(p_0) = \mathcal{H}(p_T) - \frac{1}{2}\int_0^T \mathbb{E}_{p_t(\mathbf{z}_t)}\big[2\nabla_{\mathbf{z}_t}\cdot\mathbf{f}(\mathbf{z}_t, t) + g^2(t)\|\nabla_{\mathbf{z}_t}\log p_t(\mathbf{z}_t)\|_2^2\big]\,\mathrm{d}t,$$

where $\mathbf{f}(\mathbf{z}_t, t)$ is a drift term of the diffusion for $\mathbf{z}_t$ and $p_t$ is the probability distribution of $\mathbf{z}_t$. Therefore, the negative log-likelihood becomes

$$-\mathbb{E}_{p_r(\mathbf{x}_0)}\big[\log p_{\phi,\theta}(\mathbf{x}_0)\big] = D_{KL}(p_r\|p_{\phi,\theta}) + \mathcal{H}(p_r)$$

$$\leq D_{KL}(\boldsymbol{\mu}_\phi^d\|\boldsymbol{\nu}_{\phi,\theta}^d) + \mathcal{H}(p_r)$$

$$= D_{KL}(\boldsymbol{\mu}_\phi^d\|\boldsymbol{\nu}_{\phi,\theta}^d) - \mathbb{E}_{p_r(\mathbf{x}_0)}\Big[\log\Big|\det\Big(\frac{\partial \mathbf{h}_\phi}{\partial \mathbf{x}_0}\Big)\Big|\Big] + \mathcal{H}(p_0)$$

$$= D_{KL}(\boldsymbol{\mu}_\phi^l\|\boldsymbol{\nu}_\theta^l) - \mathbb{E}_{p_r(\mathbf{x}_0)}\Big[\log\Big|\det\Big(\frac{\partial \mathbf{h}_\phi}{\partial \mathbf{x}_0}\Big)\Big|\Big] + \mathcal{H}(p_T)$$

$$- \frac{1}{2}\int_0^T \mathbb{E}_{\mathbf{z}_t^\phi}\big[-d\beta(t) + g^2(t)\|\nabla_{\mathbf{z}_t}\log p_t(\mathbf{z}_t)\|_2^2\big]\,\mathrm{d}t.$$

Now, from Theorem 1 of Song et al. (2021), the KL-divergence between the path measures becomes

$$D_{KL}(\boldsymbol{\mu}_\phi^l\|\boldsymbol{\nu}_\theta^l) = D_{KL}(p_T\|\pi) + \frac{1}{2}\int_0^T g^2(t)\mathbb{E}_{p_t(\mathbf{z}_t)}\big[\|\mathbf{s}_\theta(\mathbf{z}_t, t) - \nabla_{\mathbf{z}_t}\log p_t(\mathbf{z}_t)\|_2^2\big]\,\mathrm{d}t,$$

so if we plug in this into the negative log-likelihood, we yield the following:

$$-\mathbb{E}_{p_r(\mathbf{x}_0)}\big[\log p_{\phi,\theta}(\mathbf{x}_0)\big]$$

$$= -\mathbb{E}_{p_r(\mathbf{x}_0)}\Big[\log\Big|\det\Big(\frac{\partial\mathbf{h}_\phi}{\partial\mathbf{x}_0}\Big)\Big|\Big] + \frac{1}{2}\int_0^T g^2(t)\mathbb{E}_{\mathbf{z}_t}\big[\|\mathbf{s}_\theta(\mathbf{z}_t,t) - \nabla_{\mathbf{z}_t}\log p_t(\mathbf{z}_t)\|_2^2\big]\,\mathrm{d}t$$

$$+ D_{KL}(p_T\|\pi) + \mathcal{H}(p_T) - \frac{1}{2}\int_0^T \mathbb{E}_{\mathbf{z}_t}\big[-d\beta(t) + g^2(t)\|\nabla_{\mathbf{z}_t}\log p_t(\mathbf{z}_t)\|_2^2\big]\,\mathrm{d}t$$

$$= -\mathbb{E}_{p_r(\mathbf{x}_0)}\Big[\log\Big|\det\Big(\frac{\partial\mathbf{h}_\phi}{\partial\mathbf{x}_0}\Big)\Big|\Big] - \mathbb{E}_{\mathbf{z}_T}\big[\log\pi(\mathbf{z}_T)\big] + \frac{d}{2}\int_0^T \beta(t)\,\mathrm{d}t$$

$$+ \frac{1}{2}\int_0^T g^2(t)\mathbb{E}_{\mathbf{z}_t}\big[\|\mathbf{s}_\theta(\mathbf{z}_t,t) - \nabla_{\mathbf{z}_t}\log p_t(\mathbf{z}_t)\|_2^2 - \|\nabla_{\mathbf{z}_t}\log p_t(\mathbf{z}_t)\|_2^2\big]\,\mathrm{d}t$$

Also, we have

$$\mathbb{E}_{\mathbf{z}_t}\big[\mathbf{s}_\theta(\mathbf{z}_t,t)\cdot\nabla_{\mathbf{z}_t}\log p_t(\mathbf{z}_t)\big] = \int p_t(\mathbf{z}_t)\mathbf{s}_\theta(\mathbf{z}_t,t)\cdot\nabla_{\mathbf{z}_t}\log p_t(\mathbf{z}_t)\,\mathrm{d}\mathbf{z}_t$$

$$= \int \mathbf{s}_\theta(\mathbf{z}_t,t)\cdot\nabla_{\mathbf{z}_t}p_t(\mathbf{z}_t)\,\mathrm{d}\mathbf{z}_t$$

$$= \int \mathbf{s}_\theta(\mathbf{z}_t,t)\cdot\int p_0(\mathbf{z}_0)\nabla_{\mathbf{z}_t}p_{0t}(\mathbf{z}_t|\mathbf{z}_0)\,\mathrm{d}\mathbf{z}_0\,\mathrm{d}\mathbf{z}_t$$

$$= \mathbb{E}_{\mathbf{z}_0}\mathbb{E}_{\mathbf{z}_t|\mathbf{z}_0}\big[\mathbf{s}_\theta(\mathbf{z}_t,t)\cdot\nabla_{\mathbf{z}_t}\log p_{0t}(\mathbf{z}_t|\mathbf{z}_0)\big]$$

Therefore,

$$\frac{1}{2}\int_0^T g^2(t)\mathbb{E}_{\mathbf{z}_t}\big[\|\mathbf{s}_\theta(\mathbf{z}_t,t) - \nabla_{\mathbf{z}_t}\log p_t(\mathbf{z}_t)\|_2^2 - \|\nabla_{\mathbf{z}_t}\log p_t(\mathbf{z}_t)\|_2^2\big]$$

$$= \int_0^T g^2(t)\mathbb{E}_{\mathbf{z}_t}\Big[\frac{1}{2}\|\mathbf{s}_\theta(\mathbf{z}_t,t)\|_2^2 - \mathbf{s}_\theta(\mathbf{z}_t,t)\cdot\nabla_{\mathbf{z}_t}\log p_t(\mathbf{z}_t)\Big]$$

$$= \int_0^T g^2(t)\mathbb{E}_{\mathbf{z}_0}\mathbb{E}_{\mathbf{z}_t|\mathbf{z}_0}\Big[\frac{1}{2}\|\mathbf{s}_\theta(\mathbf{z}_t,t)\|_2^2 - \mathbf{s}_\theta(\mathbf{z}_t,t)\cdot\nabla_{\mathbf{z}_t}\log p_{0t}(\mathbf{z}_t|\mathbf{z}_0)\Big]$$

$$= \frac{1}{2}\int_0^T g^2(t)\mathbb{E}_{\mathbf{z}_0}\mathbb{E}_{\mathbf{z}_t|\mathbf{z}_0}\Big[\|\mathbf{s}_\theta(\mathbf{z}_t,t) - \nabla_{\mathbf{z}_t}\log p_{0t}(\mathbf{z}_t|\mathbf{z}_0)\|_2^2 - \|\nabla_{\mathbf{z}_t}\log p_{0t}(\mathbf{z}_t|\mathbf{z}_0)\|_2^2\Big].$$

Now, since $p_{0t}(\mathbf{z}_t|\mathbf{z}_0) = \mathcal{N}(\mathbf{z}_t;\mu(t)\mathbf{z}_t,\sigma^2(t)\mathbf{I})$ for $\mu(t)$ and $\sigma^2(t)$ determined by $\beta(t)$ and $g(t)$, we have

$$\mathbb{E}_{\mathbf{z}_t|\mathbf{z}_0}\big[\|\nabla_{\mathbf{z}_t}\log p_{0t}(\mathbf{z}_t|\mathbf{z}_0)\big] = \mathbb{E}_{\mathbf{z}_t|\mathbf{z}_0}\Big[\Big\|\frac{\mathbf{z}_t - \mu(t)\mathbf{z}_0}{\sigma^2(t)}\Big\|_2^2\Big] = \mathbb{E}_{\mathcal{N}(\mathbf{z};0,\mathbf{I})}\Big[\frac{\|\mathbf{z}\|_2^2}{\sigma^2(t)}\Big] = \frac{d}{\sigma^2(t)},$$

and we have the desired result. $\qquad\square$

**Theorem 2.** *Suppose the flow network is flexible enough to transform $p_r$ to arbitrary continuous distribution. For any $\theta$, if $\nabla.\mathbf{s}_\theta(\cdot,t)$ is symmetric for any $t \in [0,T]$ and $\nabla.\log\pi(\cdot) \equiv \mathbf{s}_\theta(\cdot,T)$, then there exists $\phi^*$ such that $\mathrm{Gap}_d(\boldsymbol{\mu}_{\phi^*}^d,\boldsymbol{\nu}_{\phi^*,\theta}^d) = 0$. Furthermore, we have*

$$\phi^* = \arg\min_\phi \mathrm{Gap}_d(\boldsymbol{\mu}_\phi^d,\boldsymbol{\nu}_{\phi,\theta}^d) = \arg\min_\phi D_{KL}(\boldsymbol{\mu}_\phi^d\|\boldsymbol{\nu}_{\phi,\theta}^d) = \arg\min_\phi D_{KL}(p_r\|p_{\phi,\theta}).$$

*Proof of Theorem 2.* The symmetry of $\nabla.\mathbf{s}_\theta(\cdot,t)$ indicates that $\mathbf{s}_\theta(\cdot,t)$ is a closed 1-form. Therefore, from the Poincare's lemma (Do Carmo, 2012), $\mathbf{s}_\theta(\cdot,t)$ is exact, and there exists a scalar field $f_t^\theta$ that satisfies $\mathbf{s}_\theta(\cdot,t) = \nabla.f_t^\theta(\cdot)$ for any $t$. If we define by $q_t^\theta(\mathbf{z}_t) = \frac{\exp(f_t^\theta(\mathbf{z}_t))}{\int\exp(f_t^\theta(\mathbf{z}))\,\mathrm{d}\mathbf{z}}$, then we have $\mathbf{s}_\theta(\mathbf{z}_t,t) = \nabla_{\mathbf{z}_t}\log q_t^\theta(\mathbf{z}_t,t)$ for any $t$. With this construction, the reverse SDE on the transformed latent space becomes

$$\mathrm{d}\mathbf{z}_t^\theta = \Big[-\frac{1}{2}\beta(t)\mathbf{z}_t^\theta - g^2(t)\nabla_{\mathbf{z}_t^\theta}\log q_t^\theta(\mathbf{z}_t^\theta)\Big]\,\mathrm{d}t + g(t)\,\mathrm{d}\bar{\boldsymbol{\omega}}_t, \quad \mathbf{z}_T^\theta \sim \pi = q_T^\theta.$$

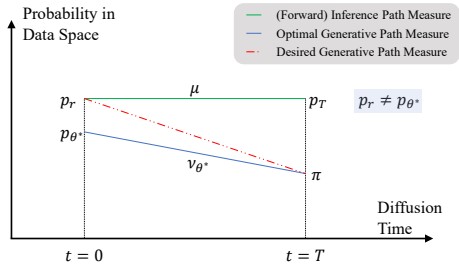
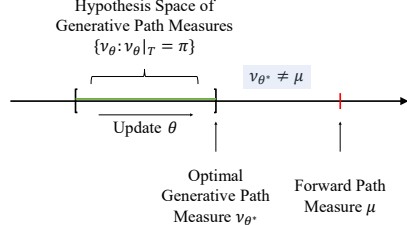

(a) Optimal Generative Distribution Not Equal to Data Distribution

(b) Forward Path Measure Not Included in Hypothesis Class of Generative Process

Figure 8: (a) For the vanilla linear diffusion model in the data space, $p_T$ and $\pi$ naturally diverges. If the score network estimates the exact data score (i.e., $\mathbf{s}_{\boldsymbol{\theta}}(\mathbf{x}_t, t) = \nabla_{\mathbf{x}_t} \log p_t(\mathbf{x}_t)$), then the (optimal) generative distribution becomes $p_{\boldsymbol{\theta}}(\mathbf{x}_0) = \int p_T(\mathbf{x}_T) p_{T0}(\mathbf{x}_0|\mathbf{x}_T)\, \mathrm{d}\mathbf{x}_T$. On the other hand, the real distribution is $p_r(\mathbf{x}_0) = \int \pi(\mathbf{x}_T) p_{T0}(\mathbf{x}_0|\mathbf{x}_T)\, \mathrm{d}\mathbf{x}_T$, with identical (reverse) transition probability, $p_{T0}(\mathbf{x}_0|\mathbf{x}_T)$ to the generative process. Mismatch of $p_T$ and $\pi$ yields that the optimality of the diffusion model does not guarantee to estimate the real distribution: $p_{\boldsymbol{\theta}}(\mathbf{x}_0) \neq \pi(\mathbf{x}_0)$. (b) Since $p_T$ and $\pi$ are distinctive distribution, the forward path measure $\boldsymbol{\mu}$ is not included in the hypothesis class of generative path measures, $\{\boldsymbol{\nu}_{\boldsymbol{\theta}} :$ the marginal distribution of $\boldsymbol{\nu}_{\boldsymbol{\theta}}$ at $t = T$ is $\pi\}$. In addition, $\boldsymbol{\mu}$ is not the element of the closure of the hypothesis class, and we conclude that $D_{KL}(p_r \| p_{\boldsymbol{\theta}}) > 0$.

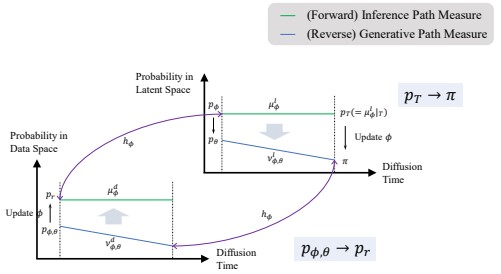
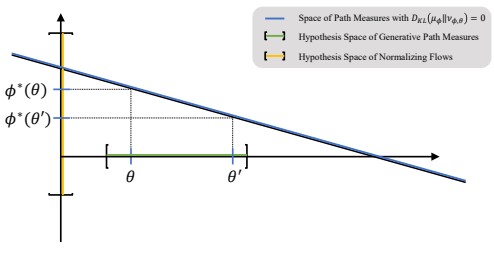

(a) Nonlinear Diffusion in Data and Latent Spaces

(b) Illustration of Theorem 2: Existence of $\boldsymbol{\phi}^*$ for any $\boldsymbol{\theta}$ such that $D_{KL}(p_r \| p_{\boldsymbol{\phi}^*, \boldsymbol{\theta}}) = 0$

Figure 9: (a) While $p_T$ and $\pi$ is different in PDM as well, the marginal distribution of $\boldsymbol{\mu}_{\boldsymbol{\phi}}$ at $t = 0$ depends on the transformation, $\mathbf{h}_{\boldsymbol{\phi}}$, modeled with normalizing flows. This $\phi$-dependent initial random variable at $\mathbf{z}_0$ influences its fully diffused variable of $\mathbf{z}_T$, and optimizing $\phi$ minimizes the discrepancy between $p_T$ and $\pi$, where $\mathbf{z}_T \sim p_T$. Theorem 2 guarantees that this discrepancy is completely eradicated with the optimum point of $\boldsymbol{\phi}^*$. (b) Theorem 2 guarantees that for any $\boldsymbol{\theta}$, there exists $\boldsymbol{\phi}^*$ such that $D_{KL}(p_r \| p_{\boldsymbol{\phi}^*, \boldsymbol{\theta}}) = 0$, which implies that $p_r(\mathbf{x}_0) = p_{\boldsymbol{\phi}^*, \boldsymbol{\theta}}(\mathbf{x}_0)$.

If $\boldsymbol{\rho}_{\boldsymbol{\theta}}$ is the path measure of the solution of the above SDE, then the marginal distribution of $\boldsymbol{\rho}_{\boldsymbol{\theta}}$ at $t$ would be $q_t^{\boldsymbol{\theta}}$. Therefore, if $\mathbf{h}_{\boldsymbol{\phi}^*}^{-1}$ maps $p_r$ to $q_0^{\boldsymbol{\theta}}$, then the forward process on the latent space $\{\mathbf{z}_t^{\boldsymbol{\phi}^*}\}_{t=0}^T$ becomes the generative process on the latent space $\{\mathbf{z}_t^{\boldsymbol{\theta}}\}_{t=0}^T$. Thus, we have $\boldsymbol{\mu}_{\boldsymbol{\phi}^*}^l = \boldsymbol{\nu}_{\boldsymbol{\theta}}^l$ in distribution, which leads $D_{KL}(\boldsymbol{\mu}_{\boldsymbol{\phi}^*}^l \| \boldsymbol{\nu}_{\boldsymbol{\theta}}^l) = 0$. We get the desired result by the change of variable formula: $D_{KL}(\boldsymbol{\mu}_{\boldsymbol{\phi}^*}^d \| \boldsymbol{\nu}_{\boldsymbol{\phi}^*, \boldsymbol{\theta}}^d) = D_{KL}(\boldsymbol{\mu}_{\boldsymbol{\phi}^*}^l \| \boldsymbol{\nu}_{\boldsymbol{\theta}}^l) = 0$. $\qquad\square$

## B DERIVATIONS

### B.1 DERIVATION OF NONLINEAR DIFFUSION TERMS

Throughout this section, we omit $\phi$ for the notational simplicity. Beginning with the linear diffusion on the latent space given by

$$d\mathbf{z}_t = -\frac{1}{2}\beta(t)\mathbf{z}_t \, dt + g(t) \, d\boldsymbol{\omega}_t, \quad \mathbf{z}_0 = \mathbf{h}(\mathbf{x}_0) \text{ with } \mathbf{x}_0 \sim p_r, \tag{11}$$

we have defined $\mathbf{x}_t = \mathbf{h}^{-1}(\mathbf{z}_t)$ as the induced random variables. Then by the multivariate Ito's Lemma (Oksendal, 2013), the $k$-th component of the induced variable satisfies

$$d\mathbf{x}_{t,k} = \frac{\partial \mathbf{h}_k^{-1}}{\partial t} \, dt + \left[\nabla_{\mathbf{z}_t} \mathbf{h}_k^{-1}(\mathbf{z}_t)\right]^T d\mathbf{z}_t + \frac{1}{2}\text{tr}\left(\nabla_{\mathbf{z}_t}^2 \mathbf{h}_k^{-1}(\mathbf{z}_t) \, d\mathbf{z}_t \, d\mathbf{z}_t^T\right) \tag{12}$$

Plugging the linear diffusion SDE 11 in the SDE of Eq. 12, Eq. 12 is derived by

$$d\mathbf{x}_{t,k} = \left[\nabla_{\mathbf{z}_t} \mathbf{h}_k^{-1}(\mathbf{z}_t)\right]^T \left\{-\frac{1}{2}\beta(t)\mathbf{z}_t \, dt + g(t) \, d\boldsymbol{\omega}_t\right\} + \frac{1}{2}g^2(t)\text{tr}\left(\nabla_{\mathbf{z}_t}^2 \mathbf{h}_k^{-1}(\mathbf{z}_t)\right) dt$$

$$= \left\{-\frac{1}{2}\beta(t)\left[\nabla_{\mathbf{z}_t} \mathbf{h}_k^{-1}(\mathbf{z}_t)\right]^T \mathbf{z}_t + \frac{1}{2}g^2(t)\text{tr}\left(\nabla_{\mathbf{z}_t}^2 \mathbf{h}_k^{-1}(\mathbf{z}_t)\right)\right\} dt + g(t)\left[\nabla_{\mathbf{z}_t} \mathbf{h}_k^{-1}(\mathbf{z}_t)\right]^T d\boldsymbol{\omega}_t, \tag{13}$$

because $\frac{\partial \mathbf{h}_k^{-1}}{\partial t} = 0$. Then, Eq. 13 in vector form becomes

$$d\mathbf{x}_t = \mathbf{f}(\mathbf{x}_t, t) \, dt + \mathbf{G}(\mathbf{x}_t, t) \, d\boldsymbol{\omega}_t,$$

where vector-valued drift and diffusion terms are given by

$$\begin{cases} \mathbf{f}(\mathbf{x}_t, t) = -\frac{1}{2}\beta(t)\nabla_{\mathbf{z}_t}\mathbf{h}^{-1}(\mathbf{z}_t)\mathbf{z}_t + \frac{1}{2}g^2(t)\text{tr}\left(\nabla_{\mathbf{z}_t}^2\mathbf{h}^{-1}(\mathbf{z}_t)\right) \\ \mathbf{G}(\mathbf{x}_t, t) = g(t)\nabla_{\mathbf{z}_t}\mathbf{h}^{-1}(\mathbf{z}_t), \end{cases} \tag{14}$$

where $\nabla_{\mathbf{z}_t}^2\mathbf{h}^{-1}(\mathbf{z}_t)$ is a 3-dimensional tensor with $(i, j, k)$-th element to be $\left(\nabla_{\mathbf{z}_t}^2\mathbf{h}_k^{-1}(\mathbf{z}_t)\right)_{i,j}$, and the trace operator applied on this tensor results in the vector of $\left[\text{tr}\left(\nabla_{\mathbf{z}_t}^2\mathbf{h}_1^{-1}(\mathbf{z}_t)\right), ..., \text{tr}\left(\nabla_{\mathbf{z}_t}^2\mathbf{h}_d^{-1}(\mathbf{z}_t)\right)\right]^T$. From the inverse function theorem (Rudin et al., 1964), the Jacobian of the inverse function $\nabla_{\mathbf{z}_t}\mathbf{h}^{-1}(\mathbf{z}_t)$ equals to the inverse Jacobian $\left[\nabla_{\mathbf{x}_t}\mathbf{h}(\mathbf{x}_t)\right]^{-1}$. Therefore, Eq. 14 is transformed to

$$\begin{cases} \mathbf{f}(\mathbf{x}_t, t) = -\frac{1}{2}\beta(t)\left[\nabla_{\mathbf{x}_t}\mathbf{h}(\mathbf{x}_t)\right]^{-1}\mathbf{h}(\mathbf{x}_t) + \frac{1}{2}g^2(t)\text{tr}\left(\nabla_{\mathbf{z}_t}^2\mathbf{h}^{-1}(\mathbf{z}_t)\right) \\ \mathbf{G}(\mathbf{x}_t, t) = g(t)\left[\nabla_{\mathbf{x}_t}\mathbf{h}(\mathbf{x}_t)\right]^{-1}. \end{cases} \tag{15}$$

Now, we derive the second term of $\mathbf{f}$ in terms of $\mathbf{x}_t$ as follows: observe that $\sum_k \frac{\partial \mathbf{h}_i^{-1}}{\partial \mathbf{z}_{t,k}} \frac{\partial \mathbf{h}_k}{\partial \mathbf{x}_{t,j}} = \delta_{i,j}$, where $\delta_{i,j} = 1$ if $i = j$ and 0 otherwise. Differentiating both sides with respect to $\mathbf{x}_{t,l}$, we have

$$\sum_k \left\{\frac{\partial}{\partial \mathbf{x}_{t,l}}\left(\frac{\partial \mathbf{h}_i^{-1}}{\partial \mathbf{z}_{t,k}}\right)\right\}\frac{\partial \mathbf{h}_k}{\partial \mathbf{x}_{t,j}} + \frac{\partial \mathbf{h}_i^{-1}}{\partial \mathbf{z}_{t,k}}\left\{\frac{\partial}{\partial \mathbf{x}_{t,l}}\left(\frac{\partial \mathbf{h}_k}{\partial \mathbf{x}_{t,j}}\right)\right\} = 0,$$

where the first term is

$$\sum_{k,m} \frac{\partial \mathbf{h}_m}{\partial \mathbf{x}_{t,l}}\left\{\frac{\partial}{\partial \mathbf{z}_{t,m}}\left(\frac{\partial \mathbf{h}_i^{-1}}{\partial \mathbf{z}_{t,k}}\right)\right\}\frac{\partial \mathbf{h}_k}{\partial \mathbf{x}_{t,j}} = \sum_{k,m} \left(\nabla_{\mathbf{x}_t}\mathbf{h}(\mathbf{x}_t)\right)_{l,m}^T\left(\nabla_{\mathbf{z}_t}^2\mathbf{h}_i^{-1}(\mathbf{z}_t)\right)_{m,k}\left(\nabla_{\mathbf{x}_t}\mathbf{h}(\mathbf{x}_t)\right)_{k,j},$$

using the chain rule, and the second term becomes

$$\sum_k \frac{\partial \mathbf{h}_i^{-1}}{\partial \mathbf{z}_{t,k}}\left\{\frac{\partial}{\partial \mathbf{x}_{t,l}}\left(\frac{\partial \mathbf{h}_k}{\partial \mathbf{x}_{t,j}}\right)\right\} = \sum_k \left(\nabla_{\mathbf{z}_t}\mathbf{h}^{-1}(\mathbf{z}_t)\right)_{i,k}\left(\nabla_{\mathbf{x}_t}^2\mathbf{h}_k(\mathbf{x}_t)\right)_{l,j}.$$

From above, we derive the trace term of $\mathbf{f}$ in Eq. 15 as

$$\text{tr}\left(\nabla_{\mathbf{z}_t}^2\mathbf{h}^{-1}(\mathbf{z}_t)\right) = -\text{tr}\left(\left[\nabla_{\mathbf{x}_t}\mathbf{h}(\mathbf{x}_t)\right]^{-T}\left(\left[\nabla_{\mathbf{x}_t}\mathbf{h}(\mathbf{x}_t)\right]^{-1} * \nabla_{\mathbf{x}_t}^2\mathbf{h}(\mathbf{x}_t)\right)\left[\nabla_{\mathbf{x}_t}\mathbf{h}(\mathbf{x}_t)\right]^{-1}\right),$$

Table 5: Comparison of variational gap between the VAE, the diffusion model, and PDM.

| Models | Diffusion | Stochastic Process | Variational Gap (=NLL$-$NELBO) |
|---|---|---|---|
| VAE | No | $\{\mathbf{x}, \mathbf{z}\}$ | $\mathbb{E}_{\mathbf{x}}\big[D_{KL}(q_{\phi}(\mathbf{z}|\mathbf{x})\|p_{\theta}(\mathbf{z}|\mathbf{x}))\big]$ |
| Diffusion Models | Data space | $\{\mathbf{x}_t\}_{t=0}^{T}$ | $\mathbb{E}_{\mathbf{x}_0}\big[D_{KL}(\boldsymbol{\mu}(\{\mathbf{x}_t\}|\mathbf{x}_0)\|\boldsymbol{\nu}_{\theta}(\{\mathbf{x}_t\}|\mathbf{x}_0))\big]$ |
| PDM | Latent space | $\{\mathbf{x}_t\}_{t=0}^{T}/\{\mathbf{z}_t\}_{t=0}^{T}$ | $\mathbb{E}_{\mathbf{z}_0}\big[D_{KL}(\boldsymbol{\mu}_{\phi}^{l}(\{\mathbf{z}_t\}|\mathbf{z}_0)\|\boldsymbol{\nu}_{\theta}^{l}(\{\mathbf{z}_t\}|\mathbf{z}_0))\big]$ |

where $\nabla_{\mathbf{x}_t}^2 \mathbf{h}(\mathbf{x}_t)$ is a 3-dimensional tensor with $(i, j, k)$-th element to be $\big(\nabla_{\mathbf{x}_t}^2 \mathbf{h}_k(\mathbf{x}_t)\big)_{i,j}$. Also, we define $*$ operation as the element-wise matrix multiplication given by

$$\left(\big[\nabla_{\mathbf{x}_t}\mathbf{h}(\mathbf{x}_t)\big]^{-1} * \nabla_{\mathbf{x}_t}^2 \mathbf{h}(\mathbf{x}_t)\right)_{i,j} := \nabla_{\mathbf{x}_t}\big[\mathbf{h}(\mathbf{x}_t)\big]^{-1}\left(\nabla_{\mathbf{x}_t}^2 \mathbf{h}(\mathbf{x}_t)\right)_{i,j}.$$

Combining all together, thus, we derive nonlinear drift term in Eq. 15 as a function of $\mathbf{x}_t$ given by

$$\mathbf{f}(\mathbf{x}_t, t) = -\frac{1}{2}\beta(t)\big[\nabla_{\mathbf{x}_t}\mathbf{h}(\mathbf{x}_t)\big]^{-1}\mathbf{h}(\mathbf{x}_t)$$
$$-\frac{1}{2}g^2(t)\mathrm{tr}\left(\big[\nabla_{\mathbf{x}_t}\mathbf{h}(\mathbf{x}_t)\big]^{-T}\left(\big[\nabla_{\mathbf{x}_t}\mathbf{h}(\mathbf{x}_t)\big]^{-1} * \nabla_{\mathbf{x}_t}^2\mathbf{h}(\mathbf{x}_t)\right)\big[\nabla_{\mathbf{x}_t}\mathbf{h}(\mathbf{x}_t)\big]^{-1}\right).$$

### B.2 Derivation of Variational Gap

First, the KL divergence between two path measures satisfies

$$D_{KL}(\boldsymbol{\mu}\|\boldsymbol{\nu}_{\theta}) = \int \boldsymbol{\mu}(\{\mathbf{x}_t\}) \log \frac{\boldsymbol{\mu}(\{\mathbf{x}_t\})}{\boldsymbol{\nu}_{\theta}(\{\mathbf{x}_t\})}\, \mathrm{d}\mathbf{x}_{0:T}$$
$$= \int p_r(\mathbf{x}_0)\boldsymbol{\mu}(\{\mathbf{x}_t\}|\mathbf{x}_0) \log \frac{p_r(\mathbf{x}_0)\boldsymbol{\mu}(\{\mathbf{x}_t\}|\mathbf{x}_0)}{p_{\theta}(\mathbf{x}_0)\boldsymbol{\nu}_{\theta}(\{\mathbf{x}_t\}|\mathbf{x}_0)}\, \mathrm{d}\mathbf{x}_{0:T}$$
$$= \int p_r(\mathbf{x}_0)\boldsymbol{\mu}(\{\mathbf{x}_t\}|\mathbf{x}_0)\left[\log \frac{p_r(\mathbf{x}_0)}{p_{\theta}(\mathbf{x}_0)} + \log \frac{\boldsymbol{\mu}(\{\mathbf{x}_t\}|\mathbf{x}_0)}{\boldsymbol{\nu}_{\theta}(\{\mathbf{x}_t\}|\mathbf{x}_0)}\right] \mathrm{d}\mathbf{x}_{0:T}$$
$$= \int p_r(\mathbf{x}_0) \log \frac{p_r(\mathbf{x}_0)}{p_{\theta}(\mathbf{x}_0)}\, \mathrm{d}\mathbf{x}_0 + \int p_r(\mathbf{x}_0)\boldsymbol{\mu}(\{\mathbf{x}_t\}|\mathbf{x}_0) \log \frac{\boldsymbol{\mu}(\{\mathbf{x}_t\}|\mathbf{x}_0)}{\boldsymbol{\nu}_{\theta}(\{\mathbf{x}_t\}|\mathbf{x}_0)}\, \mathrm{d}\mathbf{x}_{0:T}$$
$$= D_{KL}(p_r\|p_{\theta}) + \mathbb{E}_{p_r(\mathbf{x}_0)}\big[D_{KL}(\boldsymbol{\mu}(\{\mathbf{x}_t\}|\mathbf{x}_0)\|\boldsymbol{\nu}_{\theta}(\{\mathbf{x}_t\}|\mathbf{x}_0))\big],$$

where $\mathbf{x}_{0:T} = \{\mathbf{x}_t\}_{t=0}^{T}$. Therefore, variational gap is derived by

$$\mathrm{Gap}(\boldsymbol{\mu}, \boldsymbol{\nu}_{\theta}) = D_{KL}(\boldsymbol{\mu}\|\boldsymbol{\nu}_{\theta}) - D_{KL}(p_r\|p_{\theta})$$
$$= \mathbb{E}_{p_r(\mathbf{x}_0)}\big[D_{KL}(\boldsymbol{\mu}(\{\mathbf{x}_t\}|\mathbf{x}_0)\|\boldsymbol{\nu}_{\theta}(\{\mathbf{x}_t\}|\mathbf{x}_0))\big].$$

The gap $\mathrm{Gap}_d(\boldsymbol{\mu}_{\phi}^d, \boldsymbol{\nu}_{\phi,\theta}^d)$ in Eq. 10 is derived to be $\mathbb{E}_{\mathbf{z}_0}\big[D_{KL}(\boldsymbol{\mu}_{\phi}(\{\mathbf{z}_t\}|\mathbf{z}_0)\|\boldsymbol{\nu}_{\theta}(\{\mathbf{z}_t\}|\mathbf{z}_0))\big]$, analogously.

Table 6: List of path measures.

| Space (Diffusion) | Measure | Type |
|---|---|---|
| Data (Nonlinear) | $\boldsymbol{\mu}_{\phi}^d$ | Inference |
| | $\boldsymbol{\nu}_{\phi,\theta}^d$ | Generative |
| Latent (Linear) | $\boldsymbol{\mu}_{\phi}^l$ | Inference |
| | $\boldsymbol{\nu}_{\theta}^l$ | Generative |

## C Implementation Details

### C.1 Model Architecture

**Diffusion Model** We implement two diffusion models as backbone: NCSN++ (VE) (Song et al., 2020) and DDPM++ (VP) (Song et al., 2020), where two backbones are one of the best performers in CIFAR-10 dataset. In our setting, NCSN++ assumes the score network with parametrization of $\mathbf{s}_{\theta}(\mathbf{z}_t, \log \sigma^2(t))$, where $\sigma^2(t) = \sigma_{min}^2(\frac{\sigma_{max}}{\sigma_{min}})^{2t}$ is the variance of the transition probability $p_{0t}(\mathbf{z}_t|\mathbf{z}_0)$ with VESDE. As introduced in Song et al. (2020), we use the Gaussian Fourier embeddings (Tancik et al., 2020) to model the high frequency details across the temporal embedding. DDPM++ models the score network with parametrization of $\epsilon_{\theta}(\mathbf{z}_t, t)$, which

targets to estimate $-\sigma(t)\nabla_{\mathbf{z}_t} \log p_t(\mathbf{z}_t)$. We use the Transformer sinusoidal temporal embedding (Vaswani et al., 2017).

We use the U-Net (Ronneberger et al., 2015) architecture for the score networks on both NCSN++ and DDPM++ based on (Ho et al., 2020). We stack U-Net resblocks of up-and-down convolutions with skip connections that connect the identical-dimensional layers. Also, we follow Ho et al. (2020) by applying the global attention at the resolution of $16 \times 16$. We use four U-Net resblocks with four feature map resolutions ($32 \times 32$ to $4 \times 4$). There are eight and four resblocks at each resolution on CIFAR-10 and CelebA, respectively.

**Flow Model** We implement two normalizing flow models as backbone: the ResFlow (Chen et al., 2019) and the decoupled ResFlow (Ma et al., 2020). Although Ma et al. (2020) makes use of the Glow (Kingma and Dhariwal, 2018) for their flow model, we replace this glow architecture with the ResFlow, mainly because the ResFlow transforms the variables from the identity function. We name the experiments on this decoupled ResFlow as PDM (FID, dec) and PDM (NLL, dec). Otherwise, we train the model with ResFlow as default. For the ResFlow, we drop three components from the original paper: 1) the activation normalization Kingma and Dhariwal (2018), 2) the batch normalization Ioffe and Szegedy (2015), and 3) the fully connected layers.

On the deep ResFlow, we use resolutions of $32 \times 32$ and $16 \times 16$ on both CIFAR-10 and CelebA. Otherwise, we use a single resolution of $32 \times 32$ on other experiments whose models have networks with layers less than 32. We apply the LipSwish activation (Chen et al., 2019) that provides the Lipschitz coefficients less than a unity. For the multi-GPU training, we use the Neumann log-determinant gradient estimator, instead of the memory efficient estimator (Chen et al., 2019).

## C.2 EXPERIMENTAL DETAILS

**Training** All the models are trained with batch size of 128 and EMA rate of 0.999. We train each model for around 10 days on four V-100 GPUs with 128Gb GPU memory for all experiments. At the initial training phase, we train the diffusion model parameters with fixed flow parameters initialized by the identity until the sample generation performance saturates. After this pretraining phase, we start updating both diffusion and flow models. In this learning phase, we apply the learning rate scheduling after the sample performance saturation. For the diffusion model, we drop the learning rate from $2 \times 10^{-4}$ to $10^{-5}$. For the flow model, we drop the learning rate from $10^{-3}$ to $10^{-5}$ for VPSDE and $5 \times 10^{-5}$ to $10^{-5}$ for VESDE. We emphasize this learning rate scheduling takes additional performance boost.

VESDE and VPSDE have different training details. VESDE assumes $\sigma_{min} = 10^{-2}$ on all experiments. VESDE has $\sigma_{max} = 50$ on CIFAR-10 and $\sigma_{max} = 90$ on CelebA. On the other hand, VPSDE assumes $\beta(t) = \beta_{min} + (\beta_{max} - \beta_{min})t$ with $\beta_{min} = 0.1$ and $\beta_{max} = 20$. Both VESDE and VPSDE truncate the diffusion time on $[\epsilon, T]$ in order to stabilize the diffusion model, where $\epsilon = 10^{-5}$ and $T = 1$. On sample generation, however, DDPM++/PDM with VPSDE draws samples by proceeding the generative diffusion process up to $\epsilon = 10^{-3}$, which corresponds to the diffusion variance to be $\sigma_{min} = 10^{-2}$. On the sample generation, we apply the Predictor-Corrector (PC) sampler Song et al. (2020) for NCSN++/PDM with VESDE and the Probability Flow (PF) ODE sampler Song et al. (2020) for DDPM++/PDM. Throughout the experiments, we provide the identical diffusion model structures to compare the baseline model and the PDM model in a fair setting.

**Evaluation** While we use the EMA rates of 0.999 for both diffusion and flow models for training, we use this EMA only for the diffusion model on the evaluation mode because the flow model without the EMA on the evaluation gives better samples. One peculiarity occurs in the experiment on the decoupled ResFlow. We empirically find that turning off the EMA for both diffusion model and flow model on the evaluation mode provides better performances of 3.11 in test NLL and 3.17 in test NELBO as reported in Table 1 on PDM (FID, dec). With the EMA applied to the diffusion model, we have 3.15 for test NLL and 3.22 for test NELBO, which shows slight degradation on the density estimation performance.

We compute the FID score based on Song et al. (2020) for CIFAR-10 with the modified Inception V1 network[1] in order to compare PDM to the baselines (Song et al., 2020; 2021) in a fair setting. On the

---

[1]https://tfhub.dev/tensorflow/tfgan/eval/inception/1

other hand, for the CelebA dataset, we compute the clean-FID (Parmar et al., 2021) that provides consistently antialiased results.

## C.3 VARIANCE REDUCTION

**Flow Training** When we train the flow network with $\mathcal{L}_{\phi,\boldsymbol{\theta}}(\{\mathbf{x}_t\}_{t=0}^T; g^2)$, this NELBO contains the integration of $\mathcal{L}_{\boldsymbol{\theta}}(\{\mathbf{z}_t\}_{t=0}^T; g^2)$. However, previous works on the diffusion models (Nichol and Dhariwal, 2021) show that the estimation variance is largely reduced with the importance sampling, which leads the performance variations (Song et al., 2021). Concretely, the importance sampling chooses an importance weight that is proportional to $\frac{g^2(t)}{\sigma^2(t)}$, and estimates the integration by $\mathcal{L}_{\boldsymbol{\theta}}(\{\mathbf{z}_t\}_{t=0}^T; g^2) = \int_0^T g^2(t)\mathcal{L}_t(\{\mathbf{z}_t\}_{t=0}^T; \boldsymbol{\theta})\,\mathrm{d}t \approx \sum_{n=1}^N \sigma^2(t_n)\mathcal{L}_{t_n}(\{\mathbf{z}_t\}_{t=0}^T; \boldsymbol{\theta})$, where $t_n$ is sampled from the importance distribution.

For VESDE, it satisfies $\beta(t) = 0$ and $g(t) = \sigma_{min}(\frac{\sigma_{max}}{\sigma_{min}})^t \sqrt{2\log(\frac{\sigma_{max}}{\sigma_{min}})}$. Throughout the experiments, we select $\sigma_{min} = 0.01$, and, $\sigma_{max} = 50, 90$ for CIFAR-10 and CelebA, respectively. The transition probability becomes $p_{0t}(\mathbf{z}_t|\mathbf{z}_0) = \mathcal{N}(\mathbf{z}_t; \mathbf{z}_0, \sigma^2(t))$, where $\sigma^2(t) = \int_0^t g^2(s)\,\mathrm{d}s = \sigma_{min}^2[(\frac{\sigma_{max}}{\sigma_{min}})^{2t} - 1]$. Song et al. (2020) approximates this variance to be $\sigma_{app}^2(t) = \sigma_{min}^2(\frac{\sigma_{max}}{\sigma_{min}})^{2t}$ so that the variance is proportional to $g^2(t)$. Therefore, the importance weight follows the uniform distribution, and the importance sampling is equivalent with choosing the uniform $t$.

On the other hand, VPSDE satisfies $\beta(t) = \beta_{min} + (\beta_{max} - \beta_{min})t$ with $g(t) = \sqrt{\beta(t)}$. Then, the transition probability becomes $p_{0t}(\mathbf{z}_t|\mathbf{z}_0) = \mathcal{N}(\mathbf{z}_t; \mu(t)\mathbf{z}_t, \sigma^2(t)\mathbf{I})$, where $\mu(t) = e^{-\frac{1}{2}\int_0^t \beta(s)\,\mathrm{d}s}$ and $\sigma^2(t) = 1 - e^{-\int_0^t \beta(s)\,\mathrm{d}s}$. Thus, VPSDE has the importance weight of $\frac{g^2(t)}{\sigma^2(t)} = \frac{\beta(t)}{1 - e^{-\int_0^t \beta(s)\,\mathrm{d}s}}$.

The Monte-Carlo sample from this importance weight is the solution of the inverse Cumulative Distribution Function (CDF) of the importance distribution as

$$t = F^{-1}(u) \tag{16}$$

$$\Longleftrightarrow u = F(t) = \frac{1}{Z}\int_\epsilon^t \frac{g^2(s)}{\sigma^2(s)}\,\mathrm{d}s = \frac{1}{Z}\big(\mathcal{F}(t) - \mathcal{F}(\epsilon)\big), \tag{17}$$

where $u$ is a uniform sample from $[0, 1]$, $\mathcal{F}(t)$ is the antiderivative of the importance weight given by $\mathcal{F}(t) = \log(1 - e^{-0.5t^2(\beta_{max} - \beta_{min}) - t\beta_{min}}) + 0.5t^2(\beta_{max} - \beta_{min}) + t\beta_{min}$, and $Z$ is the normalizing constant given by

$$
\begin{aligned}
Z &= \int_\epsilon^T \frac{g^2(t)}{\sigma^2(t)}\,\mathrm{d}t \\
&= \left[\log(1 - e^{-0.5t^2(\beta_{max} - \beta_{min}) - t\beta_{min}}) + 0.5t^2(\beta_{max} - \beta_{min}) + t\beta_{min}\right]_\epsilon^T \\
&= \log(1 - e^{-0.5T^2(\beta_{max} - \beta_{min}) - T\beta_{min}}) - \log(1 - e^{-0.5\epsilon^2(\beta_{max} - \beta_{min}) - \epsilon\beta_{min}}) \\
&\quad + 0.5(T^2 - \epsilon^2)(\beta_{max} - \beta_{min}) + (T - \epsilon)\beta_{min} \\
&= 23.86
\end{aligned}
$$

for $T = 1$ and $\epsilon = 10^{-5}$. The solution for the inverse CDF in Eq. 16 becomes

$$
\begin{aligned}
e^{\int_0^t \beta(s)\,\mathrm{d}s} &= 1 + \exp(Zu + \mathcal{F}(\epsilon)) \\
\Longleftrightarrow \int_0^t \beta(s)\,\mathrm{d}s &= \frac{1}{2}(\beta_{max} - \beta_{min})t^2 + \beta_{min}t = \log(1 + \exp(Zu + \mathcal{F}(\epsilon))) \\
\Longleftrightarrow t &= \frac{-\beta_{min} + \sqrt{\beta_{min}^2 + 2(\beta_{max} - \beta_{min})\log(1 + \exp(Zu + \mathcal{F}(\epsilon)))}}{\beta_{max} - \beta_{min}}.
\end{aligned}
$$

The variation of the Monte-Carlo diffusion time depends on the uniform sample of $u$.

**Diffusion Training** Now, on the side of training the diffusion network, as described in the main paper, selecting the weighting function heavily influences the model performances. For instance,

Table 7: Additional experimental results on the pretrained DDPM++ with VPSDE. The reported DDPM++ result (Song et al., 2021) is not fully reproducible in density estimation performances.

| Model | tolerance | NLL | NELBO | Gap | FID |
|---|---|---|---|---|---|
| DDPM++ (NLL, IS, reported) | $10^{-5}$ | 2.91 | 3.10 | 0.19 | 5.70 |
| DDPM++ (NLL, IS, ours) | $10^{-5}$ | 3.03 | 3.20 | 0.17 | **5.67** |
| | $10^{-1}$ | 3.00 | | 0.20 | 37.28 |
| PDM (NLL, dec, IS) | $10^{-5}$ | 2.94 | **2.95** | **0.01** | 6.84 |
| | $10^{-1}$ | **2.86** | | 0.10 | 8.14 |

Table 8: Additional experimental results for precision and recall.

| SDE | Model | NLL | NELBO | Precision | Recall | FID |
|---|---|---|---|---|---|---|
| VE | NCSN++ | 3.45 | 4.43 | **0.67** | **0.61** | **2.20** |
| | PDM (FID) | **3.04** | **3.09** | **0.67** | 0.60 | 3.36 |
| VP | DDPM++ (FID) | 3.13 | 3.29 | **0.68** | 0.60 | **3.16** |
| | PDM (FID, dec) | **3.11** | **3.17** | 0.65 | **0.61** | 4.43 |
| | DDPM++ (NLL, IS, ours) | 3.03 | 3.20 | 0.59 | **0.62** | **5.67** |
| | PDM (NLL, dec, IS) | **2.94** | **2.95** | 0.59 | 0.61 | 6.84 |

Song et al. (2021) compare the weighting functions of $g^2$ and $\sigma^2$, and Song et al. (2021) conclude that the weighting function of $\lambda(t) = \sigma^2(t)$ works the best for the sample generation, and $\lambda(t) = g^2(t)$ yields the best NLL performance. We train the diffusion model with $\sigma^2(t)$ for PDM (VE/VP, FID), and with $g^2(t)$ for PDM (VP, NLL) models.

# D  ADDITIONAL EXPERIMENTAL RESULTS

## D.1  DETAILED COMPARISON AND ABLATION ON TOLERANCE

Table 7 has two implications. First, we find that the reported NLL and NELBO performances are slightly better than what we actually obtain in our pretraining step, denoted by DDPM++ (NLL, IS, ours). Comparing with our results, optimizing PDM is indeed beneficial on density estimation, and variational gap is significantly reduced from 0.17 to 0.01. Second, we find the tolerance of $10^{-1}$ for the numerical ODE solver on computing NLL is the sweet spot on obtaining best NLL of 2.86, which beats LSGM-480M (NLL) on density estimation. This is noteworthy because our model uses 120M number of parameters, a quarter of parameters in LSGM. On the other hand, NELBO is independent of the tolerance because it does not use the numerical ODE solver, and Table 7 shows that variational gap is minimized when the tolerance is small enough.

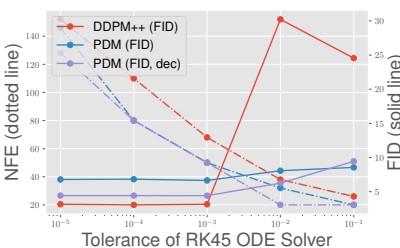

Figure 10: Ablation study on tolerance level of the RK45 ODE solver.

Figure 10 illustrates the ablation study on the tolerance for the ODE solver. Surprisingly, PDM is robust on the tolerance level up to $10^{-1}$, in contrast to LSGM with poor sample quality at $10^{-1}$ (Vahdat et al., 2021). From this robustness on the tolerance, we could reduce the sampling wall clock time from 360s (tolerance $10^{-5}$) to 72s (tolerance $10^{-1}$) for 1,024 samples, which is 5x faster, with a minor sacrifice in FID score from 6.79 to 8.56 in PDM (FID) with VPSDE.

## D.2  PRECISION AND RECALL

While FID measures the overall sample quality that counts both fidelity and diversity at once, we provide precision and recall (Kynkäänniemi et al., 2019) based on the modified Inception V1 network

Table 9: FID-5k of PDM (VE, FID) on CIFAR-10.

| | | SNR | | | |
| | | 0.14 | 0.145 | 0.15 | 0.155 |
|---|---|---|---|---|---|
| | 0.5 | 16.59 | 15.99 | 15.95 | 15.7 |
| | 0.6 | 12.61 | 12.13 | 12.17 | 11.99 |
| | 0.7 | 10.41 | 9.95 | 10.08 | 9.94 |
| | 0.8 | 9.2 | 8.75 | 8.94 | 8.84 |
| | 0.9 | 8.58 | 8.13 | 8.35 | 8.28 |
| Temperature | 1.0 | 8.31 | 7.85 | 8.07 | 8.06 |
| | 1.1 | 8.27 | **7.84** | 8.01 | 8.07 |
| | 1.2 | 8.49 | 8.11 | 8.21 | 8.31 |
| | 1.3 | 8.97 | 8.61 | 8.67 | 8.84 |
| | 1.4 | 9.76 | 9.44 | 9.45 | 9.73 |
| | 1.5 | 10.93 | 10.68 | 10.63 | 11.01 |

Table 10: FID-1k of PDM with VPSDE on CIFAR-10.

| Temperature | FID-1k | |
| | PDM (FID) | PDM (FID, dec) |
|---|---|---|
| 0.5 | 40.09 | 36.84 |
| 0.6 | 38.28 | 35.31 |
| 0.7 | 37.10 | 34.36 |
| 0.8 | 36.28 | 33.77 |
| 0.9 | 35.87 | 33.44 |
| 1.0 | 35.57 | 33.24 |
| 1.1 | 35.40 | **33.21** |
| 1.2 | 35.28 | 33.24 |
| 1.3 | **35.28** | 33.24 |
| 1.4 | 35.38 | 33.42 |
| 1.5 | 35.68 | 33.65 |

Table 11: FID-1k of PDM (VE, FID) on CelebA.

| | | SNR | | |
| | | 0.16 | 0.17 | 0.18 |
|---|---|---|---|---|
| | 0.8 | 17.37 | 16.88 | 17.8 |
| | 0.9 | 15.37 | 15.04 | 16.02 |
| | 1.0 | 14.72 | 14.56 | 15.37 |
| | 1.05 | 14.72 | **14.53** | 15.24 |
| Temperature | 1.1 | 15.00 | 14.83 | 15.47 |
| | 1.15 | 15.52 | 15.36 | 15.94 |
| | 1.2 | 16.3 | 16.14 | 16.71 |
| | 1.3 | 19.09 | 18.81 | 19.40 |

Table 12: FID-1k of PDM (VP, FID) on CelebA.

| Temperature | FID-1k |
|---|---|
| 0.8 | 15.60 |
| 0.9 | 15.27 |
| 1.0 | 15.04 |
| 1.05 | 14.95 |
| 1.1 | **14.90** |
| 1.15 | 14.94 |
| 1.2 | 15.05 |
| 1.3 | 15.44 |

[2] with 50k samples in order to separate the sample fidelity and diversity. Precision is a metric that measures the sample fidelity, and the recall is for the sample diversity. Table 8 indicates that both precision and recall remains at the identical scale, except for precision at PDM (FID, dec).

### D.3 EFFECT OF SIGNAL-TO-NOISE AND TEMPERATURE

As the flow colorizes the latent sample, we experiment the effect of temperature (Kingma and Dhariwal, 2018) on this colorization in Figure 11. With temperature $\tau$, the normalizing flow puts its latent input scaled by $\tau$ to the flow network. In our PDM, the image color with a higher temperature tends to be brighter, and we find that the optimal temperature depends on the experimental settings.

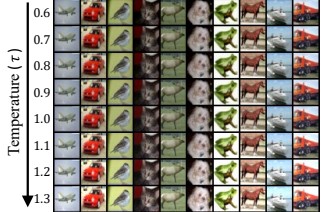

On sample generation, we draw latent samples from either the PC sampling or the PF ODE sampling. PC sampling assumes the Signal-to-Noise Ratio (SNR) that modifies the Corrector algorithm (Song et al., 2020) of the annealed Langevin dynamics (Song and Ermon,

Figure 11: Ablation study for the flow temperature.

2019). From the sampled latent variable, we scale by multiplying the temperature $\tau$ (Kingma and Dhariwal, 2018) to the latent variable. Since the diffusion model learns the global image structure, the normalizing flow only paints the color of the image, and the temperature controls the brightness of the generated image. This is a contrastive result to the previous flow models (Kingma and Dhariwal, 2018; Chen et al., 2019) that the temperature in the previous models controls both global context (such as the smoothness) and color. Therefore, we can create images with different colors by manipulating this temperature.

---

[2]Since we already used this inception network in calculating the FID score, we compute precision and recall based on this inception network. Note that we used this version of inception network to comply the baseline (Song et al., 2020).

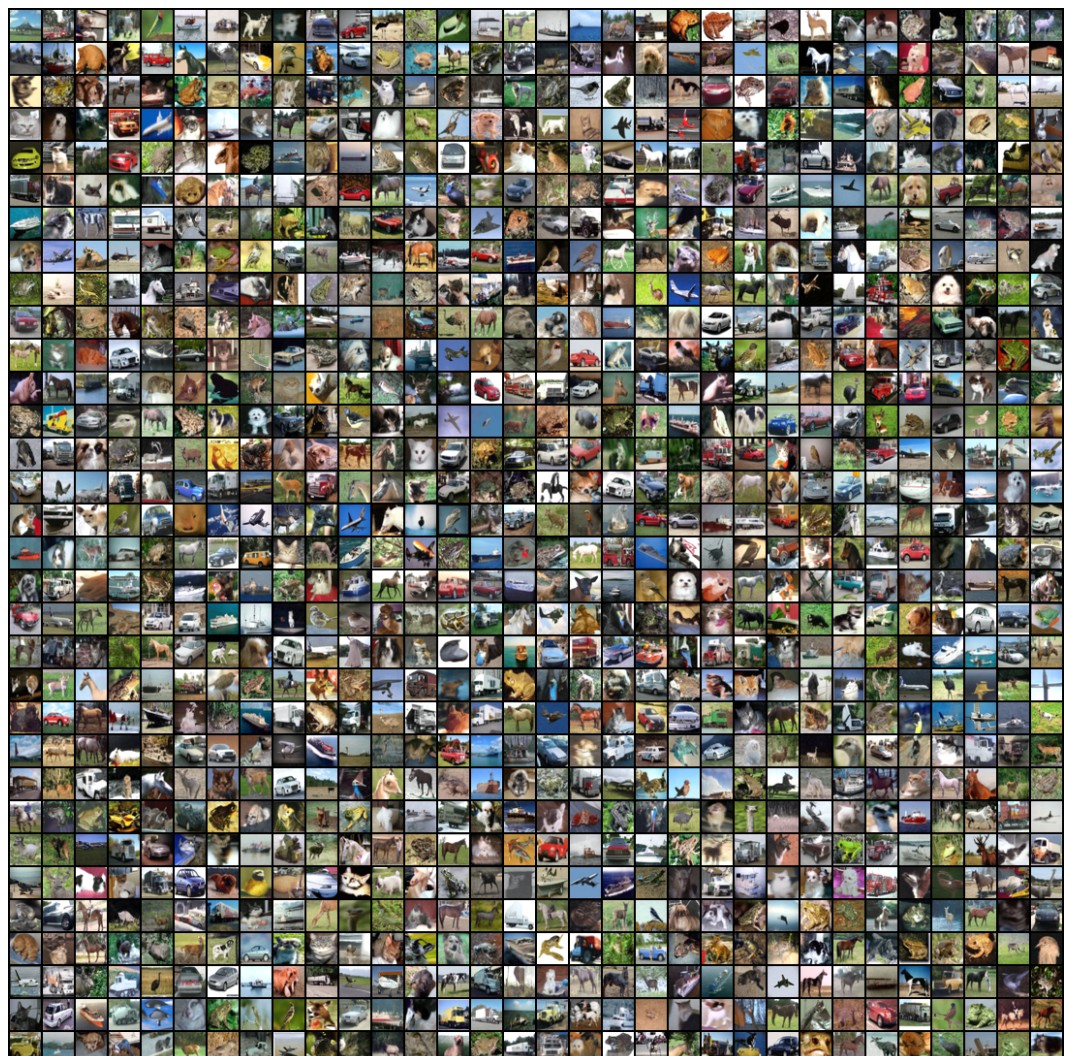

Figure 12: Non cherry-picked random samples from CIFAR-10 trained on PDM (VE, FID).

**CIFAR-10** With these hyperparameters of the SNR and the temperatures, we find optimal values in the below Tables. Table 9 computes the FID-5k score of all combinations for (SNR, $\tau$) in PDM (VE, FID) on CIFAR-10. Unlike the SNR of 0.16 that is previously suggested in (Song et al., 2020) for the vanilla diffusion model, we find that SNR of 0.145 consistently outperforming other values for SNR in terms of FID. Similarly, Table 9 presents that the optimal temperature is 1.1.

For VPSDE, Table 10 shows that the temperature of 1.3 performs the best for sample generation in PDM (VP, FID). For PDM (VP, FID, dec), we find that the temperature of 1.1 performs the best, and for PDM (VP, NLL, dec), the vanilla temperature of 1.0 generates the best samples.

**CelebA** Table 11 shows the FID-1k performance of PDM (VE, FID) on CelebA. It shows that the SNR of 0.17 proposed in (Song et al., 2020) performs the best also in our PDM. In addition, it shows that the optimal temperature is either 1.0 or 1.05. Table 12 shows the FID-1k performance of PDM (VP) on CelebA. It is clear in this table that the sample quality is the best on the temperature range of 1.05 to 1.15.

### D.4   RANDOM SAMPLES

Figure 12 and 13 show the non cherry-picked random samples from PDM (VE, FID) on CIFAR-10 and PDM (VP, FID) on CelebA, respectively.

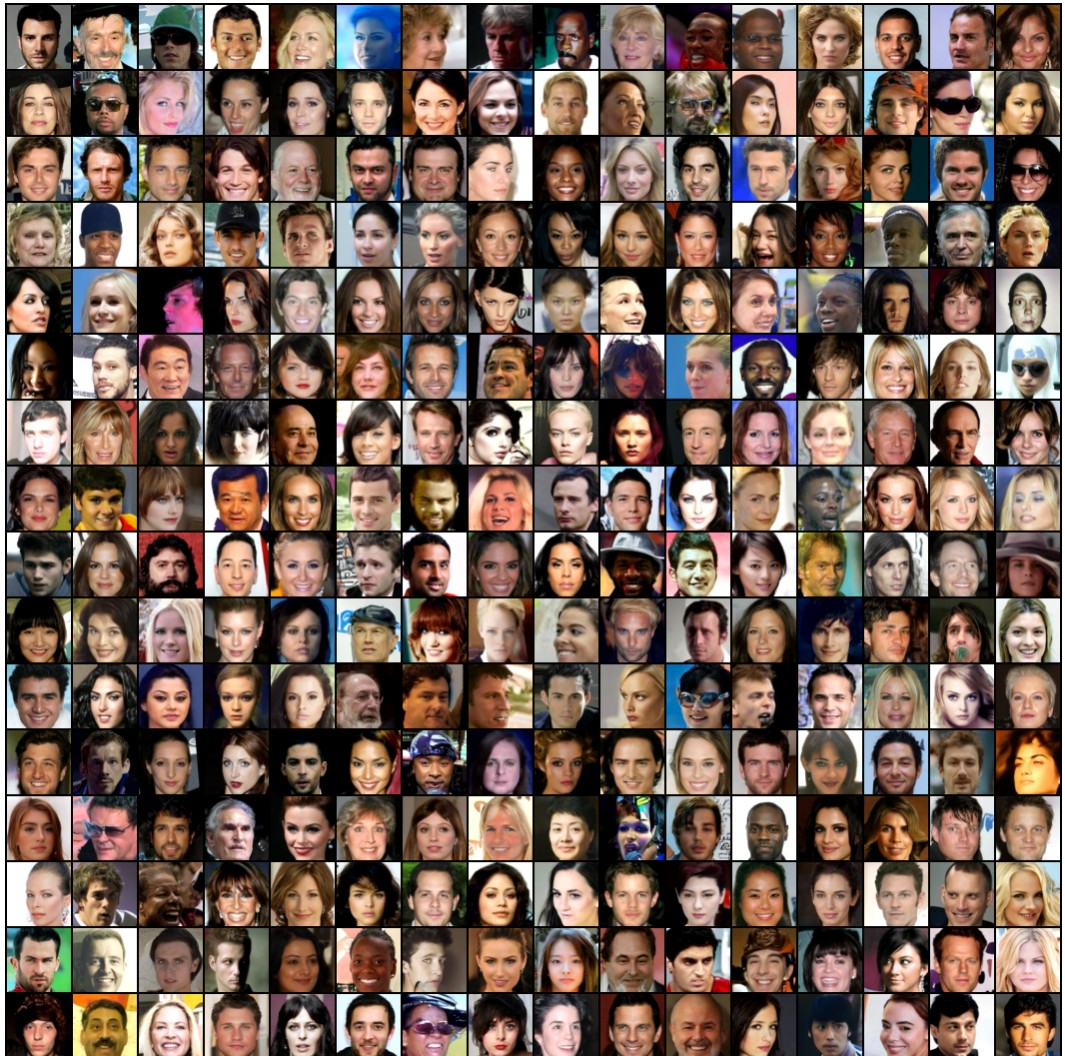

Figure 13: Non cherry-picked random samples from CelebA trained on PDM (VP, FID).

