# OpenReview forum: "Maximum Likelihood Training of Parametrized Diffusion Model"
_ICLR.cc/2022/Conference — ICLR 2022 Submitted_

### Official Review · Reviewer_qGLk · 2021-10-27

**Correctness:** 3
**Technical Novelty And Significance:** 3
**Empirical Novelty And Significance:** 2
**Recommendation:** 6
**Confidence:** 4

**Main Review:**

## Strength
The paper is clearly written and easy to understand. The overall idea and relationship to VAE are also discussed at the very beginning, which helps the understanding of the proposed method. For theoretical correctness, I have briefly checked the derivation, which seems to be correct. However, I am still a bit confused about some of the notations, which I will elaborate on later. Empirically, the author compared the proposed method with various diffusion models and demonstrated better NLL/ smaller variational gap and stable training.

## Weakness and potential improvement
I found the intuition behind the proposed method interesting and easy to follow, but I still have some concerns. First, if I understand correctly, the structure of PDM is very similar to LSGM mentioned in the related work. The only difference is the replacement of the encoder and decoder with normalizing flows. What are the actual empirical advantages? From the experiment, it seems that LSGM produces lower FID scores compared to PDM. Does PDM reduce the required number of generation steps like LSGM compared to typical linear diffusions?

For the evaluation metric, FID score is a good metric for overall sample quality, which captures both diversity and fidelity. However, since the PDM produces better NLL, I guess it can produce better distribution coverage than typical linear diffusions. So, if it does not trouble too much, **Precision** and **Recall** [1] are good metrics to separate the sample fidelity and diversity, which can provide more information than just FID.

From the generated images before and after normalizing flow, it seems that the normalizing flow only learns to colourize the image. Thus, most of the image details are generated by linear diffusions in latent space. My concern is that is it possible that the normalizing flow only learns trivial mappings? Why does normalizing flow not contribute to image details?

What is $\lambda$ in Eq.3?

At the beginning of section 4.2, I don't think all non-linear diffusions can be decoupled into normalizing flow + linear diffusion. Thus,  it is only fair to say that normalizing flow + linear diffusion produce non-linear diffusion.

What is $\sigma^2(t)$ in section 4.4?

In section 4.5 on page 7, I am still confused about why optimizing $\phi$ reduces the variational gap. I understand that the gap can be expressed as the difference in KL w.r.t. latent space path measures, but the variational gap also has the term $D_{KL}(p_\phi||p_\theta)$ which is also related to $\phi$. So why optimizing $\phi$ w.r.t. NELBO alone can reduce this gap?

[1] Tuomas Kynkaanniemi, Tero Karras, Samuli Laine, Jaakko Lehtinen, and Timo Aila. Improved precision and recall metric for assessing generative models.


**Summary Of The Paper:**

The author proposed a framework that combines normalizing flow with diffusion models, allowing non-linear diffusions to be used for inference and generation. Specifically, the author first discussed the similarity between VAE and diffusion models to motivate the importance of using non-linear diffusion and the reducing variational gap. Then, the author proposed to decouple the non-linear diffusion into an invertible transformation with the normalizing flow and typical linear diffusions on the transformed variables. A training objective based on variational lower bound was also proposed. Further, the author also showed that the variational gap between the lower bound and log-likelihood can be reduced with the proposed non-linear diffusions.

**Summary Of The Review:**

The paper is clearly written and I am also interested in the actual advantages of the proposed method compared to other diffusion models. If the author successfully addresses the above concerns, I can improve my rating of the paper.

---
I have read the revised version of the paper and the author's responses. It addressed most of my concerns. However, although I understand the difference between PDM and LSGM from a theoretical point of view, I think demonstrating it empirically is also important because the inter-changeability between latent space and data space during inference is mainly due to the merit of normalizing flow, not the PDM framework. Since concatenating the flow and diffusion model is straightforward, the resulting model (from a structure point of view) has limited novelty. On the other hand, I think the argument of reducing the variational gap is interesting, and along with extensive empirical evaluations, I will raise my score to 6. But I still suggest the author to further argue about the novelty of PDM from both theoretical and empirical points of view.

---

> ### Author Response · Authors · 2021-11-12
> **Continued**
>
> **Q. *Does PDM reduce the required number of generation steps like LSGM compared to typical linear diffusions?***
>
> A.	**[Robust Sample Fidelity to ODE Error]** Thank you for the comment! We additionally experimented as you suggested, and Figure 10 in the revised paper illustrates the result. Figure 10 shows that PDM attains the similar behavior to LSGM with respect to the number of generation steps that PDM is extremely robust to the ODE solver error tolerance. In particular, the sample performance remains stable for a high level of the tolerance with 0.1, which performed poor sample fidelity in DDPM++. It is noteworthy that this robustness to the tolerance has not been observed even in LSGM (Figure 4 of LSGM paper.)
>
> **[Mechanism for Robustness]** We attribute the normalizing flow to this nearly tolerance-free sample fidelity. The vanilla diffusion process without the normalizing flow starts from a data distribution whose support is strictly bounded on $[0,1]^{d}$. On the other hand, the diffusion process in PDM is free from such boundary condition because the normalizing flow transforms the data distribution to the latent distribution, which could span arbitrarily vast area. Since the volume of latent is extensive compared to the volume of data, the relatively high error in the numerical ODE barely harms the sample fidelity. Eventually, the Euler-Maruyama sampler with RK45 method with tolerance of 0.1 takes 72 seconds for generating 1,024 samples, which is 5x faster than the same sampler with tolerance of 1e-5 (360 seconds), with a minor sacrifice in FID score (from 6.79 to 8.56 for *PDM (FID)*).
> \
> \
> **Q. Precision and Recall are good metrics to separate the sample fidelity and diversity.**
>
> A.	Thank you for your suggestion! We have computed the precision and the recall, and you can check the performance in Table 8 of the revised paper. As you explained, the precision is the metric for sample fidelity and the recall is for the sample diversity. Table 8 indicates three implications. First, PDM on VESDE has similar sample fidelity to NCSN++ from the identical precision at the expense of slightly lowered sample diversity. In particular, on VESDE, we emphasize that our PDM largely improves the density estimation performances (NLL, NELBO) while maintaining the sample quality and diversity in terms of the precision and the recall at the identical scale. Second, PDM (FID) on VPSDE draws samples as diverse as the DDPM++ (FID) samples, but the sample quality of PDM (FID) is sacrificed compared to DDPM++ (FID). Third, PDM (NLL, dec, IS) experiment on VPSDE shows similar tendency to the experiment on VESDE, i.e., our PDM improves density estimation and performs similar precision and the recall performances to DDPM++ at the cost of small degradation on FID score. Summing all together, as the reviewer expected, we observe that the recall performance remains comparable to the original diffusion models, which has not been observed in FID score.
> \
> \
> **Q. What is $\lambda$ and $\sigma^{2}(t)$?**
>
> A.	**[Definitions]** Thank you for missing points. In Eq. (3), $\lambda$ is the weighting function that determines the level of contribution for each diffusion time on the overall loss. In Section 4.4, $\sigma^{2}(t)$ is the variance of the perturbation probability. It means that when the diffusion process is defined by $\mathrm{d}\mathbf{x_{t}}=\mathbf{f}(\mathbf{x_{t}},t)\mathrm{d}t+g(t)\mathrm{d}\omega_{t}$, the perturbation probability becomes $p_{0t}(\mathbf{x_{t}}\vert\mathbf{x_{0}})=\mathcal{N}(\mathbf{x_{t}};\mu(t)\mathbf{x_{0}},\sigma^{2}(t)\mathbf{I})$, where $\mu(t)$ and $\sigma^{2}(t)$ determines the shape of the perturbation probability. $\sigma^{2}(t)$ is determined by $g(t)$ as $\sigma^{2}(t)=\int_{0}^{t}g^{2}(t)\mathrm{d}t$. We explained about this in the revised paper.
> \
> \
> \
> [1] Anonymous, “Score-Based Generative Modeling with Critically-Damped Langevin Diffusion.” Under review in ICLR 2022, https://openreview.net/forum?id=CzceR82CYc
>
> [2] Grcić, Matej, Ivan Grubišić, and Siniša Šegvić. "Densely connected normalizing flows." arXiv preprint arXiv:2106.04627 (2021).
>
> [3] Song, Yang, et al. "Maximum Likelihood Training of Score-Based Diffusion Models." arXiv e-prints (2021): arXiv-2101.

---

> ### Author Response · Authors · 2021-11-12
> **Continued**
>
> **Q. *What are the actual empirical advantages?***
>
> A.	**[Comparison to LSGM-109M]** The original LSGM network attains 480M number of parameters, while our PDM has 120M parameters. Recently, to compare LSGM with the baseline NCSN++/DDPM++ with 108M parameters, [1] reports the result of LSGM with 109M parameters, which is denoted by LSGM-109M (balanced). Here, *balanced* means that the diffusion model for LSGM is trained with its weighting function defined by $\lambda(t)=g^{2}(t)\sigma^{2}(t)$. Table 1 in the revised paper presents that PDM outperforms LSGM-109M for FID in *PDM (FID, dec)* and NLL in *PDM (NLL, dec, IS)*. Note that [1] points out that the training of LSGM-109M (FID) is unstable.
>
> **[Comparison to LSGM-480M]** Table 7 in the revised paper shows that the numerical ODE solver with the tolerance of $0.1$ is the sweet spot for the best NLL performance. *PDM (NLL, dec, IS)* on VPSDE performs 2.86 for NLL with the tolerance of $0.1$, which beats the best LSGM NELBO performance (*LSGM-480M (NLL)*) with only using a quarter number of parameters. Also, *PDM (NLL, dec, IS)* outperforms *LSGM-480M (NLL)$ in FID score.
>
> **[Comparison to Baseline]** Our Table 1 in the revised paper compares PDM with the baseline NCSN++/DDPM++. The revised table demonstrates that PDM beats the baselines in NELBO in all the experimental cases. Also, Table 7 shows that the performance of our DDPM++ is slightly lower than that of the reported performance from [3]. Comparing PDM to the DDPM++ for our setting, we conclude that applying PDM significantly reduces both NLL and NELBO.
>
> **[Training Stability]** We emphasize that the training of PDM is highly stable as illustrated in Figures 5 and 6. In particular, one can observe contrastive test NELBO performance of PDM and the baseline in CIFAR-10 on a shallow network. When the network gets deeper, NLL on the test dataset for PDM in Figure 6 is lower than that of the baseline (linear) diffusion model. Throughout the experiment, we have used the identical network structures for PDM and the baseline diffusion model, and the number of the flow network is kept to be of 1/10 scale of the number of parameters for the diffusion model.
>
> **[Robust Performance without Importance Sampling]** Table 1 in the revised paper provides us another valuable empirical advantage. A common ground of previous research validify that the importance sampling is critical on the performance, but out PDM turns out to be robust on this importance sampling. The experiments with/without the importance sampling perform similar in both density estimation and sample generation.
>
> **[SOTA in CelebA]** While the performance on CIFAR-10 is not satisfactory, we claimed that PDM achieved the best FID performance in CelebA 64x64, and this is the state-of-the-art performance. We discussed the reason behind PDM's advantage particularly in high-dimensional sample generations in Section 6.1. Please count this SOTA performance on your evaluation.

---

> ### Author Response · Authors · 2021-11-12
> **Continued**
>
> **Q. *I don’t think all non-linear diffusions can be decoupled into normalizing flow + linear diffusion. Thus, it is only fair to say that normalizing flow + linear diffusion produce non-linear diffusion.***
>
> A.	Thank you for your detailed review. It is correct that our PDM cannot cover all the nonlinearity of the diffusion process. Instead, PDM covers nonlinearity with specific form, explained in Appendix B in details. We have tried to prove that PDM covers all the nonlinearity, but it turns out that there are some diffusion processes that PDM cannot cover with. This is a potential disadvantage of PDM, and we leave expanding the scope of nonlinearity as a further research. Having such limitation, we still value our PDM that it uses methodological and mathematical analyses that has not been tried so far in the community of diffusion models.
> \
> \
> **Q. Is it possible that the normalizing flow only learns trivial mappings? Why does normalizing flow not contribute to image details?**
>
> A.	**[Incompatibility of Identity and Optimality]** When the normalizing flow is a trivial (identity) mapping, it reduces to the original (linear) diffusion model. From the argument of the answer in **Q. *I am still confused about why optimizing $\phi$ reduces the variational gap. Why optimizing $\phi$ w.r.t. NELBO can reduce this gap?***, we have that the original diffusion does not satisfy $p_{\theta}=p_{r}$ for any case $\theta$ because of the mismatch on $p_{T}$ and $\pi$. However, the optimal point of PDM guarantees that $p_{\phi^*,\theta}=p_{r}$, so we conclude that the identity mapping and the optimality are incompatible each other.
>
> **[From Identity to Optimality]** Empirically, PDM with more emphasis on flow (i.e., more parameters on the flow network and less parameters on the diffusion network) performs significantly worse, and our setting is the sweet spot in terms of the sample performance. Actually, this is not an original observation because the normalizing flow models in general generates unrealistic images (see for instance Figure 8 of [2] with FID score of 34.9). Therefore, we rely more on the diffusion model than on the flow model for better sample fidelity. Acknowledging the original (linear) diffusion setting, we used the Residual Flow initialized with almost trivial mapping. As we assert in the above paragraph, the flow network approaches towards the (local) optimality as training proceeds, which is clearly NOT the trivial mapping, and this is evidenced in Figure 7 of the main paper.
>
> **[Bipartite Roles]** PDM with a deeper flow architecture acts in a similar way: the diffusion captures the global texture, and the flow learns coloring the sampled global context. How to explain this behavior? Recall that the major source of poor sample generation in normalizing flows comes from the poor global context [2]. In other words, the samples from normalizing flows in general colorize the image well, but a generated sample is hardly perceived as a real image because of its crude global structure. For instance, the current best normalizing flow model creates highly unrealistic sample in Figure 8 of [2] (FID 34.9). This is the fundamental issue in normalizing flows arising from the inflexible expressive power induced by the invertibility property. Having said that normalizing flow has weakness on global context, diffusion models do not suffer from this weakness. Therefore, optimizing the combined model naturally separate their roles in image generation: the flow model colorize the latent, and the diffusion model focus more on capturing global context.

---

> ### Author Response · Authors · 2021-11-12
> **Continued**
>
> **Q. *I am still confused about why optimizing $\phi$ reduces the variational gap. Why optimizing $\phi$ w.r.t. NELBO can reduce this gap?***
>
> A.	**[Theoretic Observation]** Theorem 2 states that for any $\theta$, there exists an optimal $\phi^*$ that simultaneously satisfies the following three conditions: 1) the variational gap $Gap_{d}(\mu_{\phi^*}^{d},\nu_{\phi^*,\theta}^{d})$ is minimized to be zero, 2) the NELBO $D_{KL}(\mu_{\phi^*}^{d}\Vert\nu_{\phi^*,\theta}^{d})$ is minimized to be zero, 3) the NLL $D_{KL}(p_{\phi^*}\Vert p_{\theta})$ is minimized to be zero. Conversely, if $\phi^*$ is the optimum point of NELBO $D_{KL}(\mu_{\phi}^{d}\Vert\nu_{\phi,\theta}^{d})$, then it implies that the generative distribution $p_{\phi^*,\theta}$ equals to $p_{r}$ because the generative distribution is the marginal probability of $\nu_{\phi^*,\theta}$ and the real distribution is the marginal probability of $\mu_{\phi^*}^{d}$, where two of the path measures coincide each other. Therefore, it indicates that the optimal point of NELBO satisfies $D_{KL}(p_{r}\Vert p_{\phi^*,\theta})=D_{KL}(p_{\phi^*}\Vert p_{\theta})=0$, so the variational gap at $\phi^*$ becomes zero. The above argument points out that the variational gap becomes zero if and only if NELBO is minimized w.r.t. $\phi$.
>
> **[Strictly Positive Variational Gap]** To compare with the baseline (linear) diffusion process in the data space, suppose we do not assume any normalizing flow jointly modeled with the diffusion model. Then, the variational gap $Gap(\mu,\nu_{\theta})=\mathbb{E}\big[D_{KL}(\mu(\mathbf{x_{0:T}}\vert\mathbf{x_{0}})\Vert\nu_{\theta}(\mathbf{x_{0:T}}\vert\mathbf{x_{0}}))\big]$ becomes strictly positive because the end variable of the inference path measure $\mu$ and the generative path measure $\nu_{\theta}$ are different. Concretely, $\mu$ ends at $p_{T}$, and $\nu_{\theta}$ ends at $\pi$ (see Eq. (1) and (2)), and the mismatch of $p_{T}$ and $\pi$ (because we are diffusing the data finite time) results in strictly positive variational gap at any case of $\theta$. This is the qualitative improvement from the vanilla diffusion model.
>
> **[Optimal Generative Distribution is Not Data Distribution]** One could think that the joint modeling of flow and diffusion models only make the parameter optimization more akin to MLE update by reducing the variational gap. Indeed, there is another important motivation of PDM that clarifies the validity. To begin with, observe that NELBO is equivalent to $\mathbb{E}[\Vert s_{\theta}(\mathbf{x_{t}},t)-\nabla\log{p_{t}(\mathbf{x_{t}})}\Vert_{2}^{2}]$, so the optimum point of NELBO becomes $s_{\theta^*}(\mathbf{x_{t}},t)=\nabla\log{p_{t}(\mathbf{x_{t}})}$. Then, the optimal generative distribution becomes $p_{\theta^*}(\mathbf{x_{0}})=\int \pi(\mathbf{x_{T}})p_{T0}(\mathbf{x_{0}}\vert\mathbf{x_{T}})\mathrm{d}\mathbf{x_{T}}$, where $p_{T0}(\mathbf{x_{0}}\vert\mathbf{x_{T}})$ is the generative transition probability. Since we are assuming the optimum score that exactly estimates the true data score, this generative transition probability is the exact transition probability of the reverse process, so the data distribution becomes $p_{r}(\mathbf{x_{0}})=\int p_{T}(\mathbf{x_{T}})p_{T0}(\mathbf{x_{0}}\vert\mathbf{x_{T}})\mathrm{d}\mathbf{x_{T}}$. Therefore, the optimal generative distribution $p_{\theta^*}$ will not coincide with the real distribution $p_{r}$ due to the mismatch of $p_{T}$ and $\pi$, and this is potentially a huge problem in the community of diffusion models.
>
> **[Optimality Implies Real Distribution in PDM]** On the other hand, if we join flow models to diffusion models, Theorem 2 implies that for *any* diffusion parameter $\theta$, there exists a flow parameter $\phi^*$ such that the generative distribution $p_{\phi^{*},\theta}$ matches to the real distribution $p_{r}$, so the above mentioned mismatch with the optimal generative distribution to the real distribution is resolved. To the best of our knowledge, this is the first work in diffusion models that has its optimal point to be the real distribution. All other diffusion models does not satisfy this important property.

---

> ### Author Response · Authors · 2021-11-12
> **Thank you for the integrated feedback**
>
> Thank you for the detailed review and thoughtful feedback. Below we address specific questions.
> \
> \
> **Q. *The structure of PDM is very similar to LSGM mentioned in the related work. The only difference is the replacement of the encoder and decoder with normalizing flows.***
>
> A.	**[Interpretation is the Value of PDM]** Despite of the structural similarity, there are huge differences of PDM to LSGM: the diffusion process in the latent space can be lifted up to a (nonlinear) diffusion process in the data space interchangeably *only for PDM*. From the invertibility of normalizing flows, we can transform the diffusion process in the data space to the latent space at any time back and forth. While LSGM is also able to induce a diffusion process in the data space *from* the linear diffusion process in the latent space through the decoder (by Ito’s lemma), the reverse direction is not achievable because the encoder is theoretically impossible to be invertible (see the answer to **Q. *While it is true that the encoder/decoder are not formally exact inverses of each other, in practice they do almost perfectly invert each other.*** explained in the first reviewer’s panel). Therefore, the inference part of LSGM cannot be interpreted as a (forward) diffusion process in the data space. This makes LSGM a half-way diffusion process that has only the generative diffusion. The invertibility in PDM makes the diffusion in the data and the latent space completely equivalent, which gives the originality of PDM.

---

### Official Review · Reviewer_wDxP · 2021-10-31

**Correctness:** 4
**Technical Novelty And Significance:** 3
**Empirical Novelty And Significance:** 3
**Recommendation:** 5
**Confidence:** 4

**Main Review:**

This paper is set up in the emerging toping of generative diffusion models, that is gaining some important momentum. Such models define a forward and a backward pass, that respectively consist in going from the signal domain to noise and back from noise to the signal domain.
Applications of the backward diffusion as a generative model is a very effective and popular way to achieve generative modeling.

As highlighted by the authors, current applications of such diffusion models come with linear dynamics, since those were the only ones so far to allow for close-form expressions for the required densities as a function of the time step. Their point is that such linear dynamics are limiting the expressive power of such models.

As a convenient way to circumvent this limitation, their solution is fundamentally quite simple: they propose to jointly train a normalizing flow from the data space, from which a classical linear diffusion is performed. The key point is that the method actually turns out to be equivalent to a nonlinear diffusion that would take place in the data domain, and they explicitly derive its dynamics by exploiting the Ito lemma and the fact that normalizing flows are invertible. The paper comes with much theoretical derivations that should be sufficient to support their claims.

As can be seen, I tend to think that this paper does have merits and that the study definitely may be inspiring to colleagues working on diffusion models. Although the experiments are limited, they are sufficiently convincing to me.

However, I must also say that all these remarkable developments come with an unacceptably low quality of english usage. A very important number of sentence from this paper comes with typos or with awkward usage of the wrong english words.
* The most annoying systematic typo is the systematically wrong usage of the word "the". That word is very often written at incorrect places, which reads very awkward. I am  talking of something like a hundred of occurrences.
* Then, there are other more punctual typos or awkward sentences, that I shortly review later below.

For the reason that this english usage is not acceptable I am sorry to say that I must recommend rejection. I however strongly encourage the authors to have the paper (heavily) proofread by a native english speaker and then submit it again somewhere else.

Comments on the go:

* "by maintaining the static linear diffusion": have not innovated on the topic of the diffusion mechanism that was always kept linear.
* You don't define your acronyms VE VP, SDE, MLE, etc
* Practically : in practice
* In equation 3, you should explain what P_{0t}(xt|x0) is and how you compute it
* more close to: closer to
* To be concrete: in practice
* All the suggested SDEs: As far as we know, all previously proposed
*  worth to note: worth noting
* in equation (7), how do you write and compute P_{0t}(z_t|z_0)
* sentences before 4.4 awkward.
* your ethics statements are largely exagerated. It reads like you are proposing a super dangerous technology, whereas you are basically proposing some variation over a quite established technology. Please strongly tune down. ("destroy the evidence-based justice system", come on...)



**Summary Of The Paper:**

The core contribution of the paper is to show how a nonlinear diffusion model may be constructed by cascading a normalizing flow with a linear diffusion in the resulting latent domain. The paper comes with a strong theoretical flavour and the proposed approach is well grounded.
Applications in classic image generation tasks are provided.

**Summary Of The Review:**

very interesting paper but with a very questionable english usage

---

> ### Author Response · Authors · 2021-11-12
> **Thank you for the integrated feedback**
>
> Thank you for your detailed review, and we appreciate your comments on our English. While we appreciate your statements that you could understand our theoretic and empirical contributions, we share the importance of good writings, and we promise that we will improve the writing in our final draft.
>
> It will be great if reviewers can provide us discussion points other than English, so that we can clarify our intentions. It will be great disappointments to us if this paper is rejected only because of poor English writing.
>
> Answer to comments on the go (about English)
> -	“by maintaining…..” : we updated the line as your suggestion in our revised paper.
> -	VE, VP, SDE, MLE : we defined these acronyms in our revised paper.
> -	Practically : we changed the word to the phrase, “in practice”
> -	More close to : Our mistake in grammar, and we changed the wording to “closer to”
> -	To be concrete : we changed the phrase to be “Formally”
> -	All the suggested SDEs : we updated the phrase to be “as far as we know, all previously proposed”
> -	Worth to note : we changed the wording to be “worth noting”
> -	Sentences before 4.4 : we revised the paragraph before 4.4
> -	Ethics statement : we revised the ethical statements, and please understand that the statement is written by a graduate student whose first language is not English.
>
> Answer to comments on the go (about equations)
> -	**[Transition Probability]** $p_{0t}(\mathbf{x_{t}}\vert\mathbf{x_{0}})$ is the (potentially nonlinear) transition probability of diffusion process. Here, $\mathbf{x_{0}}$ is the data instance, and $\mathbf{x_{t}}$ is the diffused data instance with noises over time progression. Otherwise the diffusion is linear that attains a closed-form (Gaussian distribution) transition probability, the transition probability in general requires a PDE/ODE solver to estimate [1], so it is strictly prohibitive applying nonlinear diffusion of which transition probability is intractable. Please refer the paper from Yang Song [1] for further explanation. We have added the definition of the transition probability in Eq. (3).
> -	**[Transition Probability in Latent Space]** $p_{0t}(\mathbf{z_{t}}\vert\mathbf{z_{0}})$ follows the same argument in the above answer, and the only difference is whether the calculation occurs on either latent space or data space. Having said that, we defined the diffusion process of the latent space to be linear, whose transition probability is derived to be a closed-form Gaussian distribution. Our contribution is bringing nonlinear and intractable $p_{0t}(\mathbf{x_{t}}\vert\mathbf{x_{0}})$ in the data space to the linear and tractable $p_{0t}(\mathbf{z_{t}}\vert\mathbf{z_{0}})$ in the latent space. We have defined the transition probability in latent space in Eq. (7).
>
> [1] Song, Y., Sohl-Dickstein, J., Kingma, D. P., Kumar, A., Ermon, S., and Poole, B. (2020). Score-based generative modeling through stochastic differential equations. arXiv preprint arXiv:2011.13456.

---

> ### Author Response · Authors · 2021-11-18
> **Please review the updated revision**
>
> We also update the usage of "the" through the paper. Please review the paper for one more time to see whether the current writing is satisfactory or not.

---

### Official Review · Reviewer_LLCZ · 2021-11-01

**Correctness:** 3
**Technical Novelty And Significance:** 3
**Empirical Novelty And Significance:** 3
**Recommendation:** 6
**Confidence:** 3

**Main Review:**

The idea of injecting nonlinearity to a linear diffusion using normalizing flow is very interesting, although somewhat similar to [Song et al. 2021]. I find the experiments section quite convincing. In particular, the variational gaps obtained using the proposed method are a lot lower than the alternatives.

I find the exposition of the paper very hard to follow. Many sentences do not make sense to me, and notations are used without being defined properly.

Detailed comments:
- In Eq (3), some reference about how $\mathcal{L}$ appears would be helpful. Also, mention again how $\mathcal{L}$ is defined in Sec 3.2 where it is used. Throughout, the role of $\mathcal{L}$ is really mysterious to me.
- Above Eq (6), "To connect this VAE ELBO to the diffusion loss, ..." How is (6) related to diffusion? I found the connection a bit far-fetched, if it's just regarding $q(x|z)$ and $p(z|x)$ are "forward" and "backward" directions respectively.
- In Sec 3.2, how does the last equation follow?
- In the last paragraph of Sec 3.2, I don't understand how the variational gap has to be strictly positive for diffusion models. Why couldn't $\nu_\theta$ be arbitrarily close to $\mu$ as we optimize $\theta$? How is this really different from the nonlinear case Eq(10)?
- In the last sentence in Sec 4.2, "from the prior to $z_t^\theta$" I think it should be $z_0^\theta$.
- Before Theorem 2, it claims "this variational gap converges to zero as we train ...". This does not seem correct. The theorem mentions nothing about convergence, but only that at optimal the gap becomes zero. The actual convergence is hard to guarantee due to the non-convexity of the optimization.




**Summary Of The Paper:**

The paper proposes parameterized diffusion model (PDM) that combines a normalizing flow on top of a linear diffusion model, effectively modeling a nonlinear diffusion. Empirically, such nonlinearity results in a tighter variational gap compared to the linear diffusion and other baselines.

**Summary Of The Review:**

The paper proposes an interesting way to model nonlinear diffusion using normalizing flows, and presents convincing experiments results to support the main claims, although the clarity of the writing can be improved.

---

> ### Author Response · Authors · 2021-11-12
> **Continued**
>
> **Q. Derivation of the last equation of Section 3.2.**
>
> A.	**[Derivation]** We provide the derivation of the last equation of Section 3.2 by below. Also, we have revised our paper to include the below derivation step-by-step in Appendix B.2.
>
> $D_{KL}(\mu\Vert\nu_{\theta})=\int\mu(\mathbf{x_{0:T}})\log{\frac{\mu(\mathbf{x_{0:T}})}{\nu_{\theta}(\mathbf{x_{0:T}})}}\mathrm{d}\mathbf{x_{0:T}}=\int p_{r}(\mathbf{x_{0}})\mu(\mathbf{x_{0:T}}\vert\mathbf{x_{0}})\log{\frac{p_{r}(\mathbf{x_{0}})\mu(\mathbf{x_{0:T}}\vert\mathbf{x_{0}})}{p_{\theta}(\mathbf{x_{0}})\nu_{\theta}(\mathbf{x_{0:T}}\vert\mathbf{x_{0}})}}\mathrm{d}\mathbf{x_{0:T}}$
> $=\int p_{r}(\mathbf{x_{0}})\mu(\mathbf{x_{0:T}}\vert\mathbf{x_{0}})\log{\frac{p_{r}(\mathbf{x_{0}})}{p_{\theta}(\mathbf{x_{0}})}}\mathrm{d}\mathbf{x_{0:T}}+\int p_{r}(\mathbf{x_{0}})\mu(\mathbf{x_{0:T}}\vert\mathbf{x_{0}})\log{\frac{\mu(\mathbf{x_{0:T}}\vert\mathbf{x_{0}})}{\nu_{\theta}(\mathbf{x_{0:T}}\vert\mathbf{x_{0}})}}\mathrm{d}\mathbf{x_{0:T}}$
> $=\int p_{r}(\mathbf{x_{0}})\log{\frac{p_{r}(\mathbf{x_{0}})}{p_{\theta}(\mathbf{x_{0}})}}\mathrm{d}\mathbf{x_{0}}+\int p_{r}(\mathbf{x_{0}})\mu(\mathbf{x_{0:T}}\vert\mathbf{x_{0}})\log{\frac{\mu(\mathbf{x_{0:T}}\vert\mathbf{x_{0}})}{\nu_{\theta}(\mathbf{x_{0:T}}\vert\mathbf{x_{0}})}}\mathrm{d}\mathbf{x_{0:T}}$
> $=D_{KL}(p_{r}\Vert p_{\theta})+E_{p_{r}(\mathbf{x_{0}})}[D_{KL}(\mu(\mathbf{x_{0:T}}\vert\mathbf{x_{0}})\Vert\nu_{\theta}(\mathbf{x_{0:T}}\vert\mathbf{x_{0}}))]$
> \
> \
> **Q. Why couldn’t $\nu_{\theta}$ be arbitrarily close to $\mu$ as we optimize $\theta$ in the vanilla diffusion model? How is this really different from the nonlinear case Eq. (10)?**
>
> A.	**[Strictly Positive Variational Gap of Linear Diffusion]** We hugely appreciate to this point because this is a missing argument of our submitted version. We included new illustrations (Figures 8 and 9) to answer this question. This question clarifies the value of Theorem 2 in our contribution. We observe that there is slight mismatch between $p_{T}$ and $\pi$ because the data distribution $p_{r}$ is not fully diffused with finite time of the linear diffusion. Then, $\mu$ and $\nu_{\theta}$ never meet each other by the following reasoning. From Eq. (1), the (forward) path measure $\mu$ is identically generated from the reverse diffusion process starting from $p_{T}$. Also, Eq. (2) creates the generative path measure $\nu_{\theta}$ starting from $\pi$. Therefore, $\mu$ and $\nu_{\theta}$ are different on their starting random variables (one for $p_{T}$ and the other for $\pi$), and this leads that those measures would not coincide each other at any case, illustrated in Figure 8 of the revised paper. Concretely, when the NELBO becomes optimal, the score network perfectly estimates the data score, $s_{\theta}(\mathbf{x_{t}},t)=\nabla\log{p_{t}(\mathbf{x_{t}})}$. Then, once $\mathbf{x_{T}}$ is given, the data denoising steps of the reverse diffusion and the generative diffusion coincides. Even at the optimum of the diffusion loss, since the starting variables $p_{T}$ and $\pi$ are different, the two path measures are not equal. In consequence, the variational gap of $\text{Gap}(\mu,\nu_{\theta})= E_{p_{r}(\mathbf{x_{0}})}[D_{KL}(\mu(\mathbf{x_{0:T}}\vert\mathbf{x_{0}})\Vert\nu_{\theta}(\mathbf{x_{0:T}}\vert\mathbf{x_{0}})]$ cannot be zero in any case.
>
> **[Reduced Variational Gap of PDM]** On the other hand, for any score network $\theta$, suppose the generative process starting from $\pi$ ends at $q_{0}^{\theta}$ on the latent space. If the normalizing flow $\mathbf{h_{\phi}}$ maps from $p_{r}$ to $q_{0}^{\theta}$ (i.e., the pushforward of $p_{r}$ under $\mathbf{h_{\phi}}$ is $q_{0}^{\theta}$), then the forward path measure $\mu_{\phi}$ on the latent space ends at $\pi$, and the forward path measure $\mu_{\phi}$ coincides with the generative path measure $\nu_{\theta}$ on the latent space as in Figure 9. This is evidenced by the minimized variational gap in Table 1 and 2.
> \
> \
> **Q. In the last sentence in Section 4.2, “from the prior to $z_{t}^{\theta}$” I think it should be $z_{0}^{\theta}$.**
>
> A.	We appreciate your detailed review! That’s what we intended to explain, and we have revised in the main paper as you suggested. Thank you.
> \
> \
> **Q. Convergence and non-convex optimization should be dealt differently.**
>
> A.	We agree with you. We have revised the sentence from “this variational gap converges to zero as we train …” to “the optimum of PDM guarantees that the generative distribution $p_{\phi^*,\theta}$ equals to the data distribution $p_{r}$, and the variational gap reduces to zero in the optimum.” Thank you for your sincere review.
>
> [1] Hoogeboom, Emiel, Taco S. Cohen, and Jakub M. Tomczak. "Learning discrete distributions by dequantization." arXiv preprint arXiv:2001.11235 (2020).

---

> > ### Comment · Reviewer_LLCZ · 2021-11-20
> > **Thank you**
> >
> > Thank you for the very detailed reply! Most of my questions have been answered adequately. I think the revised paper has much improved, especially in its clarity. However, I still think the novelty aspect of the paper is a bit weak compared to works like ScoreFlow despite the small difference in motivation. For that reason, I would like to keep my score unchanged.

---

> > > ### Author Response · Authors · 2021-11-22
> > > **Continued**
> > >
> > > One thing really confusing is that ScoreFlow also uses a normalizing flow. Regardless of the naming motivation of ScoreFlow, they have used a flow in order to reduce the dequantization gap. To explain what is the dequantization, suppose we estimate the data distribution with finite number of a given dataset. If the model distribution is flexible enough, then the model distribution would ends with the weighted delta distribution, i.e., $\sum_{i=1}^{N}w_{i}\delta_{\mathbf{x_{i}}}(\mathbf{x})$, after training. The problem is that this weighted delta distribution will have the negative infinite NLL (in bits-per-dimension scale). On the other hand, the information theory requires NLL to be strictly positive because NLL (in bits-per-dimension) is the average number of coin-tosses to find out each pixel value exactly. For instance, NLL of 3 means that it takes (averagely) three coin-tosses to find out pixel values out of [0,255]. The weighted delta distribution has the negative infinite NLL because the masses are concentrated in a few points. Dequantization stochastically perturb the given training dataset to avoid the trained distribution being concentrated in a few data points. Instead, dequantization technique eventually yields the trained distribution to be spread in the data space so that no degenerated delta distribution occurs.
> > >
> > > ScoreFlow uses a flow-based dequantization trick, called the variational dequantization. The variational dequantization uses an auxiliary normalizing flow $q_{\psi}$. Concretely, a normalizing flow transforms the data variable $\mathbf{x}\in [0,255]^{d}$ into a small dequantization variable $\mathbf{u}\in [0,1]^{d}$, and subsequently add $\mathbf{u}$ into $\mathbf{x}$. With this quantity $\mathbf{x}+\mathbf{u}\in [0,256]^{d}$, the diffusion model trains to estimate the data density. In loss, ScoreFlow optimizes $\int p_{r}(\mathbf{x})\int_{[0,1]^{d}} q_{\psi}(\mathbf{u}\vert\mathbf{x})\log{\frac{NELBO_{ScoreFlow}(\mathbf{x}+\mathbf{u})}{q_{\psi}(\mathbf{u}\vert\mathbf{x})}}\mathrm{d}\mathbf{u}\mathrm{d}\mathbf{x}$. The use of a normalizing flow in PDM is *completely* different, although it is similar at a glance. We uses a normalizing flow to directly transform $x\in [0,255]^{d}$ into a latent variable $z\in \mathbf{R^{d}}$, and we start our diffusion process with this latent $\mathbf{z}$. Indeed, PDM also applies dequantization tricks (because PDM is a diffusion model), so it applies the dequantized data $\mathbf{x}+\mathbf{u}\in [0,256]^{d}$ to a latent $\mathbf{z}$. If PDM uses the variational dequantization as ScoreFlow did, and if the auxiliary normalizing flow for the variational dequantization is denoted by $q_{\psi}$, then the loss becomes $\int p_{r}(\mathbf{x}) \int_{[0,1]^{d}} q_{\psi}(\mathbf{u}\vert\mathbf{x})\log{\frac{NELBO_{PDM}(\mathbf{x}+\mathbf{u})}{q_{\psi}(\mathbf{u}\vert\mathbf{x})}}\mathrm{d}\mathbf{u}\mathrm{d}\mathbf{x}$, where $NELBO_{PDM}(\mathbf{x}+\mathbf{u})$ is Eq. (7) of the revised paper.
> > >
> > > Summing altogether, the motivations of ScoreFlow and PDM are significantly different, and the contributions are not complementary each other. We argue that all contributions of PDM is tightly linked together from the introduction of normalizing flows jointly combined to diffusion models. We humbly request the reviewer to understand the difference of two methodologies.
> > > \
> > > \
> > > \
> > > [1] Song, Yang, et al. "Maximum Likelihood Training of Score-Based Diffusion Models." Thirty-Fifth Conference on Neural Information Processing Systems. 2021.
> > >
> > > [2] Song, Yang, et al. "Score-Based Generative Modeling through Stochastic Differential Equations." International Conference on Learning Representations. 2020.
> > >
> > > [3] Ho, Jonathan, et al. "Flow++: Improving flow-based generative models with variational dequantization and architecture design." International Conference on Machine Learning. PMLR, 2019.

---

> > > ### Author Response · Authors · 2021-11-22
> > > **Continued**
> > >
> > > Having said that NELBO is related with SDE and NLL is related with ODE, variational gap (between NELBO and NLL) quantifies how $p_{\theta}^{SDE}$ and $p_{\theta}^{ODE}$ differs. Table 2 of Song et al. [1] demonstrates that there is a huge gap between NLL and NELBO. This implies that the estimated probability flow ODE ($p_{\theta}^{ODE}$) and the estimated generative SDE ($p_{\theta}^{SDE}$) do not coincide each other. Song et al. [1] have not properly investigated this, and our work is focused on filling the gap.
> > >
> > > PDM introduces a normalizing flow to optimize $p_{T}$ so to satisfy $p_{T}=\pi$, which is equivalently formulated as learning the nonlinear diffusion process. This was never satisfied in ScoreFlow or any other diffusion models because $p_{T}$ is not optimizable due to the fixed forward diffusions in all prior works throughout the training procedure. In contrast, PDM provides a ground to optimize the final distribution $p_{T}$ towards $\pi$. This has not been introduced in ScoreFlow. In ScoreFlow, the forward diffusion has been limited to a fixed linear diffusion process, which attains the closed-form transition probability. The diffusion processes with closed-form transition probability are strictly limited to the family of linear diffusions, so expanding the forward inference diffusion process is a nontrivial step in the community of diffusion models.
> > >
> > > The second qualitative difference of PDM with ScoreFlow is that PDM optimizes its parameters more akin to MLE than ScoreFlow. Note that both ScoreFlow and PDM optimize their parameters through NELBO update, but the optimization of ScoreFlow is far from MLE because variational gap (between NELBO and NLL) is significant. On the other hand, to the best of our knowledge, PDM is the first diffusion model that updates the model parameters via MLE by reducing variational gap theoretically (Theorem 2) and empirically (Table 1 for CIFAR-10 and Table 2 for CelebA). Reduced variational gap implies that training PDM with NELBO is indeed optimizing NLL, and MLE update finally results in performing the state-of-the-art performance in CelebA.
> > >
> > > Third, from $p_{T}\ne \pi$ in ScoreFlow, even though $s_{\theta}(\mathbf{x_{t}},t)=\nabla_{\mathbf{x_{t}}}\log{p_{t}(\mathbf{x_{t}})}$ holds, the generative distribution of ScoreFlow ($p_{\theta}^{ODE}$) never coincides to the data distribution ($p_{r}$) by any chance. It is also true for the case of the generative distribution of SDE ($p_{\theta}^{SDE}$). On the other hand, Theorem 2 in our paper clarifies that our PDM model achieves the exact optimality, i.e., $p_{\theta,\phi}^{PDM}=p_{r}$, at any $\theta$ if $\phi$ becomes the optimal point. To the best of our knowledge, again, this is the first diffusion model that satisfies the optimality to the data distribution.
> > >
> > > The original paper only focused on the structural similarity between ScoreFlow and PDM. We emphasized in such a way in order to clarify the usage of a normalizing flow. ScoreFlow is clearly defined as an ODE model with *a previously proposed variational dequantization* [3]. Unlike our PDM, they have used a normalizing flow in their model only to reduce the dequantization gap using the variational dequantization, which is completely different from their naming motivation (SDE $\iff$ ODE). Their ScoreFlow naming was originated from conceptual exchangeability of diffusion models (SDE) and continuous normalizing flow models (ODE). In other words, Song et al. [1] named their model by ScoreFlow because the probability flow ODE [2], $\frac{\mathrm{d}\mathbf{x_{t}}}{\mathrm{d}t}=f(\mathbf{x_{t}},t)-\frac{1}{2}g^{2}(t)s_{\theta}(\mathbf{x_{t}},t)$, can be interpreted as a continuous normalizing flow. Having said that, ScoreFlow is completely different from our PDM because the major usability of a normalizing flow in PDM is optimizing the diffusion process nonlinearly. This is a nontrivial improvement from ScoreFlow, and neither one of ScoreFlow and PDM is a complementary of the other. To the best of our knowledge, PDM is the first paper that enables optimizing the (forward) inference diffusion process. ScoreFlow is limited to the linear inference diffusion, and their forward inference diffusion process are not trainable, in contrast to our model.

---

> > > ### Author Response · Authors · 2021-11-22
> > > **Thank you for the sincere review**
> > >
> > > We really thank you for your sincere review. Although we are really pleasing that your review is affirmative, we would like to answer to your novelty claim. Below is the answer to your concern to our paper.
> > >
> > > **Q. I still think the novelty aspect of the paper is a bit weak compared to works like ScoreFlow despite the small difference in motivation.**
> > >
> > > A.	**[Short Answer]** We ask the reviewer to consider our contribution one more time because we think that we may not have conveyed our contribution clearly. Our proposed model, **PDM, is different from ScoreFlow because only PDM can parameterize the nonlinear forward diffusion process**. This nonlinear forward diffusion by **PDM enables the tighter ELBO, which ScoreFlow cannot achieve**. This **tighter ELBO achieves the minimal variational gap, so it enables MLE update of the model parameters, which ScoreFlow cannot accomplish**. Also, PDM has significantly better performance in density estimation on CIFAR-10 than ScoreFlow. Finally, this parameterized nonlinear diffusion allows the State-Of-The-Art performance on CelebA compared to existing Deep Generative Models.
> > > \
> > > \
> > > **[Long Answer]**	The reviewer points out that there is small differences in motivation between ScoreFlow and PDM. However, we humbly argue that the differences are significant between two of them, so we describe the differences with details in below.
> > >
> > > We noticed that we have only focused on the structural differences between ScoreFlow and PDM in our first submission. To discuss the qualitative differences, we borrow the definition of ScoreFlow in Song et al. [1] as “*We term* $p_{\theta}^{ODE}$ *a ScoreFlow when its corresponding score-based model* $s_{\theta}(\mathbf{x},t)$ *is trained with likelihood weighting, importance sampling, and variational dequantization combined*.”
> > >
> > > Before we develop our argument, let us clarify what is $p_{\theta}^{ODE}$ in ScoreFlow. In Song et al. [2], the SDE with $\mathrm{d}\mathbf{x_{t}}=f(\mathbf{x_{t}},t)\mathrm{d}t+g(t)\mathrm{d}\omega_{t}$ is actually equivalent with a probability flow ODE of $\frac{\mathrm{d}\mathbf{x_{t}}}{\mathrm{d}t}=f(\mathbf{x_{t}},t)-\frac{1}{2}g^{2}(t)\nabla_{\mathbf{x_{t}}}\log{p_{t}(\mathbf{x_{t}})}$. In Song et al. [1], they pointed out that if we replace the data score into the score network, then the estimated generative process with SDE of $\mathrm{d}\mathbf{x_{t}}=[f(\mathbf{x_{t}},t)-g^{2}(t)s_{\theta}(\mathbf{x_{t}},t)]\mathrm{d}t+g(t)\mathrm{d}\bar{\omega_{t}}$ and the estimated generative process with ODE of $\frac{\mathrm{d}\mathbf{x_{t}}}{\mathrm{d}t}=f(\mathbf{x_{t}},t)-\frac{1}{2}g^{2}(t)s_{\theta}(\mathbf{x_{t}},t)$ create different models. Consequently, Song et. al. [1] defined $p_{\theta}^{SDE}$ as the marginal distribution at $t=0$ after solving the estimated generative SDE time reversely, and they defined $p_{\theta}^{ODE}$ as the marginal distribution at $t=0$ after solving the estimated generative ODE time reversely. Theorem 2 in their paper claims that $p_{\theta}^{ODE}$ and $p_{\theta}^{SDE}$ coincide each other if 1) $p_{T}=\pi$ and 2) $s_{\theta}(\mathbf{x_{t}},t)=\nabla_{\mathbf{x_{t}}}\log{p_{t}(\mathbf{x_{t}})}$. Furthermore, Theorem 2 in their paper proves that NELBO would coincide to NLL if the above two conditions are met. (Here, NELBO is the upper bound of NLL, and the continuous diffusion loss becomes NELBO if the weighting function is the likelihood weighting as in Theorem 1 of Song et al. [1].) After training, NLL of a diffusion model is obtained by solving the probability flow ODE, so it is the density of $p_{\theta}^{ODE}$; and NELBO of a diffusion model is obtained from the optimization loss, so it is the NELBO of $p_{\theta}^{SDE}$. When NELBO and NLL are similar, then $p_{\theta}^{ODE}$ and $p_{\theta}^{SDE}$ are similar, and we could exchange the trained SDE and ODE back and forth.

---

> ### Author Response · Authors · 2021-11-12
> **Thank you for the integrated feedback**
>
> Thank you for the detailed review and thoughtful feedback. Below we address specific questions.
> \
> \
> **Q. What is the difference of PDM to ScoreFlow [Song et al. 2021]?**
>
> A.	**[Different Modeling Purposes]** Actually, we anticipated this question, and we dedicated “Section 5. Related Work” to discuss the difference clearly in the original submission. Although there is a structural similarity between PDM and ScoreFlow at a glance in Figure 2 (PDM) and Figure 3 (ScoreFlow), we emphasize a couple of differences between the two of them. First, they differ by their purposes. ScoreFlow uses the normalizing flow to reduce the dequantization gap [1], while PDM minimizes the variational gap. ScoreFlow uses the flow component for optimizing the dequantization region on $[0,1)^{d}$, and the flow component contributes to the loss function by simply adding the sampled dequantization quantity ($\mathbf{u}$) to the quantized data ($\mathbf{x}$). However, our PDM jointly connects the flow and the diffusion models, and we optimize the jointly combined loss function suggested in Theorem 1.
>
> **[ScoreFlow Unable to Reduce Variational Gap]** Second, ScoreFlow is not a complementary model to PDM. Optimizing ScoreFlow loss does not reduce the variational gap (Theorem 2) because the normalizing flow in ScoreFlow does not affect to the forward diffusing mechanism. Rather, ScoreFlow still has the linear diffusing mechanism in the data space after the optimization. Therefore, PDM and ScoreFlow share similar structures, but they have significant differences in their modeling purposes and actual optimization results.
> \
> \
> **Q. Undefined notations, including $\mathcal{L}$ in Eq. (3) and $p_{0t}(\mathbf{x_{t}}\vert \mathbf{x_{0}})$ in the next line of Eq. (3), should be clarified.**
>
> A.	**[Definition of $\mathcal{L}$]** We thank for the detailed review. First, we define $\mathcal{L}(\mathbf{x_{0:T}},\lambda;\theta)$ as the diffusion loss function with $\mathcal{L}(\mathbf{x_{0:T}},\lambda;\theta)=\int_{0}^{T}\lambda(t)\mathcal{L_{t}}(\mathbf{x_{0:T}};\theta)\mathrm{d}t$ [1]. Here, $\lambda(t)$ is the weight function of diffusion loss at each time, and $\mathcal{L_{t}}(\mathbf{x_{0:T}};\theta)$ is the diffusion loss at time $t$. We denote $\mathbf{x_{0:T}}$ as the stochastic process of the forward diffusion because the middle brace { and } in markdown are not allowed inside the Equations.
>
> **[Definition of $p_{0t}$]** $p_{0t}(\mathbf{x_{t}}\vert \mathbf{x_{0}})$ (in the next line of Eq. (3)) is the transition probability of the diffusion model. When a diffusion starts from $\mathbf{x_{0}}$, $p_{0t}(\mathbf{x_{t}}\vert \mathbf{x_{0}})$ assigns the probability of the stochastically diffused data from $\mathbf{x_{0}}$ equals to $\mathbf{x_{t}}$. We note that our major novelty comes from observing this transition probability is intractable as the diffusion becomes nonlinear (Section 4.1). PDM detours this intractability issue by combining the flow and the diffusion models that enables the tractable training of the diffusion model with nonlinear diffusing mechanism.
> \
> \
> **Q. How is Eq. (6) related to diffusion?**
>
> A.	**[Connection of VAE to Diffusion Model]** The diffusion loss is the KL divergence $D_{KL}(\mu\Vert\nu_{\theta})$ between the (forward) inference path measure $\mu$ and the (reverse) generative path measure $\nu_{\theta}$. On the other hand, the VAE ELBO is the KL divergence $D_{KL}(q_{\phi}(\mathbf{x},\mathbf{z})\Vert p_{\theta}(\mathbf{x},\mathbf{z}))$ between the (forward) inference joint distribution $q_{\phi}(\mathbf{x},\mathbf{z})$ and the (reverse) generative joint distribution $p_{\theta}(\mathbf{x},\mathbf{z})$. The connection of the diffusion loss and the VAE ELBO would be unclear if we insist to view the VAE ELBO as the form of Eq. (4) or (5). From the point of Eq. (6), the diffusion model could be interpreted as modeling the intermediate data processing steps, which is the continuum ($\mathbf{x}_{0:T}$) of the discrete VAE that models only the coupling of the beginning ($\mathbf{x}$) and the final ($\mathbf{z}$) random variables. From this interpretation, we regard the diffusion model to be the generalization of the VAE model with multiple hierarchical latent variables at the expense of losing the dimension reduction property.
>
> **[Measure Theoretic View]** We acknowledge that the analysis based on path measure is hard to follow, but it is inevitable to introduce such measure theoretic analysis in order to construct the theory of diffusion models with mathematical rigour. Eq. (6) restates the VAE ELBO in terms of the stochastic process, which is a proper language to the measure theory.

---

### Official Review · Reviewer_4sPP · 2021-11-02

**Correctness:** 4
**Technical Novelty And Significance:** 2
**Empirical Novelty And Significance:** 2
**Recommendation:** 5
**Confidence:** 4

**Main Review:**

**Strengths**:
- Combining a Normalizing Flow with a Diffusion Model as proposed in the paper has not been done before.
- The model allows to calculate exact data log-likelihoods using the diffusion model's probability flow ODE together with the tractable log-likelihood of the Normalizing Flow component.
- The deterministic transformation defined by the Normalizing Flow component allows to formally relate the combined flow and diffusion model to a non-linear diffusion directly in data space via Ito's Lemma.
- It is interesting that the method results in small variational gaps between log-likelihoods and evidence lower bounds used for training.

**Weaknesses**:
- It is mathematically elegant that we can formally relate the method to a non-linear diffusion in data space via Ito's Lemma. But what is the practical value of that? Both training and sampling still decompose the model into its separate flow and diffusion components and do not leverage this connection. Also, the experimental results suggest that the model only reduces variational gaps, but does not improve the actual likelihood itself (for example, *DDPM++ (IS)* vs. *PDM (NLL, dec)* in Table 3, also see points below).
- The method is very incremental compared to LSGM [1]. It is essentially equivalent with the only difference being that LSGM uses a VAE framework with separate, non-invertible encoder and decoder, while PDM uses an invertible neural network, so encoder and decoder are simply the inverse of each other. However, this has multiple downsides: (a) Invertible neural networks have significantly lower expressivity than free neural networks due to the invertibility constraint. Hence, PDM has less modeling power than LSGM. (b) Using free neural networks for encoder and decoder, LSGM allows to use smaller or larger latent spaces, which can affect sampling speed, and it also allows to tailor the encoder and decoder better to different data types. No such things are possible with Normalizing Flows. Compared to LSGM, the only advantage of the invertible and deterministic encoder/decoder networks (deterministic in contrast to VAEs, where encoder and decoder formally define distributions) is that we can use Ito's Lemma to define a non-linear diffusion in data space. However, as mentioned, this advantage is not practically leveraged in any way, it seems, and the model is still trained separating the regular flow and the regular linear latent-space diffusion.
- Related to that, I think the statement "LSGM cannot provide the diffused data learning at the middle of the process because LSGM does not induce the diffusion process on the data space due to the lack of VAE’s invertibility" is a bit misleading. While it is true that the encoder/decoder are not formally exact inverses of each other, in practice they do almost perfectly invert each other as indicated by the high reconstruction quality of modern VAEs and behave almost like deterministic functions. Therefore, also a model like LSGM can easily take the latent variable from anywhere along the diffusion process and map it back to data space with the decoder, just like PDM. In fact, such an experiment is demonstrated in Figure 15 in [1] (therefore, also the strict comparison between the pipelines in Figure 2 and 3 is a bit questionable).
- I think the paper misses some important baselines on the CIFAR-10 experiments: Recently, [2] achieved state-of-the-art log-likelihoods using diffusion models. This work should be cited and would also be an appropriate baseline. Furthermore, when comparing to [1], the authors choose only one setup that was tailored to very high FID at the cost of likelihood. However, PDM not only focuses on FID, but primarily at NLL, lower bounds, and smaller variational gaps. Hence, it would be more appropriate to also compare to *LSGM (NLL)* and *LSGM (balanced)* (Table 2 in [1]).
- The experimental results are not overly strong. Considering these further baselines, the strongest competitive diffusion-based works [1,2,3] all achieve likelihoods or lower bounds around 2.90 or lower, while PDM's best value is 3.11. Also the FID is significantly higher than recent works in the literature. I don't think that only demonstrating small variational gaps is a significant result, as long as the actual performance is not competitive. Note that I am giving more weight to the CIFAR-10 experiments, since this is a widely used benchmark within the relevant literature (i.e. diffusion models), while CelebA 64x64 is not that widely used.

**Questions**:
- The NLL for *PDM (FID, dec)* is actually better than the NLL of *PDM (NLL, dec)* in Table 3. Why is that? Wouldn't we expect to achieve better likelihoods when optimizing with the "likelihood" weighting $\lambda(t)=g^2(t)$?
- Why does *PDM (NLL, dec)* achieve so much worse NLL/NELBO than *DDPM++ (IS)*? My understanding is that PDM uses a similar DDPM++ backbone, which is trained first (after this training stage, it should have similar NLL/NELBO as the pure diffusion *DDPM++ (IS)* baseline). Now, we additionally train the flow. Shouldn't it only further improve the results?
- Based on Figure 7 and as discussed in the text, it seems the flow component's main job is to adjust colour saturation of the synthesized images. I am wondering whether this is related to the training strategy: In the appendix, it is mentioned that first the diffusion component is trained and only then the flow is added. Hence, the flow will afterwards only improve or "clean up" whatever the diffusion model missed and this seems to be related to colour saturation. What if we trained the other way around and first trained the flow to transform the data distribution into a Normal distribution (standard flow training), and then trained the diffusion model on the actually achieved embedding distribution? We may see different results here. Also, what would happen if we do not do stage-wise training but directly trained all components end-to-end simultaneously from scratch?
- I am not sure I am fully understanding the value of Theorem 2: If we assume that our score model, i.e. diffusion model, is flexible enough to cover any path measure, then why do we require a flow component in the first place? With this assumption, the diffusion model should be able to model the data distribution itself (diffused for all $t$) perfectly without requiring an additional flow component, I would think?

**Minor Comments**:
- Equation 4 seems to be the negative ELBO, not ELBO.
- In Section 3.2, the paper discusses the variational gap and refers to previous works on diffusion models. Previous work showed that the generative distribution defined by the probability flow ODE and the stochastic SDE formulation are not identical, unless the model has learnt the ground truth score perfectly, which is not the case in practice (see [3]). Note that the base for the KL derivations is the SDE formulation, while the NLL calculation in practice relies on the probability flow ODE formulation. I think we are essentially assuming here that ODE and SDE-based models are similar. I believe this subtlety should be highlighted.
- Page 8, last sentence: Did the authors mean "destabilize", rather than "stabilize"?
- Section 9: "prodive" -> "provide".

[1] Vahdat et al., "Score-based Generative Modeling in Latent Space", 2021.

[2] Kingma et al., "Variational Diffusion Models", 2021.

[3] Song et al., "Maximum likelihood training of score-based diffusion models", 2021.

**Summary Of The Paper:**

The paper proposes to combine Normalizing Flows with generative Diffusion Models in such a way that the target data is first non-linearly transformed via the flow and then the distribution over latent embeddings is modeled with the diffusion model. The authors call their model *Parametrized Diffusion Model (PDM)*. Since the flow is invertible and defined by a deterministic function, it is possible to formally relate the combined flow and diffusion process to a non-linear diffusion directly in data space, leveraging Ito's Lemma. The paper validates the PDM on image modeling benchmarks (CIFAR-10 and CelebA 64x64). The experiments show that the method leads to reduced variational gaps between the data likelihood and the evidence lower bound that is used for training the model, compared to several baselines.

**Summary Of The Review:**

It is interesting that the flow transformation allows for a formal definition of a non-linear diffusion in data space via Ito's Lemma, but it seems this cannot be leveraged for improved modeling power, as directly training this non-linear diffusion is not possible, unfortunately. Hence, the paper resorts to training a regular flow together with a regular "linear" diffusion, separating the two components. With this in mind, I think methodologically the paper is fairly incremental compared to LSGM [1]. It is basically the same with the encoder/decoder replaced with invertible neural networks and the invertibility does not provide any practical advantages. The paper's experimental results are not impressive and I think some relevant baselines are missing. Therefore, in conclusion I recommend rejection.

---

> ### Author Response · Authors · 2021-11-12
> **Continued**
>
> **Q. *The NLL for *PDM (FID, dec)* is actually better than the NLL of *PDM (NLL, dec)* in Table 3. Why is that?***
>
> A.	**[Revised Experiments]** As we explained in Appendix C.3, we first apply $\sigma^{2}$ on the initial phase of training for the diffusion model, and we apply $g^{2}$ on the later phase in *PDM (NLL, dec)*. This is the reason of *PDM (NLL, dec)* performing similar to *PDM (FID, dec)*. To compare with the baseline *DDPM++ (NLL)* in a fairer setting, we retrained the model with only applying $g^{2}$, and the revised Table 3 (Table 1 of the revised paper) reports that *PDM (NLL)* experiments all outperforming *PDM (FID)* experiments in density estimation. Please refer Table 1 of our revised paper.
> \
> \
> **Q. *Why does PDM (NLL, dec) achieve so much worse NLL/NELBO than DDPM++?***
>
> A.	**[Performance Gain Exists]** The DDPM++ performances in the revised Table 3 (Table 1 in the revised paper) are the reported performance in [3]. We found that the reported DDPM++ performance is not perfectly replicable particularly in density estimation, so we compare our PDM with the pretrained DDPM++ in Table 7 of the revised paper. Our pretrained DDPM++ performs NLL of 3.03 after the saturation, and the additional training of PDM reduces this NLL to 2.94. Also, PDM performs NELBO of 2.95 after the training, while the pretrained DDPM++ performs 3.20 for NELBO. We conclude that additional training indeed gives additional performance gain.
> \
> \
> **Q. *The experimental results are not overly strong. It would be more appropriate to also compare to LSGM (NLL) and LSGM (balanced).***
>
> A.	**[Comparison to LSGM-109M]** Table 1 in the revised version presents new experimental results (see revised Appendix for the details). We emphasize that all the experimental results are based on the uniform dequantization. Our PDM performs NLL of 2.94 and NELBO of 2.95, which is comparable to LSGM-109M (balanced) NELBO of 2.96 as reported in [4] with 109M number of parameters. Here, the baseline diffusion model [5] uses 108M number of parameters, and our PDM model uses 120M number of parameters.
>
> **[Comparison to LSGM-480M]** The ablation study on the tolerance for the probability flow ODE in Table 7 of the revised paper shows that NLL of *PDM (NLL, dec, IS)* is 2.86 with tolerance of 1e-1. Note that this experimental result outperforms the density estimation performance of 2.87 for *LSGM-480M (NLL)*, which is believed enough to be strong. In addition, we emphasize that *PDM (NLL, dec, IS)* performs slightly better FID score compared to *LSGM-480M (NLL)* with a quarter number of parameters.
>
> **[Comparison to LSGM]** We also acknowledge that LSGM could be a good model to compare with, and we provided our discussion comparing PDM and LSGM to highlight its non-invertibility of LSGM and its consequential incapability on the inference diffusion mechanism, above. While we accept the constructive comparison by reviewers, we also note that LSGM is released on Jun 10, 2021, which also makes LSGM as a contemporary development. Given this guideline and similarity, we ask the reviewer to focus on the theoretic development, not the experimental performance, which could have been improved if there were more time.
>
> **[Performance on CelebA]** Also, we have claimed that PDM achieved the best FID performance in CelebA 64x64, and this is the state-of-the-art performance. We discussed the reason behind PDM's advantage particularly in high-dimensional sample generation in Section 6.1. Please count this performance on your evaluation.
> \
> \
> \
> [1] Kingma, Diederik P., et al. "Variational diffusion models." arXiv preprint arXiv:2107.00630 (2021).
>
> [2] Grcić, Matej, Ivan Grubišić, and Siniša Šegvić. "Densely connected normalizing flows." arXiv preprint arXiv:2106.04627 (2021).
>
> [3] Song, Yang, et al. "Maximum Likelihood Training of Score-Based Diffusion Models." arXiv e-prints (2021): arXiv-2101.
>
> [4] Anonymous, “Score-Based Generative Modeling with Critically-Damped Langevin Diffusion.” Under review in ICLR 2022, https://openreview.net/forum?id=CzceR82CYc
>
> [5] Song, Y., Sohl-Dickstein, J., Kingma, D. P., Kumar, A., Ermon, S., and Poole, B. (2020). Score-based generative modeling through stochastic differential equations. arXiv preprint arXiv:2011.13456.

---

> > ### Comment · Reviewer_4sPP · 2021-11-19
> > **Thank you for the reply.**
> >
> > I appreciate the authors’ detailed reply and would like to thank them for their efforts. Unfortunately, I do not entirely agree with all points:
> > - My understanding is that Ito’s Lemma does not require invertible functions, but deterministic functions (and mild smoothness conditions). The reason why this works here is not invertibility, but the fact that Normalizing flows define deterministic mappings (as opposed to the encoders and decoders of VAEs, which define distributions - albeit usually extremely sharp ones, as discussed).
> > - I disagree with or don’t understand the argument with regards to the inference part and LSGM. Inference vs. generation in a diffusion model amounts to running the forward diffusion vs. running the reverse denoising process. In both cases, we can leverage the decoder, which is not fully but almost deterministic, to map back to data space and get corresponding “data space inference or generation paths”.
> > - I checked the modified theorem 2: We know that flow networks are not very flexible due to the invertibility constraints - in fact the authors mentioned that themselves. This is a well-known issue of Normalizing flows. Hence, an assumption like “Suppose a flow network is flexible enough to transform $p_r$ to arbitrary continuous distribution” seems contradictory and does not hold in practice.
> > - The mismatch between the standard prior leveraged during sampling (“starting distribution”) and the fully diffused distribution during the forward diffusion is usually extremely tiny. In fact, that is how diffusion models are designed. See, for example, Ho et al. [1], which makes sure that the KL is $\approx10^{-5}$  bits per dimension. I am not convinced that this tiny remaining mismatch plays any significant role here and would require experimental evidence to be convinced otherwise.
> >
> > I would like to apologize for not being aware of the exact dates regarding what should be considered concurrent work. Therefore, I am okay with not quantitatively comparing to VDM [2]. I would still suggest the authors to add a pointer to such relevant concurrent works in their related works section, though. With regards to LSGM, which was made public only few days after the cutoff date for concurrent works, I believe it is the most relevant and related work and does deserve a detailed comparison. Since the authors did discuss and compare to LSGM in their original version, it seems acceptable to me to take this into account.
> >
> > Overall, I appreciate the effort by the authors, the further explanations and the improved results in the revised version. I still have reservations, as I don’t agree with the authors about all points, but I raised my score from 3 to 5.
> >
> > [1] Ho et al., "Denoising Diffusion Probabilistic Models", 2020.
> >
> > [2] Kingma et al., "Variational Diffusion Models", 2021.

---

> > > ### Author Response · Authors · 2021-11-20
> > > **Continued**
> > >
> > > **Q. Hence, an assumption like “Suppose a flow network is flexible enough to transform $p_{r}$ to arbitrary continuous distribution” seems contradictory and does not hold in practice.**
> > >
> > > A.	The reviewer questions that our assumption of Theorem 2 is too ideal by claiming normalizing flow as being flexible enough to transform the data distribution to an arbitrary continuous distribution.
> > >
> > > Given diffusion models are merged to other types of deep generative models, we think that the innovation of synergies from two types of models should not be limited by shortcomings of sub-components. For instance, VAE is neither a deterministic model nor a perfect reconstruction model. VAE is known to have a poor reconstruction performance, so VAE cannot be used as an inverse function. However, VAE+Diffusion provides a certain value, so LSGM becomes published. Similarly, Flow+Diffusion as our PDM provides the first parameterization on the forward diffusion process. Whereas we honestly noted the limitation of normalizing flow in terms of empirical performances, this does not limit the innovation made by normalizing flow. We expect the normalizing flow performance will become better as the VAE will be.
> > >
> > > Regardless of whether it is satisfied in practice or not, theoretic analyses always assume the idealized situation. If this idealized situation is not the subject of endeavor to overcome, then the theory is useless. Think about the well-known “Universal Approximation Theorem” that gives a guarantee that large portion of functions can be approximated by a single layered neural network. Finding such a neural network was highly impractical at the time the theorem was suggested, but it was valued by its theoretic ground of deep learning then and now. Analogously, our assumption may seem to be too ideal to the reviewer, but as the normalizing flow community grows, more flexible networks would satisfy our assumption.
> > > \
> > > \
> > > **Q. The mismatch is usually extremely tiny. I am not convinced that this tiny remaining mismatch plays any significant role here and would require experimental evidence to be convinced otherwise.**
> > >
> > > A.	The reviewer questions whether the tiny mismatch at the prior distribution is serious, or not. The prior mismatch itself could be tiny value. However, the prior mismatch invokes much more significant gap in training diffusion model from deviating MLE.
> > >
> > > We quantify the “gap” to MLE training with “variational gap” in Table 1, which shows successful MLE training in our PDM by reducing the gap from 0.19 in *DDPM++ (NLL, IS)* to 0.01 in *PDM (NLL, dec, IS)*. Particularly, this is important because NLL (in bpd scale) is around 3.0, so the variational gap becomes significant if it is measured by 0.19.
> > >
> > > While we show the significant variational gap between NLL and NELBO, the reviewer may also wonder whether this gap is meaningful or not in its practical usage. Our results in Tables 1 and 7 of the revised version show that NLL/NELBO of PDM are better that those of LSGM. Also, we show the state-of-the-art FID performance on CelebA in Table 2. We interpret this success comes from the tightened NELBO to NLL, as discussed by “Second, our PDM report … datasets of higher dimensions” in Section 6.1 of the revised paper. Again, we repeat that PDM enables the nonlinear forward diffusion, and the nonlinear forward diffusion enables the tightened NELBO, and the MLE training with tightened NELBO enables the state-of-the-art FID on the CelebA dataset.

---

> > > > ### Comment · Reviewer_4sPP · 2021-11-25
> > > > **Thank you for reply.**
> > > >
> > > > I would like to thank the authors for their further explanations and acknowledge that I have read them. However, my raised concerns mostly remain. Therefore, I will keep my score.

---

> > > ### Author Response · Authors · 2021-11-20
> > > **Thank you for the sincere review**
> > >
> > > We appreciate for your sincere review service. In below, we provide additional explanations on the questions raised by the reviewer.
> > >
> > > **Q. I disagree with or don’t understand the argument with regards to the inference part and LSGM. We can leverage the decoder to map back to data space and get corresponding “data space inference or generation paths”.**
> > >
> > > A. As the reviewer pointed out, we could construct two diffusion processes in the data space on LSGM using the decoder mapping: *generative diffusion in data* and *inference diffusion in data*. *generative diffusion in data* transforms the latent generative diffusion process back to the data space through the decoder network, and *inference diffusion in data* could transform the latent forward diffusion process to the data space by the decoder network as proposed by the reviewer.
> > >
> > > The major difference of *inference diffusion in data* between PDM and LSGM is two different initial random variables: $p_{r}$ for PDM; and $\hat{p_r}=\text{Dec}(\text{Enc}(p_{r}))$ for LSGM. To describe further, let us clarify the reasoning by equations. In PDM, if the forward diffusion process in the latent space is $\mathrm{d}z_{t}=f_{latent} (z_{t},t)\mathrm{d}t+G_{latent}(t)\mathrm{d}\omega$ with initial variable of $z_{0}\sim \text{Flow}(p_{r})$, then the (forward) inference diffusion process in the data space becomes $\mathrm{d}x_{t}=f_{data}(x_{t},t)\mathrm{d}t+G_{data}(x_{t},t)\mathrm{d}\omega$ that starts from $x_{0}\sim \text{Flow}^{-1}(\text{Flow}(p_{r}))=p_{r}$.
> > >
> > > On the other hand, in LSGM, if the forward diffusion process in the latent space is defined by $\mathrm{d}z_{t}=f_{latent}(z_{t},t)\mathrm{d}t+G_{latent}(t)\mathrm{d}\omega$ starting from $z_{0}\sim \text{Enc}(p_{r})$, then *inference diffusion in data* becomes $\mathrm{d}x_{t}=f_{data}(x_{t},t)\mathrm{d}t+G_{data}(x_{t},t)\mathrm{d}\omega$ with the initial random variable to be $x_{0}\sim \text{Dec}(\text{Enc}(p_{r}))$. Therefore, the starting random variable of PDM is the data distribution ($p_{r}$), but that of LSGM becomes $\text{Dec}(\text{Enc}(p_{r}))$, which is the regenerated data distribution ($\hat{p}_{r}$).
> > >
> > > It is well-known that $p_{r}$ and $\hat{p_{r}}$ are not similar to each other because of the KL divergence regularization and the NELBO approximation. Ultimately, NELBO is not NLL, so the perfect reconstruction is unlikely. Hence, we claim that PDM starts from the data distribution ($p_{r}$), but LSGM does not start from the same $p_{r}$. While we may accept that this regenerated data distribution is close to the data distribution, we could argue that *inference diffusion in data* is not a diffusion process of the *data distribution*. Theoretically, from the *Trilemma of Inversion*, the decoder cannot be an inverse of the encoder, so the regenerated data distribution cannot estimate the exact data distribution. From this reasoning, we cannot simply regard “inference diffusion in data” in LSGM constructed with the decoder network as a diffusion process of the data distribution.
> > >
> > > Moreover, note that the original motivation of *inference diffusion in data* is that we train the nonlinear forward diffusing mechanism by training the networks. However, in the middle of the training procedure, the regenerated data distribution ($\hat{p_{r}}$) is not similar to the data distribution ($p_{r}$) because of the immature training. Therefore, it is problematic to simply regard *inference diffusion in data* with the decoder network as the nonlinear forward diffusion process in the data space, in the middle of the VAE training; and it is inappropriate to interpret LSGM as an algorithm that trains the inference diffusion process in a nonlinear fashion. Under the same assumption of the middle of training procedure, the flow itself can be invertible at any level of training maturity, so the immature training does not incur the mismatch of the initial variables (i.e., $\hat{p_{r}}\ne p_{r}$) because of the flow invertibility at any moment of training.

---

> ### Author Response · Authors · 2021-11-12
> **Continued**
>
> **Q. *I am not sure I am fully understanding the value of Theorem 2.***
>
> A.	**[Optimality of vanilla diffusion model]** The NELBO is minimized if the score network perfectly estimates the data score: $s_{\theta}(\mathbf{x_{t}},t)=\nabla\log{p_{t}(\mathbf{x_{t}})}$. However, this optimal score does not guarantee the generative distribution $p_{\theta}$ to be equal to the data distribution $p_{r}$ because the generative process and the reverse diffusion has exactly identical denoising process at the optimality, but the starting distributions differ by $\pi$ (generative) and $p_{T}$ (reverse) in Eq. (2) and (1), respectively. Therefore, the generative distribution $p_{\theta}$ is not equal to $p_{r}$ even in the optimal score. This is an important problem of any diffusion models that has not been addressed formally by any researcher to the best of our knowledge.
>
> **[Optimality of PDM]** In contrast to the vanilla diffusion model, Theorem 2 shows that for any score parameter $\theta$, there exists optimal flow network $\phi^*$ that makes the generative distribution $p_{\phi^{*},\theta}$ equal to the data distribution $p_{r}$. This means that the diffusion model solely cannot reach to the data distribution, but with the aid of normalizing flows, there are always optimal nonlinear diffusing mechanisms that yield the data distribution. Therefore, Theorem 2 implies that PDM solves the problem mentioned in the above paragraph, which makes our work genuine. We added further explanation in Appendix A to provide the interpretation on Theorem 2 and associated illustration for the potential problem of the diffusion models without the normalizing flow.
>
> **[Issue on Statement]** There are minor issue in the first sentence of Theorem 2 as the reviewer has pointed out. The sentence means that the normalizing flow is flexible enough to transform the data distribution to arbitrary continuous distribution. Please check the modified statement in the revision.
> \
> \
> **Q. *This work (VDM [1]) should be cited and would also be an appropriate baseline.***
>
> A.	**[Code Release Required]** We are willing to add VDM [1] as a baseline. However, there are two difficulties to cite VDM: first, the score architecture of VDM is different from the baseline score architecture we used in the experiment. To compare PDM and VDM in a fair setting, we need to train PDM with VDM’s network architecture. We could simply compare PDM to VDM, but we think such unfair comparison does not imply anything. Rather, our primary focus is on the mathematical formulation of the nonlinear diffusing mechanism, so the scope of this paper is far from showing state-of-the-art performances. Second, we could not regenerate the performance of VDM in our implementation with only the released information on its paper. Therefore, we would compare PDM to VDM either if 1) the official code is released, or if 2) our replicated code regenerates the claimed performance on its paper.
>
> **[ICLR Reviewer Guideline]** Furthermore, I ask reviewer to consider the *Reviewer Guideline of ICLR 2022* (https://iclr.cc/Conferences/2022/ReviewerGuide). The guideline clearly states that “authors are not required to compare their own work to that paper”, and “that paper” is a paper published after June 5, 2021. According to Arxiv (https://arxiv.org/abs/2107.00630), VDM has been publicly available since Jul 1, 2021. Therefore, the guideline says that we are not required to compare our work to VDM.
> Having said that, we are more than willing to compare our model to VDM, but they have not released any codes or details other than one-page implementation details, and we cannot regenerate their results with the given information.
> \
> \
> **Q. *It seems the flow component’s main job is to adjust colour saturation of the synthesized images. I am wonder whether this is related to the training strategy.***
>
> A.	**[Natural Role Separation]** We observe that PDM with pretrained flow (or training both flow/diffusion from scratch) acts in a similar way: the diffusion captures the global texture, and the flow learns coloring the sampled global context. To provide the reasoning of such role separation, recall that the major source of poor sample generation in normalizing flows lies in the poor global context. In other words, the samples from normalizing flows in general colorize the image well, but a generated sample is hardly perceived as a real image. For instance, the best normalizing flow [2] creates unrealistic samples in Figure 8 of [2] (FID 34.9). This is the fundamental issue in normalizing flows arising from the inflexible expressive power induced by the invertibility condition. Having said that normalizing flows have weakness on capturing the global context, diffusion models do not suffer from this weakness. Therefore, optimizing the combined model naturally separate their roles in image generation: the flow model colorizes the latent, and the diffusion model focuses more on capturing global context.

---

> ### Author Response · Authors · 2021-11-12
> **Thank you for the integrated feedback**
>
> Thank you for the detailed review and thoughtful feedback. Below we address specific questions.
> \
> \
> **Q. *While it is true that the encoder/decoder are not formally exact inverses of each other, in practice they do almost perfectly invert each other.***
>
> A.	**[Trilemma of Inversion]** We admit that the encoder/decoder are practically inverting each other. However, it is important to note that “There is no continuous, invertible map from $\mathbf{R^{n}}$ to $\mathbf{R^{m}}$ if $n\ne m$.” Indeed, we can say a bit more: “No function satisfies all three properties: continuity, invertibility, and different dimensionality between its domain and codomain.” From it, we conclude that a (continuous) neural network should attain the same dimensionality in order to keep the invertibility. I think that the reviewer is already aware of this theoretic property from the basic calculus.
>
> **[VAE NOT Invertible]** Theoretically, VAE is not invertible because of the above mathematical fact, as long as the dimension of the latent differs from the data dimension, which is equivalent to the normalizing flow setting. In practice, we acknowledge that VAE almost perfectly inverts the data and the latent spaces, but the invertibility in theory never holds.
>
> **[Contribution of Mathematical Inversion]** Our lens become different as we talk the same objective from either practical or theoretical perspective. Theoretically, the proposed model, PDM, is different from LSGM due to the theoretic invertibility and consequential benefit of the application of Ito’s lemma. Practically, the application of Ito’s lemma enables the inversion of the latent space at any moment during the diffusion process, which contributes generating Figure 2 in the main paper. Particularly, it should be noted that we provide the diffusion process at the inference process of PDM, which cannot be generated by LSGM. Reviewer’s suggestion (LSGM can produce the diffusion process image, i.e. Figure 15 in LSGM paper) is only true for the generation process, not the inference process. It should be noted that LSGM is incapable of creating a forward diffusion in the inference process. See the answer to **Q. *Therefore, a model like LSGM can easily take the latent variable from anywhere along the diffusion process and map it back to data space with the decoder, just like PDM.***
> \
> \
> **Q. *Therefore, a model like LSGM can easily take the latent variable from anywhere along the diffusion process and map it back to data space with the decoder, just like PDM.***
>
> A.	**[No Inference Data Diffusion in LSGM]** When you consider the generative process of LSGM, it is correct that the diffusion process in the latent space can be lifted up to the diffusion in the data space by decoder mapping. However, if you think about the forward inference procedure, the diffusion process in the latent space cannot be lifted up to the data space because the inference network is not invertible as discussed in **Q. *While it is true that the encoder/decoder are not formally exact inverses of each other, in practice they do almost perfectly invert each other.***
>
> **[Qualitative Difference]** Therefore, LSGM and PDM are qualitatively different because LSGM is actually not diffusing the data space at the encoding part. In other words, Figure 15 in [1] is the generation process in the data space, but the diffusing process at the encoding part cannot be sketched in the data space. In contrast, Figure 2 in our paper illustrates the data diffusing at the inference step. This leads the argument of PDM original.
> \
> \
> **Q. *Both training and sampling still decompose the model into its separate flow and diffusion components and do not leverage this connection.***
>
> A.	**[PDM Loss is Jointly Leveraging the Connection]** If we were to accept the reviewer’s argued perspective, LSGM does not leverage this connection, either, which is not true. As Eq. (7) shows, the latent variable of $\mathbf{z_{0}}$ is simultaneously utilized by both flow (the first term of Eq. (7), note that $\mathbf{h_{\phi}}$ is $\mathbf{z_{0}}$ by definition) and diffusion (the second term of Eq. (7)). Therefore, the PDM loss of Eq. (7) cannot be simply decomposed into two parts, as LSGM cannot be decomposed, either.
>
> **[Interpretation is the Value of PDM]** The difference between our PDM and LSGM is that *only PDM* is interpretable as imposing data-adaptive nonlinear diffusing mechanism in the data space. The diffusion parameters are updated through the lens of NELBO, which is not the MLE update. Our model optimizes the nonlinearity of the diffusing mechanism (in the data space) that eventually reduces the variational gap, so the parameter update becomes more akin to MLE. On the contrary, LSGM cannot be inspected to check the variational gap because the VAE structure in LSGM does not allow to compute tractable NLL within reasonable computational cost. This is the major different between LSGM and PDM.

---

### Author Response · Authors · 2021-11-18
**Additional comments by authors**

Dear all reviewers:

**[Comparison with LSGM]** We acknowledge that LSGM could be a good model to compare with, and we provided our discussion comparing PDM and LSGM to highlight its non-invertibility of LSGM and its consequential incapability on the (forward) inference diffusion mechanism. While we accept the constructive comparison by reviewers, we also note that LSGM is released on Jun 10, 2021 (in arXiv), which also makes LSGM as a contemporary development. Given the *Reviewer guideline of ICLR 2022* (https://iclr.cc/Conferences/2022/ReviewerGuide), we ask reviewers to focus on the theoretic development, not the experimental performance, which could have been improved if there were more time.

**[SOTA FID in CelebA]** Reviewers point out that PDM shows weak performance. On CIFAR-10, we present the revised performance with improved NLL (bits per dimension) in Tables 1/7/8. Also, Table 2 claims that PDM achieves the best FID performance in CelebA 64x64, and this is the state-of-the-art performance. We discussed the reason behind PDM's advantage particularly in high-dimensional sample generations in Section 6.1. Please count this SOTA performance on evaluation.

Thanks,

Authors.

---

### Author Response · Authors · 2021-11-20
**Additional comments on SOTA performance in CelebA**

Dear all reviewers:

We request reviewers to consider that PDM achieves the State-Of-The-Art (SOTA) FID performance in CelebA. We describe the reasoning of the excellent performance on CelebA in below.

**[Reasoning of SOTA performance in CelebA]** Our PDM reduces the variational gap by training the nonlinearity of the (parametrized) forward diffusion process in the data space. This reduced variational gap induces the MLE training of our model. In particular, the variational gap turns out to be strictly positive for the vanilla diffusion model, whereas our PDM could make the gap to be zero, evidenced both in theoretically (Theorem 2) and in empirically. This MLE training directly minimizes the KL divergence between the data distribution ($p_{r}$) and the generative distribution ($p_{\theta}$) at time $t=0$, so it guarantees better training performance. Specifically, our experiment shows that training the parametrized forward diffusion is more effective in CelebA 64x64 than CIFAR-10 32x32 because the data-adaptive optimal diffusing mechanism is more likely to deviate from linear diffusions in a higher-dimensional dataset. In other words, the merit of training the nonlinear diffusion weigh more on higher-dimensional dataset because the data-adaptive optimal diffusing mechanism is likely to be more nonlinear as dimension increases. This is the major reason of the SOTA FID performance in the CelebA dataset.

Thanks,

Authors.

---

### Decision · Program_Chairs · 2022-01-20

**Decision:**

Reject

**Comment:**

This paper presents a simple approach called PDM for composing non-linear and complex normalizing flows with score-based generative models. Since score-based models can be considered as a special form of continuous-time normalizing flows, PDM corresponds to a composition of different classes of normalizing flows.

Pros:
* Combining generic normalizing flows with score-based models is an interesting direction as they have different characteristics and can be complementary to each other.
* Using Ito's lemma to show that the model learns a non-linear SDE in data space is valuable.
* The authors show that the variational gap can be reduced using normalizing flows.

Cons:
* The proposed method does not exhibit a clear advantage compared to the diffusion baseline without the normalizing flow component. On the CIFAR10 dataset, the best NLL and FID results are obtained by the diffusion baseline.

* Theorem 2 makes a very unrealistic assumption that a flow network is flexible enough to transform $p_r$ to any arbitrary distribution. If this holds, we wouldn't need the score-based generation model anymore. We could simply train the normalizing flow to map the input data distribution to a Normal distribution.

* This submission chooses to discuss differences with the recent LSGM framework. However, in doing so, several inaccurate claims are made. The lack of inference data diffusion in LSGM is mentioned as one of its drawbacks. However, it is not clear what is the value of having such a mechanism and what implications it may have on the expressivity of the model. Note that mapping from data space to latent space in VAEs can be considered as a stochastic inversion rather than an exact inversion. Ito's lemma does not require invertibility and it can be easily applied to the forward and generative diffusion in LSGM. The authors argue that applying it to the forward diffusion in LSGM will result in $\hat{p_{r}}\ne p_{r}$. But, $\hat{p_{r}}$ would be only considered for visualization of the forward diffusion and it is not used for training or any other purposes. LSGM, the proposed PDM, and score-based models are all trained with a reweighting of ELBO (see [here](https://arxiv.org/abs/2106.02808)). It is not clear if the drawback mentioned above has an impact on the training or expressivity of the model.

* The presentation in the paper requires improvement. The motivation on why invertibility plays a key role is not clear beyond generating the visualization in Figure 2.

In summary, the paper proposes an interesting idea and explores directions very relevant to the current focus in generative learning. However, given the concerns above, we don't believe that the paper in its current form is ready for presentation at ICLR.